# Insights into the inhibition of type I-F CRISPR-Cas system by a multifunctional anti-CRISPR protein AcrIF24

Lingguang Yang [1,2,6], Laixing Zhang [3,6], Peipei Yin[1,2,6], Hao Ding[1,6], Yu Xiao[3], Jianwei Zeng [3], Wenhe Wang[3], Huan Zhou[4], Qisheng Wang[4], Yi Zhang [1], Zeliang Chen[1,5], Maojun Yang[3] & Yue Feng [1✉]

CRISPR-Cas systems are prokaryotic adaptive immune systems and phages use anti-CRISPR proteins (Acrs) to counteract these systems. Here, we report the structures of AcrIF24 and its complex with the crRNA-guided surveillance (Csy) complex. The HTH motif of AcrIF24 can bind the Acr promoter region and repress its transcription, suggesting its role as an Aca gene in self-regulation. AcrIF24 forms a homodimer and further induces dimerization of the Csy complex. Apart from blocking the hybridization of target DNA to the crRNA, AcrIF24 also induces the binding of non-sequence-specific dsDNA to the Csy complex, similar to AcrIF9, although this binding seems to play a minor role in AcrIF24 inhibitory capacity. Further structural and biochemical studies of the Csy-AcrIF24-dsDNA complexes and of AcrIF24 mutants reveal that the HTH motif of AcrIF24 and the PAM recognition loop of the Csy complex are structural elements essential for this non-specific dsDNA binding. Moreover, AcrIF24 and AcrIF9 display distinct characteristics in inducing non-specific DNA binding. Together, our findings highlight a multifunctional Acr and suggest potential wide distribution of Acr-induced non-specific DNA binding.

[1] Beijing Advanced Innovation Center for Soft Matter Science and Engineering, Beijing Key Laboratory of Bioprocess, State Key Laboratory of Chemical Resource Engineering, College of Life Science and Technology, Beijing University of Chemical Technology, Beijing 100029, China. [2] Jiangxi Provincial Key Laboratory of Natural Active Pharmaceutical Constituents, Department of Chemistry and Bioengineering, Yichun University, Yichun 336000, China. [3] Ministry of Education Key Laboratory of Protein Science, Beijing Advanced Innovation Center for Structural Biology, Beijing Frontier Research Center for Biological Structure, School of Life Sciences, Tsinghua University, Tsinghua-Peking Center for Life Sciences, Beijing 100084, China. [4] Shanghai Synchrotron Radiation Facility, Shanghai Advanced Research Institute, Chinese Academy of Sciences, Shanghai 201204, China. [5] Key Laboratory of Livestock Infectious Diseases in Northeast China, Ministry of Education, College of Animal Science and Veterinary Medicine, Shenyang Agricultural University, Liaoning Province Shenyang 110866, China. [6] These authors contributed equally: Lingguang Yang, Laixing Zhang, Peipei Yin, Hao Ding. ✉email: fengyue@mail.buct.edu.cn

In bacteria and archaea, CRISPRs (clustered regularly inter-spaced short palindromic repeats) and associated Cas (CRISPR associated) genes provide acquired immunity against foreign mobile genetic elements (MGEs), such as bacteriophages and plasmids[1]. The CRISPR-Cas immunity is divided into three stages: adaptation, processing/biogenesis, and interference[2]. In the adaptation stage, segments of an invading genetic element are processed and integrated into the CRISPR arrays as new "spacers". During the biogenesis stage, the CRISPR array is transcribed as a long transcript (called pre-CRISPR RNA, or pre-crRNA), which is further processed into single crRNAs. In the interference stage, Cas nucleases guided by the crRNA first recognize protospacer adjacent motifs (PAMs) within the viral DNA and then cleave the DNA, which harbors complementary sequences to spacers in the crRNA. CRISPR-Cas systems are classified into class 1 and class 2, which are further divided into six types (I–VI) based upon their signature proteins[3]. Class 1 systems deploy multiple-subunit surveillance complexes, and class 2 systems depend on single-subunit Cas proteins. Among the CRISPR-Cas systems, type I system is the most ubiquitous one. Its subtype, I-F system, is composed of Cas1, Cas2/3 fusion protein, Cas8f (formerly Csy1), Cas5f (formerly Csy2), Cas7f (formerly Csy3), and Cas6f (formerly Csy4). In the type I-F system, after pre-crRNA is transcribed, Cas6f processes the pre-crRNA into small crRNAs, binds it, and further assembles with one Cas8f, one Cas5f, and six Cas7f proteins to generate the crRNA-guided surveillance complex (Csy complex).

However, to fight against the CRISPR-Cas system and survive from this adaptive immunity, phages have developed anti-CRISPR proteins (Acrs) to inactive the CRISPR-Cas system. From the first discovery of five distinct "anti-CRISPR" genes (*acrIF1-5*) in the genomes of bacteriophages infecting *Pseudomonas aeruginosa* in 2013[4], ~90 families of Acrs, which belong to 12 subtypes, have been discovered[5]. For type I-F CRISPR-Cas system, 24 families of Acrs in total have been identified[4,6–8], and the structures of 11 of them have been determined to elucidate their inhibition mechanisms. Regarding the mechanism of action, most of the type I-F Acrs were found to directly interact with the Csy complex (AcrIF1/2/4/6/7/8/9/10/14)[9–14] or Cas2/3 (AcrIF3)[15,16] to avoid recognition (AcrIF1/2/6/7/8/9/10/14) or cleavage (AcrIF3/4) of the target DNA. Among these, AcrIF9 not only sterically inhibits base-pairing between target DNA and crRNA, but also helps to tether non-sequence-specific DNA to the Csy complex[14,17]. Distinct from direct and stable interaction, we previously showed that AcrIF11 functions as an ADP-ribosyltransferase to modify N250 of the Cas8f subunit of the Csy complex, a key residue required for PAM recognition[18]. AcrIF1-5-AcrIF24 was identified in a recent study[6], and no structural or mechanistic studies have been reported for these ten Acrs. Interestingly, among these newly identified Acrs, AcrIF24 was predicted to harbor a helix-turn-helix (HTH) motif in its C-terminus, implying a possible dual Acr-Aca (Aca, Acr-associated) function, a similar role of which was previously only shown for class 2 Acrs AcrIIA1[19] and AcrIIA13–15[20].

In this study, we elucidate the mechanism of how AcrIF24 inactivates the type I-F CRISPR system. The biochemical and structural data revealed that AcrIF24 is a dimer and able to further induce the dimerization of the Csy complex. AcrIF24 binds the *acrIF23-24* promoter region in *P. aeruginosa* using its HTH motif with high affinity and acts as a transcriptional repressor. For its inhibition mechanism, one AcrIF24 molecule competes with the target DNA to bind the Csy complex through engaging five Cas7f subunits, and meanwhile promotes non-sequence-specific binding of dsDNA to the Csy complex via its HTH motif. Importantly, AcrIF24 differs from AcrIF9 in the mechanisms of inducing non-sequence-specific binding of dsDNA. In all,

our study uncovers the mechanism of inhibition of the CRISPR-Cas system by this multifunctional Acr, AcrIF24.

## Results

**AcrIF24 is a dimer and inhibits the activity of type I-F CRISPR-Cas system in vitro.** First, we investigated the inhibitory effects of AcrIF24 on the endonuclease activity of the type I-F CRISPR-Cas system. Consistent with the results of a previous in vivo assay[6], the in vitro cleavage activity of the CRISPR-Cas system was markedly inhibited in the presence of AcrIF24 (Fig. 1a and Supplementary Fig. 1). To gain structural insight into the inhibitory mechanism of AcrIF24, we first tried to solve its crystal structure. During the process of structure determination, we found that the residues ranging from 73 to 135 of AcrIF24 tend to be disordered, with electronic density too weak to model the structure of this region. Therefore, we made another AcrIF24 construct (AcrIF24 Δmiddle domain, short for AcrIF24ΔMD hereafter) by deleting this region and linking the remaining two parts of the protein with a "GSGSSG" linker. Interestingly, AcrIF24ΔMD did not show inhibitory effects in the in vitro assay (Fig. 1a and Supplementary Fig. 1), suggesting the important role of MD of AcrIF24 in inhibiting the type I-F CRISPR-Cas system. We solved its crystal structure to a resolution of 2.10 Å (Fig. 1b and Supplementary Table 1). There is only one AcrIF24ΔMD molecule in the asymmetric unit. However, gel filtration coupled with static light scattering (SLS) analyses indicated that both the full-length and the ΔMD variant of AcrIF24 exist as dimers (Fig. 1c), consistent with the dimerization forms of canonical HTH-containing proteins[21]. In the crystal, two AcrIF24 protomers with twofold symmetry interact with each other with several hydrophobic interactions from their CTDs (Fig. 1d and Supplementary Fig. 2a, b), extending to a length of ~113 Å. To validate the roles of these interactions, we introduced mutations of the involved residues. The results showed that mutating F212/H216/Y217 to either alanines or glutamates changed AcrIF24ΔMD to monomers as indicated by the assays of gel filtration chromatography (Supplementary Fig. 2c). Importantly, the circular dichroism (CD) spectrum indicated that these mutants fold normally as wild-type AcrIF24 (Supplementary Fig. 2d).

The structure of AcrIF24ΔMD revealed a two-domain architecture, with an extended N-terminal domain (residues 1–72, NTD) and a globular C-terminal domain (residues 136–228, CTD) (Fig. 1b). The NTD comprises a five-stranded antiparallel β sheet and a bent α helix. A Dali search with this domain revealed that the closest structural homolog is an auxiliary protein of soluble methane monooxygenase hydroxylase (MMOH), MMOD (Supplementary Fig. 2e, f)[22,23]. Interestingly, MMOD is an inhibitor of MMOH by associating with the canyon region of MMOH and inducing conformational changes in it. The CTD comprises five α helices and a single $3_{10}$ helix (Fig. 1b). As predicted from the previous study[6], a Dali search with the CTD returned mainly HTH-containing proteins as structural homologs (Fig. 1e and Supplementary Fig. 2g). In the structure, the two helices ranging from residue 181–198 of AcrIF24[CTD] (Fig. 1b, colored orange), almost perpendicular to each other, constitute an HTH motif, linked by the characteristic sharp turn (residues 189–191).

**AcrIF24[CTD] functions as an Aca protein of the *acrIF23-24* locus.** Since no known Aca genes form part of the *acrIF23-24* locus in *P. aeruginosa*, AcrIF24 was proposed to have a dual Acr-Aca function with the HTH motif in its CTD[6]. A hallmark of Aca proteins is the presence of an HTH motif, which is necessary for the transcriptional repression of an *acr* operon[24,25]. HTH-containing proteins normally bind to inverted repeat sequences[21].

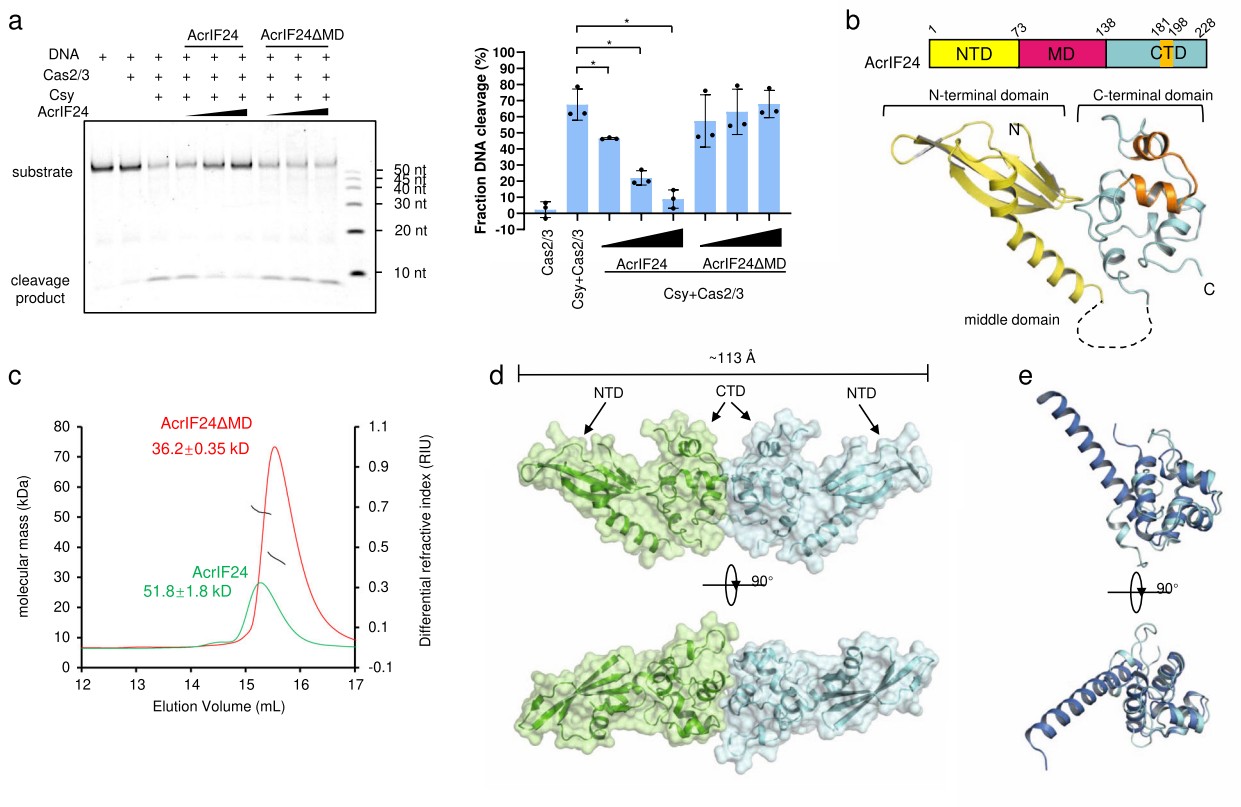

**Fig. 1 The crystal structure of AcrIF24, a dimer that inhibits the type I-F CRISPR-Cas system. a** AcrIF24 but not AcrIF24ΔMD inhibits the in vitro cleavage activity of the type I-F CRISPR-Cas system. Reactions were performed with 0.32 μM Csy complex, 0.16 μM Cas2/3, and 0.04 μM 54-bp dsDNA (5′-FAM in the non-target DNA strand, NTS). AcrIF24 or AcrIF24ΔMD was added with concentrations of 0.16, 0.32, and 0.64 μM following the order indicated by the black triangles. Fraction DNA cleavage was calculated as mentioned in the Methods section. Data are presented as mean values ± SD; $n = 3$. Two-sided $t$ test was performed. The "*" indicates that there is a significant difference between the quantitative data ($P < 0.05$). *$P = 0.0199$, 0.0018, 0.0008 (left to right). **b** The crystal structure of AcrIF24 is shown in the cartoon model, together with its schematic illustration. The NTD and CTD are colored in yellow and cyan, respectively. The HTH motif is colored in orange. The unmodelled MD is indicated by the dashed line and colored hot pink in the schematic illustration. **c** Static light scattering (SLS) studies of AcrIF24 and AcrIF24ΔMD. The calculated molecular weights of the main peaks of the two profiles are shown above the peaks. **d** The AcrIF24ΔMD dimer is shown in the cartoon and surface model. Two perpendicular views are shown. **e** Structural superimposition between AcrIF24$^{CTD}$ and the clp gene regulator ClgR from *Corynebacterium glutamicum* (PDB: 3F51). AcrIF24$^{CTD}$ and ClgR are colored in cyan and marine, respectively. Two perpendicular views are shown.

Therefore, we searched the *acrIF23-24* promoter region and identified such a site, which we refer as inverted repeat 23-24 (IR23-24). IR23-24 lies between the -35 and -10 regions (Fig. 2a). To determine whether AcrIF24 could bind the *acr* promoter region, we mixed AcrIF24 or AcrIF24ΔMD with a 23-bp dsDNA fragment containing IR23-24, and performed an electrophoretic mobility shift assay (EMSA). The results showed that both AcrIF24 and AcrIF24ΔMD bind the Acr promoter with high affinity ($K_D = 20.18$ nM and 49.06 nM, respectively, Fig. 2b and Supplementary Fig. 3). However, mutagenesis of the nucleotides within the inverted repeats, while maintaining its palindromic feature, severely decreased the binding of AcrIF24 (Fig. 2c). Superposition of AcrIF24$^{CTD}$ with the HTH domain of a controller protein C. Csp231I in a complex with DNA[26] revealed that the helix ranging from A192 to T199, with strong electropositivity, should interact with the major groove of the dsDNA (Fig. 2d). Consistently, mutagenesis experiments revealed that N194, R196, and K197 play important roles in DNA binding, as mutations of each residue severely decreased or abolished its binding to IR23-24 (Fig. 2e). The above results suggested that AcrIF24$^{CTD}$ might function as an Aca of the *acrIF23-24* locus. To investigate this, we performed an in vivo transcriptional assay using β-galactosidase fusion system. The results showed that the

IR23-24 region is indeed the Acr promoter and that AcrIF24 but not its HTH motif mutant represses transcription from this promoter (Fig. 2f). Importantly, the F212A/H216A/Y217A and F212E/H216E/Y217E mutations of AcrIF24ΔMD, which disrupt its dimerization (Supplementary Fig. 2c), also abolished the DNA binding activity of AcrIF24ΔMD, again suggesting the importance of dimerization of HTH-containing proteins in their DNA binding functions (Fig. 2g). Similar to some HTH-containing proteins, AcrIF24 also maintains a weak binding affinity to DNA molecules without inverted repeat sequences, such as the dsDNA molecules used in the in vitro cleavage assay of this study (Fig. 2b and Supplementary Fig. 2h). Therefore, to avoid the impact of non-specific binding of target dsDNA by AcrIF24, all the in vitro cleavage assays in this study were performed under an AcrIF24 concentration that does not cause its non-specific binding with target dsDNA (Supplementary Fig. 2i). Taken together, AcrIF24$^{CTD}$ functions as an Aca with transcriptional repression of the *acr* operon, as reported for AcrIIA1[19] and AcrIIA13–15[20].

**AcrIF24 forms a heterotetrameric complex with the Csy complex.** To further investigate the inhibitory mechanism of AcrIF24, we performed size-exclusion chromatography (SEC) assay to test

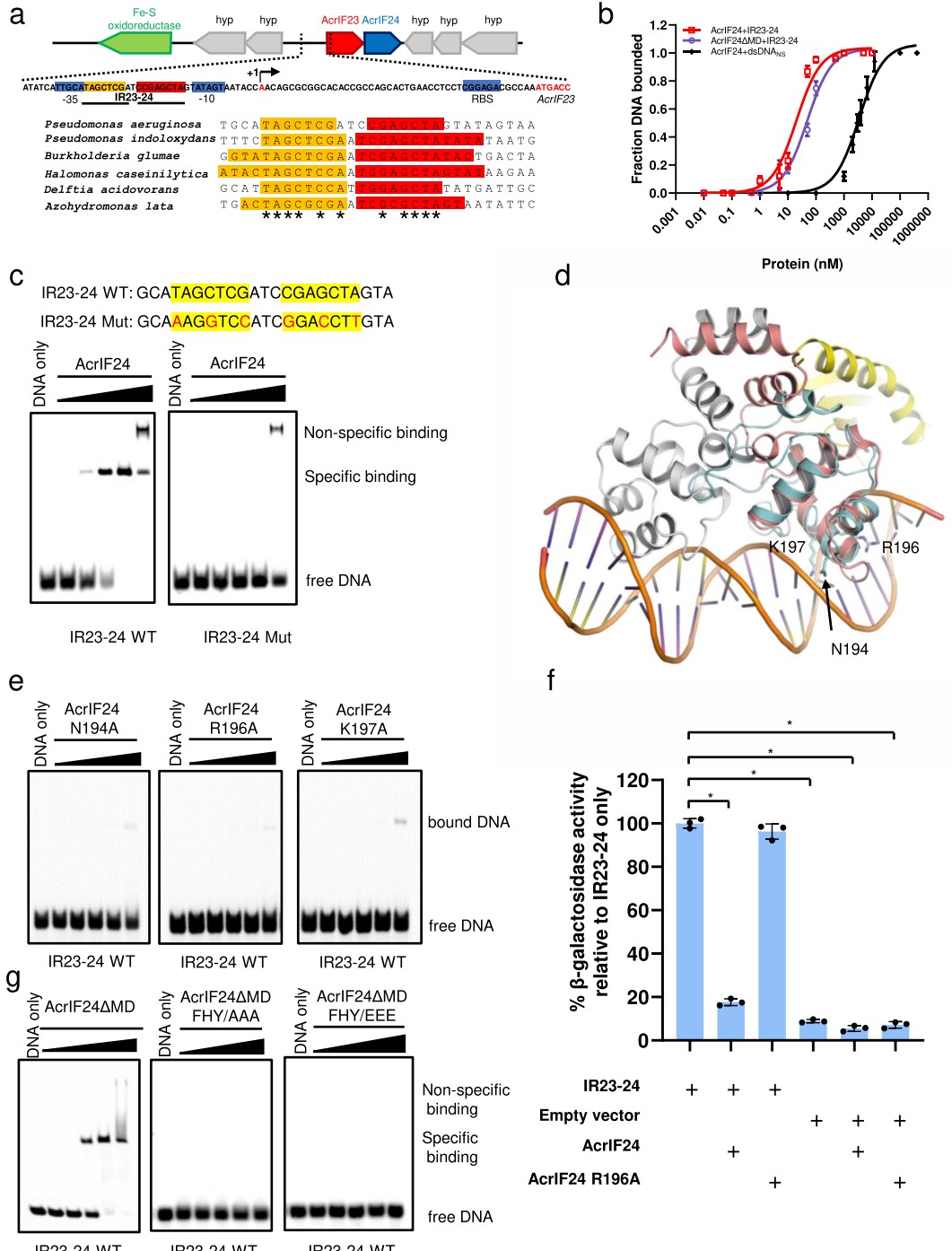

whether AcrIF24 interacts with the Csy complex or Cas2/3 from *P. aeruginosa*. The results showed that AcrIF24 co-elutes with the Csy complex but not the Cas2/3 nuclease (Supplementary Fig. 4a). Surface plasmon resonance (SPR) studies further showed that AcrIF24 exhibits strong binding to the Csy complex, with a binding $K_D$ of 4.21 nM, and $k_{on}$ and $k_{off}$ of $2.81 \times 10^5 M^{-1} s^{-1}$ and $1.18 \times 10^{-3} s^{-1}$, respectively (Fig. 3a). Interestingly, further gel filtration coupled with SLS analyses revealed that the Csy-AcrIF24 complex exists in two forms with different binding stoichiometries under different incubation ratios. When incubated with a ratio of AcrIF24: Csy at 5:1, the calculated molecular weight of the predominant form of their complex corresponds to an AcrIF24 homodimer and a single copy of the Csy complex, while that of the minor form of the complex corresponds to two

copies of Csy binding to an AcrIF24 dimer (Fig. 3b and Supplementary Table 2). However, when decreasing the ratio to 1:1 (AcrIF24: Csy), the predominant form turns out to be the heterotetramer (two copies of Csy binding to an AcrIF24 dimer). Further decreasing the incubation ratio to 1:5 (AcrIF24: Csy) then produces the heterotetramer as the only form of their complex, which is also the only form of their complex when AcrIF24 and the Csy complex were co-expressed (Fig. 3b, Supplementary Fig. 4b, and Supplementary Table 2). To further analyze the binding between Csy and AcrIF24, we used single-particle cryo-EM to determine the structure of the co-expressed Csy-AcrIF24 heterotetrameric complex at 2.88 Å resolution (Supplementary Table 3, Fig. 3c, d, and Supplementary Fig. 5). Consistent with the SLS result, the majority of the particles contain two copies of Csy

**Fig. 2 The HTH motif of AcrIF24 binds the Acr promoter with high affinity. a** Inverted repeat sequence are conserved in the upstream promoters of AcrIF24 homologs. Genomic context of the *acrIF23–acrIF24* locus of *Pseudomonas aeruginosa* prophage with inverted repeat sequence (shades of orange and red for the two half-sites of the respective repeat). Predicted regulatory sequences are shaded in blue and the predicted transcription start site (+1) is indicated by an arrow. The lower panel shows the alignment of *acrIF23–acrIF24* operon promoter from *P. aeruginosa* prophage with those of other species in which inverted repeats are displayed as above. Invariant residues are indicated by an asterisk. **b** EMSA shows that AcrIF24 and AcrIF24ΔMD both exhibit high affinity to the IR23-24 DNA. The $K_D$ values are 20.18 and 49.06 nM for AcrIF24 and AcrIF24ΔMD, respectively. AcrIF24 displays a weak binding to non-sequence-specific dsDNA (dsDNA$_{NS}$), with a $K_D$ value of 3.461 μM. Data are presented as mean values ± SD; $n = 3$. Raw data for these curves are shown in Supplementary Fig. 3. **c** EMSA used to test the binding affinities of AcrIF24 for IR23-24 and mutated IR23-24. The sequences of WT and mutated IR23-24 are shown above the gel. Reactions were performed with 1 μM WT or mutated 23-bp IR23-24 and AcrIF24 concentrations of 0.125, 0.5, 2, 8, and 32 μM following the order indicated by the black triangle. **d** Structural superimposition between AcrIF24 and a restriction-modification controller protein C.Csp231I complexed with DNA (PDB: 4JQD). The two protomers of C.Csp231I are colored in pink and gray, respectively. The NTD and CTD of AcrIF24 are colored in yellow and cyan, respectively. N194, R196, and K197 of AcrIF24 are shown in sticks. **e** EMSA was used to test the IR23-24 binding activity of AcrIF24 mutants. Reactions are performed as in **c** with IR23-24 WT. Please see the left panel of **c** for comparison with WT control. **f** The promoter region containing IR23-24 sequence from *P. aeruginosa* prophage was cloned upstream of a promoterless lacZ gene. β-galactosidase activity was measured in *P. aeruginosa* PA14 strain. The β-galactosidase activity relative to IR23-24 is shown as mean ± SD; $n = 3$. Two-sided *t* test was performed. The "*" indicates that there is a significant difference between the quantitative data ($P < 0.05$). *$P = 6.937e-7$, $2.599e-7$, $3.447e-7$, $4.101e-7$ (left to right). **g** EMSA was used to test the IR23-24 binding activity of AcrIF24ΔMD and its mutants. Reactions were performed as in **e**. FHY/AAA and FHY/EEE represent F212A/H216A/Y217A and F212E/H216E/Y217E, respectively.

complexes binding to an AcrIF24 dimer (Fig. 3c, d and Supplementary Fig. 4b). The two copies of the AcrIF24-Csy complex were nearly identical with a root mean square deviation (RMSD) value of 0.209 Å with 2605 aligned Cα atoms. The density map of the MD of AcrIF24, which is disordered in the crystallographic data, was clear enough for de novo atomic modeling (Fig. 3e, f). The MD domain of AcrIF24 folds into a mixed α/β structure, with a three-stranded β sheet followed by a helix (Fig. 3f). A Dali search with the MD did not return entries with Z scores of higher than 3.5, suggesting that this domain represents a novel fold. The two AcrIF24 protomers also align well with an RMSD value of 0.001 Å with 227 aligned Cα atoms (Supplementary Fig. 4c), and their NTD and CTD are also similar to those of the AcrIF24 molecule in crystal, with RMSD values of 0.469 Å with 140 aligned Cα atoms for both protomers compared with that in crystal (Supplementary Fig. 4d).

**Detailed interactions between AcrIF24 and the Csy complex.** In the structure of the Csy-AcrIF24 complex, each AcrIF24 interacts with the Cas7f subunits of one Csy complex. Previous to our study, AcrIF1/8/9/14 are four majorly Cas7f-interacting AcrIF proteins, in which AcrIF1/9/14 each uses two copies of protein to interact with Cas7.4f and 7.6f, respectively, and AcrIF8 mainly interacts with Cas7.5f and Cas7.6 f (Supplementary Figs. 6a–d). Distinct from these proteins, each AcrIF24 protomer lies over almost the whole Cas7f backbone through engaging five Cas7f subunits (Cas7.2-Cas7.6f, Fig. 4a and Supplementary Fig. 6e) with a buried surface area of ~2140 Å$^2$. Moreover, the bound regions on the Csy by AcrIF24 are also different from those by other Acrs, suggesting a unique binding manner (Supplementary Figs. 6e, f). Interactions between AcrIF24 and the Csy complex mainly involve the NTD and MD of AcrIF24 and the Cas7.2-Cas7.6 f subunits of the Csy complex (Fig. 4a). The NTD and MD of AcrIF24 form interfaces with the Csy complex with a surface area of 968.2 and 1002.6 Å$^2$, respectively. The NTD of AcrIF24 mainly interacts with the Cas7.6 f and Cas7.5 f subunits of the Csy complex. In the AcrIF24-Cas7.6 f interface, extensive hydrogen bond interactions are involved, e.g., H19$_{AcrIF24}$:S73$_{Cas7.6f}$, Y20$_{AcrIF24}$:T78$_{Cas7.6f}$, H19$_{AcrIF24}$:P74$_{Cas7.6f}$, G21$_{AcrIF24}$:L76$_{Cas7.6f}$, H19$_{AcrIF24}$:L76$_{Cas7.6f}$ and S45$_{AcrIF24}$:D237$_{Cas7.6f}$ (Fig. 4b). In the AcrIF24-Cas7.5 f interface, hydrogen bonds are also formed, e.g., I39$_{AcrIF24}$:K235$_{Cas7.5f}$ and T41$_{AcrIF24}$:L233$_{Cas7.5f}$ (Fig. 4c). E36$_{AcrIF24}$ also potentially interacts with K235$_{Cas7.5f}$ through a salt bridge (Fig. 4c). On the other hand, AcrIF24$^{MD}$ also mediates extensive interactions with the Csy complex, which might explain

the difference between its density maps of the crystallographic and cryo-EM data. That is, this domain might be disordered in nature, but becomes ordered when interacting with the Csy complex. AcrIF24$^{MD}$ mainly interacts with the Cas7.4f and Cas7.2f subunits of the Csy complex (Fig. 4d, e). In the AcrIF24-Cas7.4f interface, D73$_{AcrIF24}$ forms a salt bridge with K66$_{Cas7.4f}$ (Fig. 4d). Moreover, the sidechain nitrogen atom of K58$_{Cas7.4f}$ also forms hydrogen bonds with the carbonyl groups of T144$_{AcrIF24}$, G145$_{AcrIF24}$, and L147$_{AcrIF24}$ from the CTD of AcrIF24. Furthermore, P117 and L119 of AcrIF24$^{MD}$ also form hydrophobic interactions with the thumb domain of Cas7.4f (Fig. 4d). In the AcrIF24-Cas7.2 f interface, the sidechains of D105 and R116 of AcrIF24$^{MD}$ interact with those of R314 and E338 through salt bridges, respectively. The sidechain group of R106$_{AcrIF24}$ and the amide group of I102$_{AcrIF24}$ are hydrogen-bonded to the carbonyl oxygen atoms of L6$_{Cas7.2f}$ and T8$_{Cas7.2f}$, respectively. Moreover, the sidechain atoms of D104$_{AcrIF24}$ also form hydrophobic interactions with L313$_{Cas7.2f}$ (Fig. 4e). Compared with the four Cas7f subunits, AcrIF24 forms fewer interactions with the Cas7.3 f subunit. To investigate the importance of the observed interactions in the Csy-AcrIF24 structure, we generated point mutations on AcrIF24 for biochemical studies. Interestingly, under lower concentrations of WT AcrIF24 (lane 3 and 4 in Fig. 4f), only Csy(2)-AcrIF24(2) complex forms with the excess of Csy. Under the highest concentration of AcrIF24 used (lane 5 in Fig. 4f), however, Csy(1)-AcrIF24(2) complex starts to form when Csy is completely bound. Consistent with our structural observations, mutations of the AcrIF24 residues involved in the above respective interfaces or deletion of the AcrIF24$^{MD}$ all severely decreased the binding of AcrIF24 to the Csy complex (Fig. 4f). Moreover, these mutations also markedly impaired the inhibition capacity of AcrIF24, as revealed by the in vitro dsDNA cleavage assays (Fig. 4g). Taken together, AcrIF24 engages five Cas7f subunits, and the interactions between AcrIF24 and the Cas7f subunits are essential for the inhibitory effect of AcrIF24.

**AcrIF24 induces strong non-specific DNA binding activity in the Csy complex.** A comparison between the structures of the Csy-AcrIF24 and the Csy-dsDNA suggests that AcrIF24 competes with target dsDNA to interact with Cas7f subunits (Supplementary Figs. 6g, h). Therefore, we speculated that AcrIF24 would sterically block target dsDNA binding. To test this, we performed EMSA to determine whether AcrIF24 is able to block the interactions between target dsDNA and the Csy complex (Fig. 5a). However, no inhibition of target dsDNA binding was

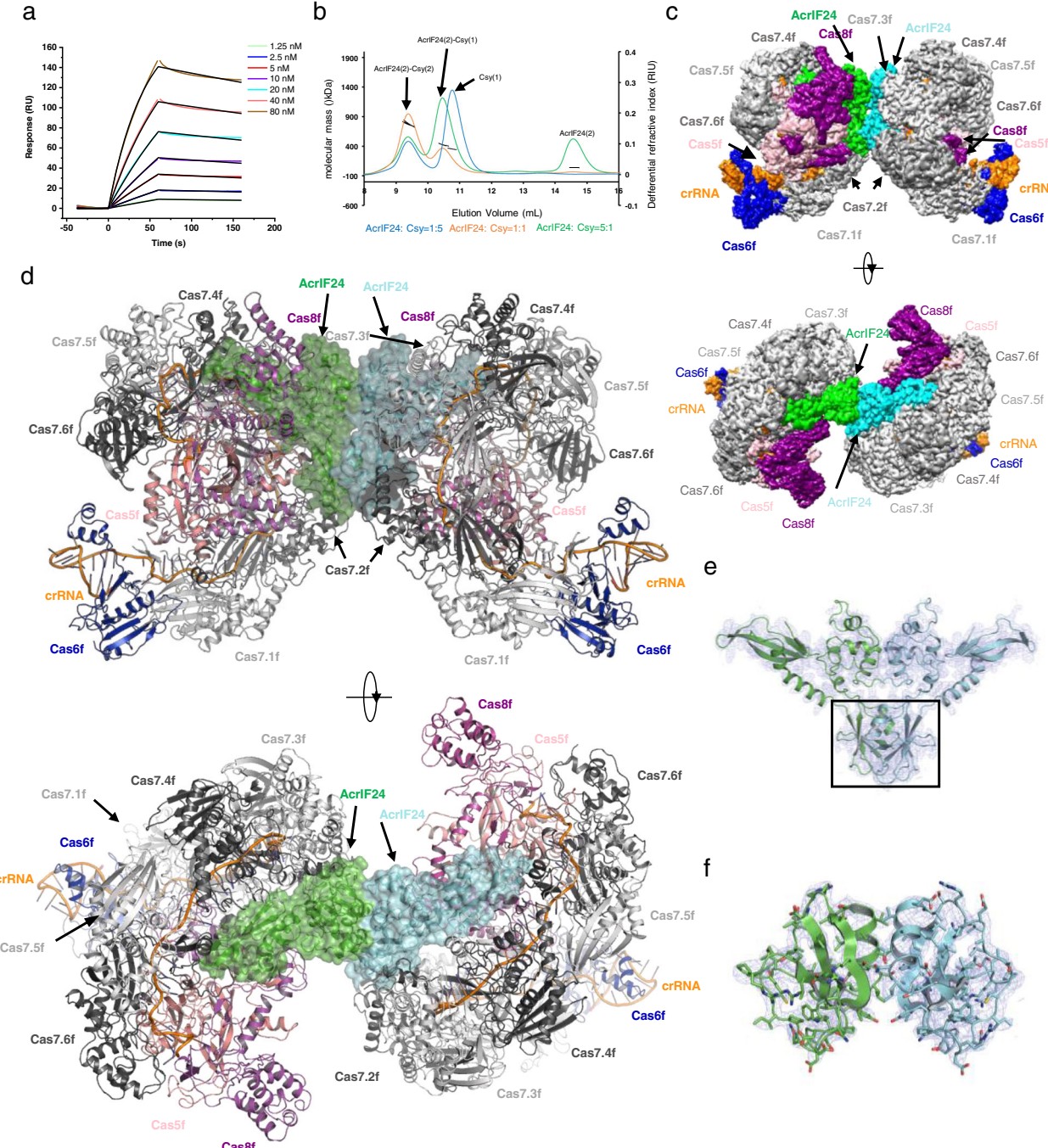

**Fig. 3 Cryo-EM structure of the Csy-AcrIF24 complex. a** Binding kinetics of AcrIF24 with the Csy complex, measured by surface plasmon resonance (SPR) method. SPR curves (colored curves) were fit kinetically using a 1:1 Langmuir binding model (black lines). The data shown are representative of three independent experiments. **b** Static light scattering (SLS) studies of the mix of AcrIF24 and the Csy complex under different molar ratios. The predicted composition of each peak based on the calculated molecular weight is shown above the peaks. The calculated molecular weights of the peaks are shown in Supplementary Table 2. **c** Cryo-EM map of the Csy-AcrIF24 complex with each subunit color-coded. Two views are shown. **d** Atomic structure of Csy-AcrIF24 in cartoon representation with each subunit colored as in **c**. Two views are shown. **e** The AcrIF24 dimer in the Csy-AcrIF24 structure. Cryo-EM density is shown in a slate mesh. The MD of AcrIF24 is marked in a box. **f** Close-up view of the MD of AcrIF24. Cryo-EM density is shown in slate mesh.

observed at any concentrations of AcrIF24 used in our assay, but a clear band with increased size compared to that of the Csy-dsDNA complex was observed at higher AcrIF24 concentrations (Fig. 5a, left half). This result suggests the formation of a ternary complex (i.e., Csy-AcrIF24-dsDNA). Because AcrIF24 competes with the target dsDNA (designated as dsDNA$_{SP}$ hereafter) to bind the Csy complex based on the structure, we speculated that dsDNA may not rely on hybridization to the crRNA guide within

the ternary complex. This is reminiscent of AcrIF9, which also tethers non-sequence-specific dsDNA to the Csy complex[14,17]. To test this hypothesis, we performed a similar EMSA using a non-specific dsDNA, short for dsDNA$_{NS}$, which possesses the same length and base composition as the specific target dsDNA, but randomized sequence and no PAM at the corresponding site[14,17]. As expected, while the Csy complex itself exhibited no affinity for dsDNA$_{NS}$, a supershifted band with the same position as the

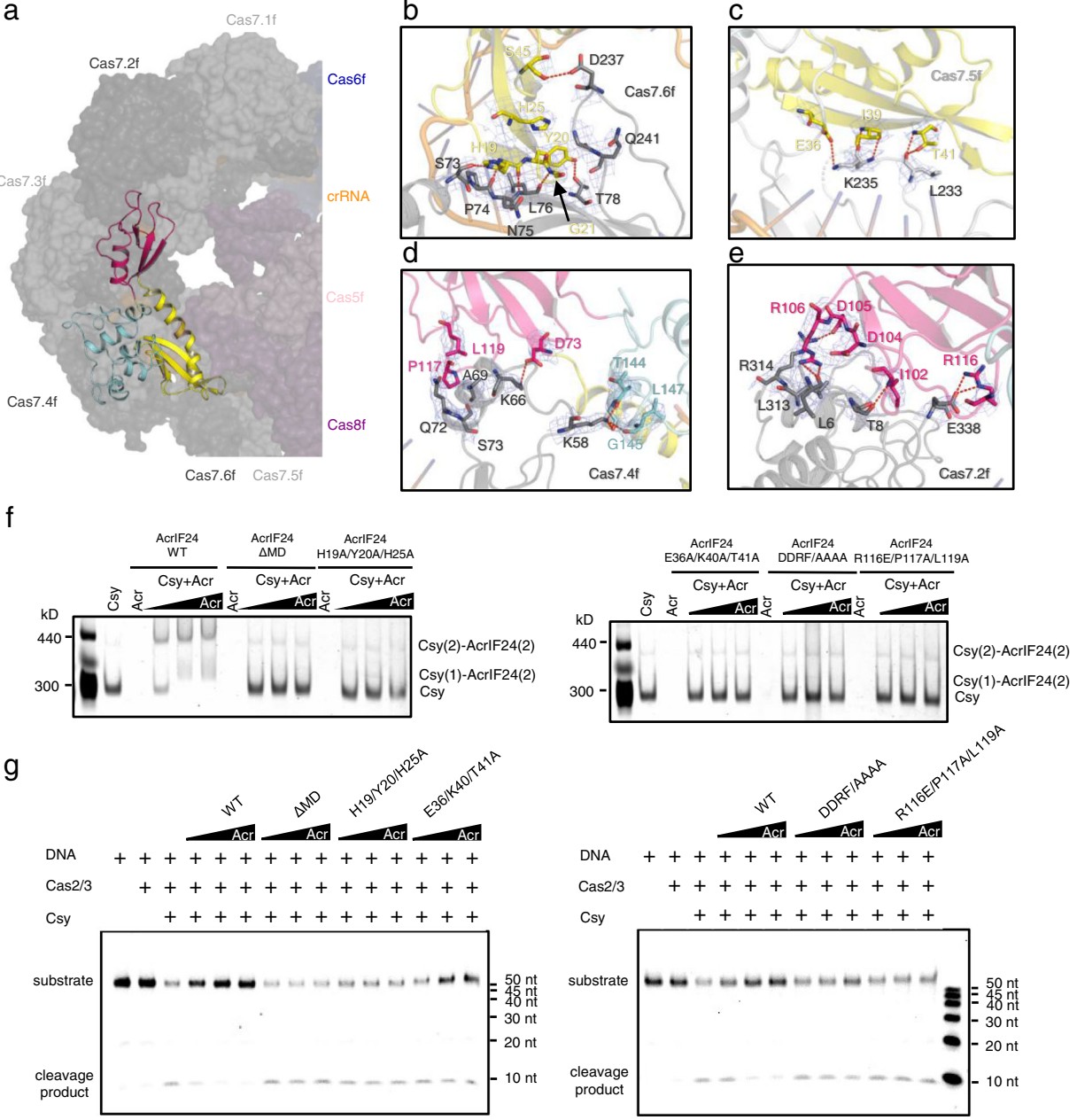

**Fig. 4 Detailed interactions between AcrIF24 and the Csy complex, which is essential to the inhibition capacity. a** Overall structure of the Csy-AcrIF24 complex with one AcrIF24 bound to one Csy complex. The Csy complex is shown in the surface model with each subunit colored as in Fig. 3d. AcrIF24 is shown in cartoon model colored in yellow, hot pink, and cyan for its NTD, MD, and CTD, respectively. **b–e** Close-up view of the interfaces between AcrIF24 and the Csy complex. The interfaces of AcrIF24NTD-Cas7.6f (**b**), AcrIF24NTD-Cas7.5f (**c**), AcrIF24MD-Cas7.4f (**d**), and AcrIF24MD-Cas7.2f (**e**) are shown. AcrIF24 is colored as in **a,** and interacting residues are shown as sticks with cryo-EM density map shown in mesh. **f** Native gel was used to test the binding between the Csy complex and AcrIF24 or its mutants. Reactions were performed with 0.32 μM Csy complex and AcrIF24 concentrations of 0.16, 0.32, and 0.64 μM following the order indicated by the black triangle. DDRF/AAAA represents the D104A/D105A/R106A/F107A mutant. The gel was stained with Coomassie blue staining. **g** Mutations of the interface residues of AcrIF24 decreased its inhibition capacity of the in vitro cleavage activity of the type I-F CRISPR system. Reactions are performed as in Fig. 1a. DDRF/AAAA represents the D104A/D105A/R106A/F107A mutant.

proposed ternary complex was also observed as the concentration of AcrIF24 increased (Fig. 5a, right half). Notably, this migration behavior specifically requires the Csy complex, as AcrIF24 alone binds both dsDNA molecules weakly even at the highest concentration used in the above assay (Fig. 5a, lanes 7 of both gels). Moreover, the AcrIF24 mutations which impair the binding between AcrIF24 and the Csy complex also decreased the supershifted bands in the EMSA assays with dsDNASP/dsDNANS (Fig. 5b). That is, AcrIF24 induces strong non-specific DNA

binding activity in the Csy complex, which is dependent on both AcrIF24 and Csy, and their interactions. To further investigate this activity of AcrIF24, we purified Csy with AcrIF24 as a complex by co-expression. Titration experiments with both the apo Csy and Csy-AcrIF24 complex indicated that the Csy-AcrIF24 complex binds dsDNASP with a comparable affinity as the apo Csy (Fig. 5c). Moreover, the Csy-AcrIF24 complex could bind dsDNANS with a similar affinity as dsDNASP (Fig. 5d). To investigate whether dsDNASP and dsDNANS bind at similar sites

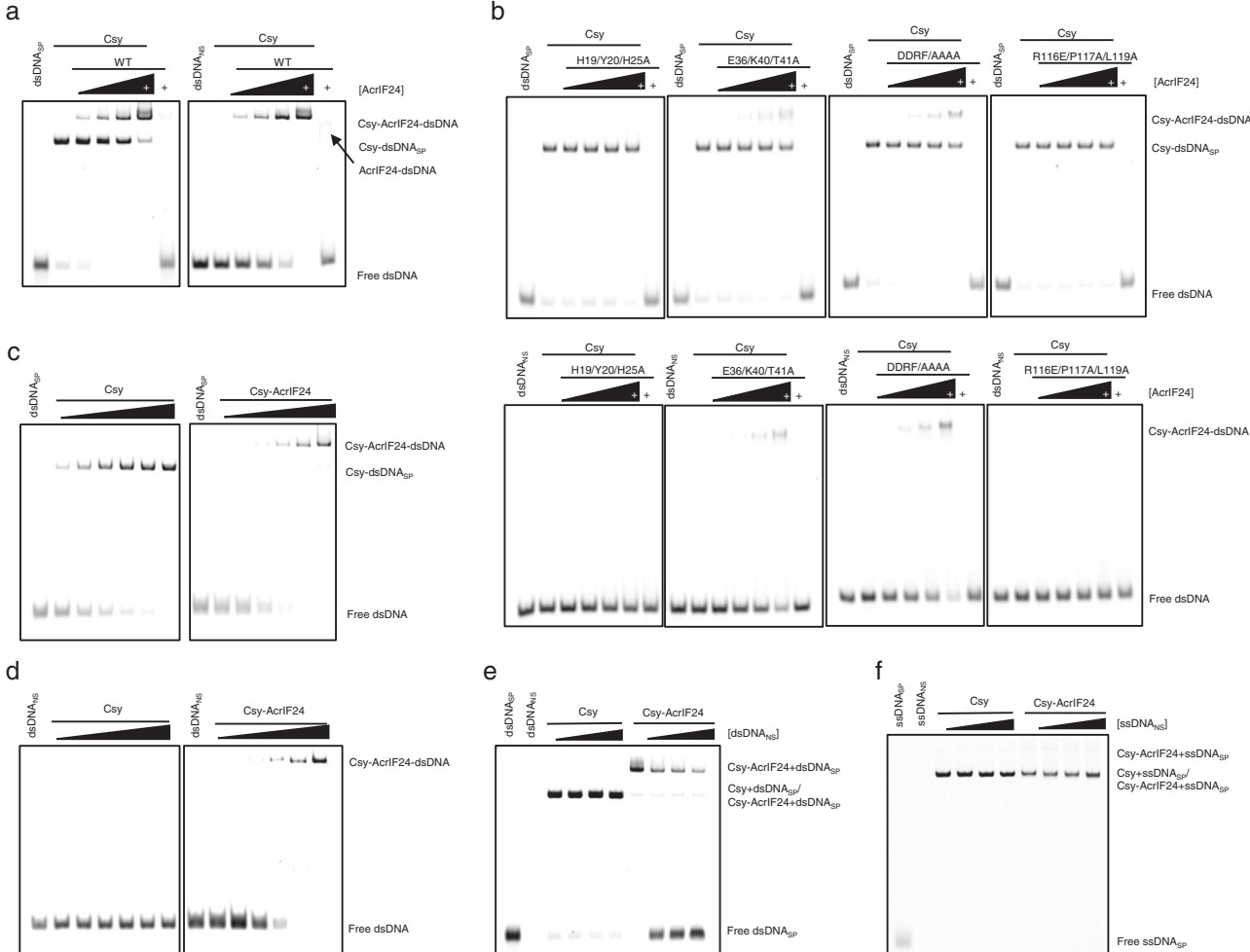

**Fig. 5 AcrIF24 induces strong non-specific DNA binding activity in the Csy complex. a** EMSA was used to test the induction of non-specific DNA binding by AcrIF24 using $dsDNA_{SP}$ and $dsDNA_{NS}$. Reactions were performed with 1 μM $dsDNA_{SP}$ or $dsDNA_{NS}$, 8 μM Csy, and concentrations of AcrIF24 were set as 1, 2, 4, and 8 μM following the order indicated by the black triangle. The "+" represents 8 μM of AcrIF24. **b** EMSA was used to test the induction of non-specific DNA binding by AcrIF24 mutants using $dsDNA_{SP}$ and $dsDNA_{NS}$. Reactions were performed as in **a**. DDRF/AAAA represents the D104A/D105A/R106A/F107A mutant. Please see the left and right panels of **a** for WT controls of the experiments using $dsDNA_{SP}$ and $dsDNA_{NS}$, respectively. **c–d** EMSA was used to test the DNA binding of Csy complex and Csy-AcrIF24 using $dsDNA_{SP}$ (**c**) and $dsDNA_{NS}$ (**d**). $dsDNA_{SP}$ and $dsDNA_{NS}$ were added as 1 μM, and the concentrations of Csy complex and Csy-AcrIF24 were set as 0.5, 1, 2, 4, 8, and 16 μM following the order indicated by the black triangle. **e** EMSA was used to test the binding competition between $dsDNA_{SP}$ and $dsDNA_{NS}$ on Csy or Csy-AcrIF24 complex. Reactions were started with incubation of 1 μM $dsDNA_{SP}$ and 10 μM Csy or Csy-AcrIF24 complex, and $dsDNA_{NS}$ without FAM label was then added with concentrations of 5, 10, and 20 μM following the order indicated by the black triangle. **f** EMSA was used to test the binding competition between $ssDNA_{SP}$ and $ssDNA_{NS}$ on Csy or Csy-AcrIF24 complex. Reactions were exactly the same as in **e** except that all DNA molecules involved were single-stranded.

within the observed ternary complex, we performed a competition EMSA with both the Csy and the preformed Csy-AcrIF24 complex. While the addition of $dsDNANS$ even at a 20-fold concentration of $dsDNA_{SP}$ caused no reduction in binding of $dsDNA_{SP}$ in the Csy complex (Fig. 5e, lanes 3–6), addition of $dsDNANS$ to the Csy-AcrIF24 complex led to markedly increased levels of free $dsDNASP$ (Fig. 5e, lanes 7–10). This suggests that 54-bp $dsDNA_{SP}$ and $dsDNA_{NS}$ bind at overlapping sites on the Csy-AcrIF24 complex, also reminiscent of AcrIF9. The Csy-AcrIF24 complex also interacts with $ssDNA_{SP}$, but the competition EMSA showed that $ssDNA_{NS}$ cannot compete $ssDNA_{SP}$ off the Csy-AcrIF24 complex (Fig. 5f), suggesting that the 54-nt target and non-specific ssDNA do not bind at the same site in the Csy-AcrIF24 complex, and Csy-AcrIF24 may bind $ssDNA_{SP}$ specifically, as Csy-AcrIF9 does[17]. Then we moved on to test the binding affinity of Csy-AcrIF24 and Csy-AcrIF9 for the above

four types of DNA molecules. The MST (microscale thermophoresis) assay suggested that the binding mode of the four types of DNA molecules are similar for Csy-AcrIF24 and Csy-AcrIF9 (Supplementary Fig. 7a, b). While each Csy-Acr complex displays comparable binding affinities for $dsDNA_{SP}$ and $dsDNA_{NS}$, it displays the highest binding affinity for $ssDNA_{SP}$, approximately 10-fold higher than that of $ssDNA_{NS}$. This is consistent with the results of the competition experiments using $dsDNA_{SP}$-$dsDNA_{NS}$ pair (Fig. 5e) and $ssDNA_{SP}$-$ssDNA_{NS}$ pair (Fig. 5f) for Csy-AcrIF24 and the corresponding experiments for Csy-AcrIF9 complex[17], again suggesting that both Csy-AcrIF24 and Csy-AcrIF9 complexes bind specifically to $ssDNA_{SP}$. Notably, the Csy-AcrIF9 complex displays a slightly higher binding affinity than Csy-AcrIF24 for each of the four types of DNA molecules. Taken together, AcrIF24 induces strong non-specific DNA binding activity in the Csy complex, reminiscent of AcrIF9.

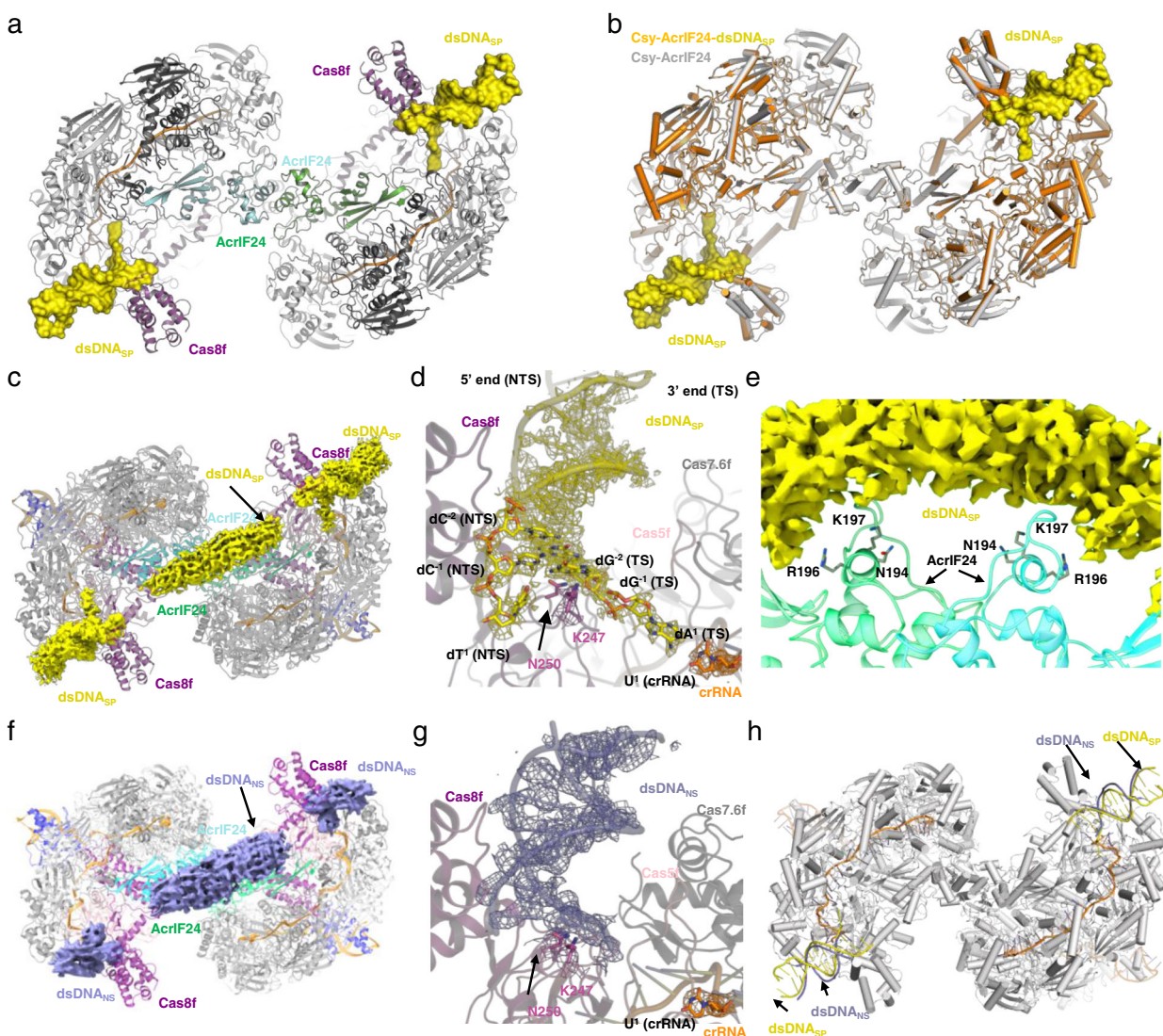

**Fig. 6 Cryo-EM structure of Csy-AcrIF24-dsDNA complexes. a** Atomic structure of Csy-AcrIF24-dsDNA$_{SP}$ in cartoon representation with each subunit colored as in Fig. 3d. The bound dsDNA$_{SP}$ molecules are shown in surface model colored in yellow. **b** Structural superimposition between Csy-AcrIF24-dsDNA$_{SP}$ and Csy-AcrIF24. The structures of Csy-AcrIF24-dsDNA$_{SP}$ and Csy-AcrIF24 are colored in orange and gray, respectively. The dsDNA$_{SP}$ is shown in the surface model colored in yellow. **c** The densities corresponding to dsDNA molecules in the Csy-AcrIF24-dsDNA$_{SP}$ structure are shown in yellow mesh. The Csy and AcrIF24 are colored as in Fig. 3d. **d** Close-up view of the binding site of dsDNA$_{SP}$ near the PAM recognition loop in Cas8f in the Csy-AcrIF24-dsDNA$_{SP}$ complex. The cryo-EM density maps for dsDNA$_{SP}$, K247 of Cas8f and U1 of the crRNA are shown in mesh. **e** Close-up view of the interfaces between AcrIF24 and the dsDNA$_{SP}$ density within the Csy-AcrIF24-dsDNA$_{SP}$ complex. The density corresponding to dsDNA is shown as yellow mesh and N194/R196/K197 is shown in sticks. **f** The densities corresponding to dsDNA molecules in the Csy-AcrIF24-dsDNA$_{NS}$ structure are shown in the slate mesh. The Csy and AcrIF24 are colored as in Fig. 3d. **g** Close-up view of the binding site of dsDNA$_{NS}$ near the PAM recognition loop in Cas8f in the Csy-AcrIF24-dsDNA$_{NS}$ complex. The cryo-EM density maps for dsDNA$_{NS}$, K247, and N250 of Cas8f and U1 of the crRNA are shown in the mesh. The base groups of dsDNA$_{NS}$ are shown in cartoon, although the sequence of dsDNA$_{NS}$ cannot be determined. **h** Structural superimposition between Csy-AcrIF24-dsDNA$_{SP}$ and Csy-AcrIF24-dsDNA$_{NS}$. Only dsDNA$_{NS}$ (colored slate) molecules are shown in Csy-AcrIF24-dsDNA$_{NS}$ structure. The dsDNA$_{SP}$ is colored in yellow.

**Structural overview of the Csy-AcrIF24-dsDNA$_{SP}$ complex**. To determine how the Csy-AcrIF24 complex recruits dsDNA$_{SP}$, we purified the Csy-AcrIF24-dsDNA$_{SP}$ complex by incubating the purified co-expressed Csy-AcrIF24 complex with dsDNA$_{SP}$ and solved its structure at a resolution of 3.02 Å (Fig. 6a, Supplementary Fig. 8, and Supplementary Table 3). The overall structure of the Csy-AcrIF24-dsDNA$_{SP}$ complex aligns well with that of Csy-AcrIF24 with an RMSD of 0.458 Å with 5313 aligned Cα atoms (Fig. 6b). The reconstruction of the Csy-AcrIF24-dsDNA$_{SP}$ complex reveals three discontinuous stretches of density that are not accounted for Csy or AcrIF24: two helical densities each

bound near the hook domain of Cas8f, and one vague stretch of density with a cylindrical shape bound above the AcrIF24$^{CTD}$ (Fig. 6c-e). The helical densities near the hook domain of each Cas8f allowed us to model two short segments of dsDNA containing the PAM (G-C/G-C) sequence (Fig. 6a). In the Csy-AcrIF24-dsDNA$_{SP}$ complex, while the dsDNA$_{SP}$ is bound at the similar site of Cas8f as in the previously solved structure of Csy-dsDNA complex[10,27], the density was only visible for 13-bp of dsDNA$_{SP}$ with merely one nucleotide (A$^1$ in the target strand, TS) of the protospacer (Fig. 6d). The dsDNA duplex is clearly separated at A$^1$(TS)-T$^1$(NTS) position, but no density is observed for

the following sequence of the protospacer (Fig. 6d). For the cylindrical density above the AcrIF24$^{CTD}$, however, it is difficult for us to determine the orientation of this segment of dsDNA (Fig. 6e). We speculate that the cylindrical density might be contributed by dsDNA molecules bound from opposite orientations. Taken together, dsDNA$_{SP}$ is bound both at the hook domains of Cas8f subunits and above AcrIF24$^{CTD}$ in the Csy-AcrIF24 complex.

**Structural overview of the Csy-AcrIF24-dsDNA$_{NS}$ complex**. To determine how the Csy-AcrIF24 complex recognizes dsDNA$_{NS}$, we also prepared the Csy-AcrIF24-dsDNA$_{NS}$ complex similarly and solved its structure at a resolution of 3.20 Å (Fig. 6f, Supplementary Fig. 9, and Supplementary Table 3), which also revealed three discontinuous stretches of density for DNA, two near the hook domains of the two Cas8f subunits and one over the AcrIF24 CTD. The cylindrical density above the AcrIF24 CTD in the Csy-AcrIF24-dsDNA$_{NS}$ structure is similar to that in the cryo-EM map of Csy-AcrIF24-dsDNA$_{SP}$, which makes it difficult to determine the orientation of this segment of dsDNA. Meanwhile, in the Csy-AcrIF24-dsDNA$_{NS}$ structure, the densities near the hook domains of both Cas8f subunits are also much weaker than those in the Csy-AcrIF24-dsDNA$_{SP}$ structure, making it only possible to determine the orientation but not the sequence of the DNA segments (Fig. 6g). Structural superimposition between the two complexes showed that dsDNA$_{NS}$ and dsDNA$_{SP}$ share a similar binding site near the hook domain of Cas8f (Fig. 6h). Taken together, dsDNA$_{NS}$ is bound at similar sites as dsDNA$_{SP}$ in the Csy-AcrIF24 complex.

**Structural elements involved in dsDNA binding by the Csy-AcrIF24 complex**. In the structure of Csy-AcrIF24-dsDNA$_{SP}$, K247 and N250 of the Cas8f subunit are involved in dsDNA$_{SP}$ binding (Fig. 6d), as revealed in the Csy-dsDNA structure[27]. Meanwhile, in the structure of Csy-AcrIF24-dsDNA$_{NS}$, these two residues are also adjacent to the density of dsDNA molecules. In our previous study, we showed that the K247E and N250D mutations of the Cas8f subunit both abolished dsDNA binding by the Csy complex[18]. To investigate the roles of these two residues in non-specific DNA binding by the Csy-AcrIF24 complex, we performed EMSA of both dsDNA$_{SP}$ and dsDNA$_{NS}$ using mutated Csy-AcrIF24 complex in which the Csy complex harbors the Cas8f K247E, Cas8f N250D or Cas8f K247E/N250D mutations. Surprisingly, these mutations decreased the binding of not only dsDNA$_{SP}$, but also that of dsDNA$_{NS}$ to the Csy-AcrIF24 complex (Fig. 7a). This suggests that the "K-wedge" (PAM recognition loop) of Cas8f mediates the binding of both dsDNA$_{SP}$ and dsDNA$_{NS}$ to the Csy-AcrIF24 complex. Next, we investigated which residues of AcrIF24 are involved in dsDNA binding. The density map showed that the cylindrical density corresponding to dsDNA segment is also near the HTH motif of AcrIF24, especially R196 and K197 (Fig. 6e). To test whether the HTH motif of AcrIF24 is involved in the dsDNA binding in the Csy-AcrIF24-dsDNA complex, we performed EMSA of both dsDNA$_{SP}$ and dsDNA$_{NS}$ with the Csy complex in the presence of WT AcrIF24 or its HTH motif mutants. Consistent with the observance of the density map, whereas the N194A mutant displayed the same activity in inducing non-sequence-specific DNA binding as WT AcrIF24, the R196A and K197A mutations markedly decreased this activity (Fig. 7b). Notably, none of the mutations interfered with its binding affinity for the Csy complex (Supplementary Fig. 10a). Interestingly, while all the three mutations of AcrIF24 separately decreased its binding to IR23-24 (Fig. 2e), only the R196A and K197A mutations decreased its binding to dsDNA$_{SP}$/dsDNA$_{NS}$, i.e. non-sequence-specific DNA

(Supplementary Fig. 10b). That is, the ability of inducing non-sequence-specific DNA binding in the Csy-AcrIF24 complex is closely related to the activity of non-sequence-specific DNA binding of AcrIF24 itself. This indicates that the specific DNA binding sites of AcrIF24 partially overlap with those involved in non-sequence-specific DNA binding. Taken together, the PAM recognition loop of Cas8f, and the HTH motif of AcrIF24 are the structural elements involved in both dsDNA$_{SP}$ and dsDNA$_{NS}$ binding by the Csy-AcrIF24 complex. Consistent with this, the Csy-AcrIF24 complex with both Cas8f K247E/N250D and AcrIF24 R196A or K197A mutations displayed completely no binding to dsDNA$_{SP}$/dsDNA$_{NS}$ (Fig. 7c).

**Induction of non-specific DNA binding by AcrIF24 contributes to its inhibitory capacity**. Then we moved on to investigate whether the activity of tethering non-sequence-specific dsDNA to the Csy complex is related to the inhibitory capacity of AcrIF24. We first compared the inhibitory effects of AcrIF24 and its mutants N194A, R196A, and K197A through in vitro DNA cleavage assay. The results showed that the R196A and K197A single mutants both display decreased inhibitory capacity compared to WT AcrIF24 when they are added with the same concentrations (Fig. 7d and Supplementary Fig. 11). Again, the N194A mutant exhibited similar activity as WT AcrIF24, which is consistent with its unaffected capacity to induce non-specific DNA binding in the Csy complex (Fig. 7b). To investigate the biological relevance of the non-specific DNA binding activity, we used a phage plaquing assay by transforming the type I-F CRISPR-Cas system from *P. aeruginosa* into *Escherichia coli* to target lambda (λ) phage[28]. By co-transforming AcrIF24 or its mutants in *E. coli* cells, their Acr activity were tested. The results showed that while λ phage can infect *E. coli* NovaBlue (DE3) cells transformed with empty vectors, cells expressing Csy, Cas2/3, and a synthetic CRISPR array designed to target λ phage successfully caused about 6-log reduction of plaques (Supplementary Fig. 12). However, co-transforming AcrIF24 into the cells expressing Csy, Cas2/3, and the synthetic CRISPR array rendered the cells sensitive to λ phage again, indicating the inhibition of the type I-F CRISPR-Cas by AcrIF24 (Supplementary Fig. 12). As revealed in Fig. 7e, while AcrIF24 N194A exhibits similar inhibitory activity as WT AcrIF24, both AcrIF24 R196A and AcrIF24 K197A showed about two to threefold reductions in activity. This is also consistent with their roles in inducing non-specific DNA binding and in vitro DNA cleavage assay. As mentioned above, mutations disrupting dimerization of AcrIF24 (F212/H216/Y217 mutations to A or E) also result in extremely poor solubility of the protein during expression. Therefore, we used the above in vivo phage plaquing assay to test the activity of CTD-deletion and dimerization mutants of AcrIF24. As revealed by the in vivo assay, AcrIF24ΔCTD and AcrIF24 FHY/EEE both exhibited reductions in inhibitory activity (Fig. 7e). This result also provided the biological relevance of the non-specific DNA binding activity endowed by the HTH motif of AcrIF24. Consistently, a recent study about AcrIF9 showed that mutations of the sites of AcrIF9 involved in non-sequence-specific DNA binding also decreased the inhibition activity of AcrIF9 in vivo[17]. However, it is worth noting that ~10-fold reductions in activity, as seen in the experiments of AcrIF9 point mutations, should be the borderline for mutants of a key residue in phage plaquing assays. Therefore, the results here may suggest that the non-specific DNA-binding activity of AcrIF24 does contribute to its inhibitory activity, but may play a minor role. Together, these results indicated that AcrIF24 inhibits the activity of the Csy complex at least through two approaches: AcrIF24 primarily engages the target DNA binding site in the Csy complex to block the hybridization of

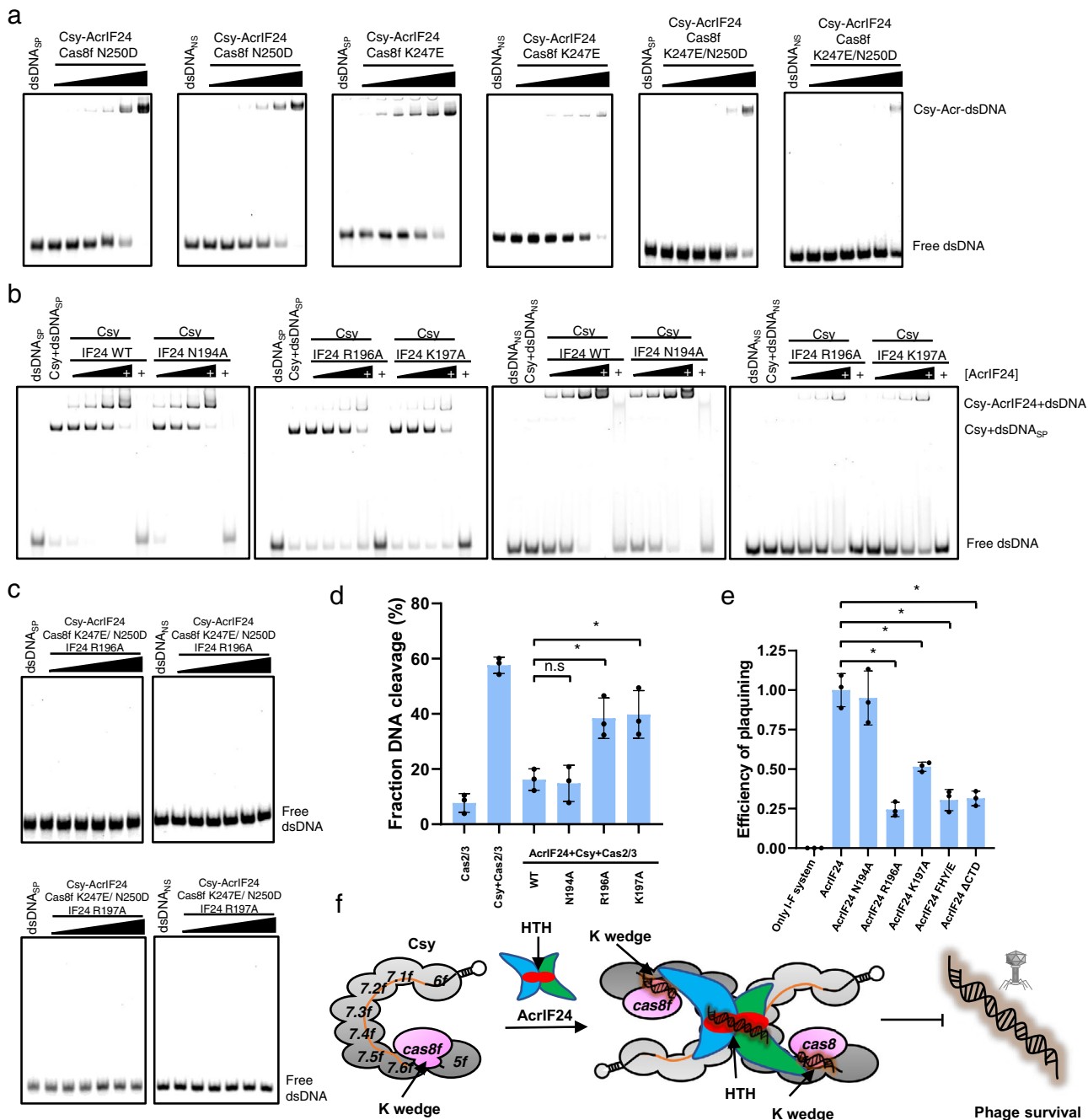

**Fig. 7 Structural elements involved in dsDNA binding by the Csy-AcrIF24 complex. a** EMSA used to test the non-specific DNA binding ability of Csy-AcrIF24 with mutations in the PAM recognition loop of Cas8f using dsDNA$_{SP}$ and dsDNA$_{NS}$. Reactions were performed as in Fig. 5c. Please see the right panels of Figs. 5c, d for WT controls of the experiments using dsDNA$_{SP}$ and dsDNA$_{NS}$, respectively. **b** EMSA used to test the induction of non-specific DNA binding by AcrIF24 or its mutants using dsDNA$_{SP}$ and dsDNA$_{NS}$. Reactions were performed as in Fig. 5a. **c** EMSA used to test the non-specific DNA binding ability of Csy-AcrIF24 with mutations both in AcrIF24 and Csy using dsDNA$_{SP}$ and dsDNA$_{NS}$. Reactions were performed as in Fig. 5c. Please see the right panels of Figs. 5c, d for WT controls of the experiments using dsDNA$_{SP}$ and dsDNA$_{NS}$, respectively. **d** Mutations impairing the induction of non-specific DNA binding of AcrIF24 decreased its capacity of inhibition in the in vitro cleavage activity assay. Reactions were performed as in Fig. 1a except that the concentrations of AcrIF24 or its mutants were set as 0.16 μM. Data are presented as mean values ± SD; $n = 3$. Two-sided $t$ test was performed. The "*" indicates that there is a significant difference between the quantitative data ($P < 0.05$). *$P = 0.0098$, 0.0125 (left to right). n.s, not significant, $P = 0.7685$. **e** Phage plaque assay verified that mutations impairing the induction of non-specific DNA binding of AcrIF24 decreased its capacity of inhibition of the type I-F system in vivo. Data are presented as mean values ± SD; $n = 3$. A two-sided $t$ test was performed. The "*" indicates that there is a significant difference between the quantitative data ($P < 0.05$). *$P = 0.0003$, 0.0015, 0.0006, 0.0005 (left to right). **f** Schematic illustration of working mechanisms of AcrIF24 on the inhibition of type I-F CRISPR-Cas system. Direct interaction with Csy and induction of its non-specific DNA binding both contribute to the inhibition by AcrIF24.

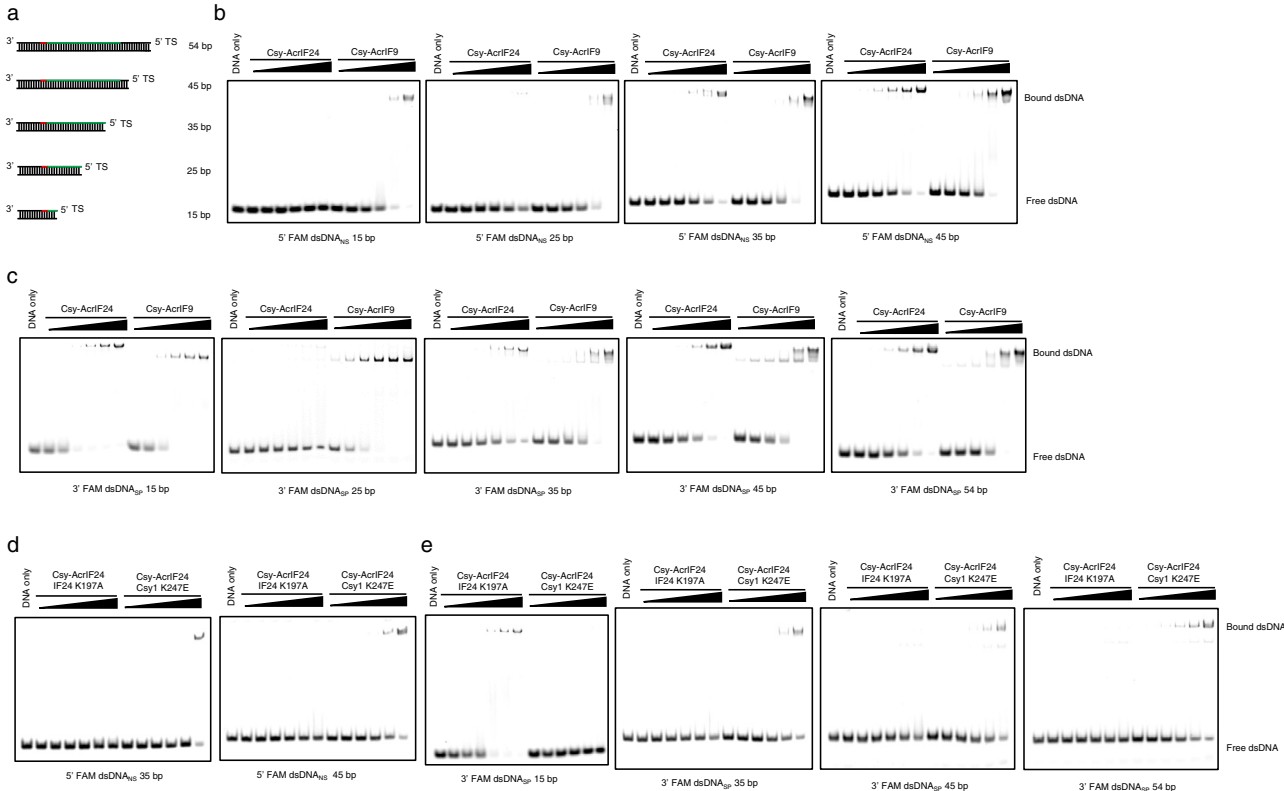

**Fig. 8 The Csy-AcrIF24 complex interacts with dsDNA_SP and dsDNA_NS through similar mechanisms. a** Schematic illustration of the dsDNA_SP with different lengths is used in the present study. Red and green lines represent PAM and protospacer sequences, respectively. **b–c** EMSA was used to test the non-specific DNA binding of Csy-AcrIF24 or Csy-AcrIF9 using dsDNA_NS with 5′ FAM labels (**b**) and dsDNA_SP with 3′ FAM labels (**c**) of different lengths. Reactions were performed as in Fig. 5c. **d–e** EMSA was used to test the non-specific DNA binding of Csy-AcrIF24 mutants using dsDNA_NS with 5′ FAM labels (**d**) and dsDNA_SP with 3′ FAM labels (**e**) of different lengths. Reactions were performed as in Fig. 5c. Please see the right two panels of **b** for WT controls of the experiments using dsDNA_NS 35 bp and 45 bp in **d**, respectively.

DNA to the crRNA. In addition, AcrIF24 may also sequester the Csy complex through associations with non-target dsDNA (Fig. 7f).

**The Csy-AcrIF24 complex interacts with dsDNA_SP and dsDNA_NS through similar mechanisms.** Next, we moved on to investigate whether the Csy-AcrIF24 complex interacts with dsDNA_SP and dsDNA_NS through similar mechanisms. Previous competition EMSA suggested that 54-bp dsDNA_NS and dsDNA_SP bind at overlapping sites on the Csy-AcrIF24 complex, and here we tried dsDNA molecules with different lengths to investigate whether there is difference between the binding mechanism of dsDNA_SP and dsDNA_NS by the Csy-AcrIF24 complex. To this end, we performed EMSA of Csy-AcrIF24 with 15-, 25-, 35-, 45-, and 54-bp dsDNA_SP and dsDNA_NS molecules (Fig. 8a). While the binding between 54-bp dsDNA_SP/dsDNA_NS and Csy-AcrIF24 was not affected by the position of the fluorescin label in the target strand (Fig. 5c, d, 8c, and Supplementary Fig. 13a), it is unknown whether this is also the case for dsDNA_SP/dsDNA_NS of other lengths. Therefore, to consider the impact caused by fluorescin labels, we tried dsDNA_NS and dsDNA_SP molecules of different lengths which are fluorescently labeled either at the 5′ or 3′-end of the target strand (Fig. 8a). For dsDNA_NS, similar results were obtained with the two labeling positions (Fig. 8b and Supplementary Fig. 13a, the results of 5′ labeling DNA are used hereafter). For dsDNA_SP, however, the Csy-AcrIF24 complex bound 3′ but not 5′-end fluorescently labeled 15-bp dsDNA_SP, while similar results were obtained for dsDNA_SP molecules of

other lengths with the two labeling positions (Fig. 8c and Supplementary Fig. 13b). This can be explained by the binding position of the visible 13-bp DNA duplex in the Csy-AcrIF24-dsDNA_SP complex, which shows the fluorescein label at the 3′ end of the target strand will cause negligible sterical hindrance (Fig. 6d). Therefore, we used the results obtained with the 3′ end fluorescently labeled dsDNA_SP molecules. Interestingly, the Csy-AcrIF24 complex displayed almost no binding to 15- and 25-bp dsDNA_NS, and gradually increased affinity to dsDNA_NS in the order of 35-, 45- and 54-bp length (Figs. 8b and 5c). For dsDNA_SP, the Csy-AcrIF24 complex displays moderate binding to dsDNA molecules of all lengths except 25-bp (Fig. 8c). To investigate which structural elements of Csy-AcrIF24 are involved in the dsDNA binding above, we performed EMSA with Csy-AcrIF24 mutants harboring either the "K-wedge" mutation (Cas8f K247E) or HTH motif mutation (AcrIF24 K197A). For dsDNA_NS, similar to the results of 54-bp dsDNA_NS (Fig. 7a, b), both mutations markedly decreased the binding to 35- and 45-bp dsDNA_NS (Fig. 8d), suggesting the involvement of both of them. For dsDNA_SP, however, the binding of 15-bp dsDNA_SP was abolished by the Cas8f K247E but not affected by the AcrIF24 K197A mutation (Fig. 8e). Consistent with this, the Csy complex itself showed comparable affinity for the same 15-bp dsDNA_SP as the Csy-AcrIF24 complex (Supplementary Fig. 13c). For the dsDNA_SP with lengths from 35- to 54-bp, similar to the results of 5′ labeled 54-bp dsDNA_SP (Fig. 7a, b), their bindings were decreased by both mutations, suggesting the involvement of both structural elements (Fig. 8e). Taken together, the above results indicated that the Csy-AcrIF24 complex interacts with dsDNA_SP

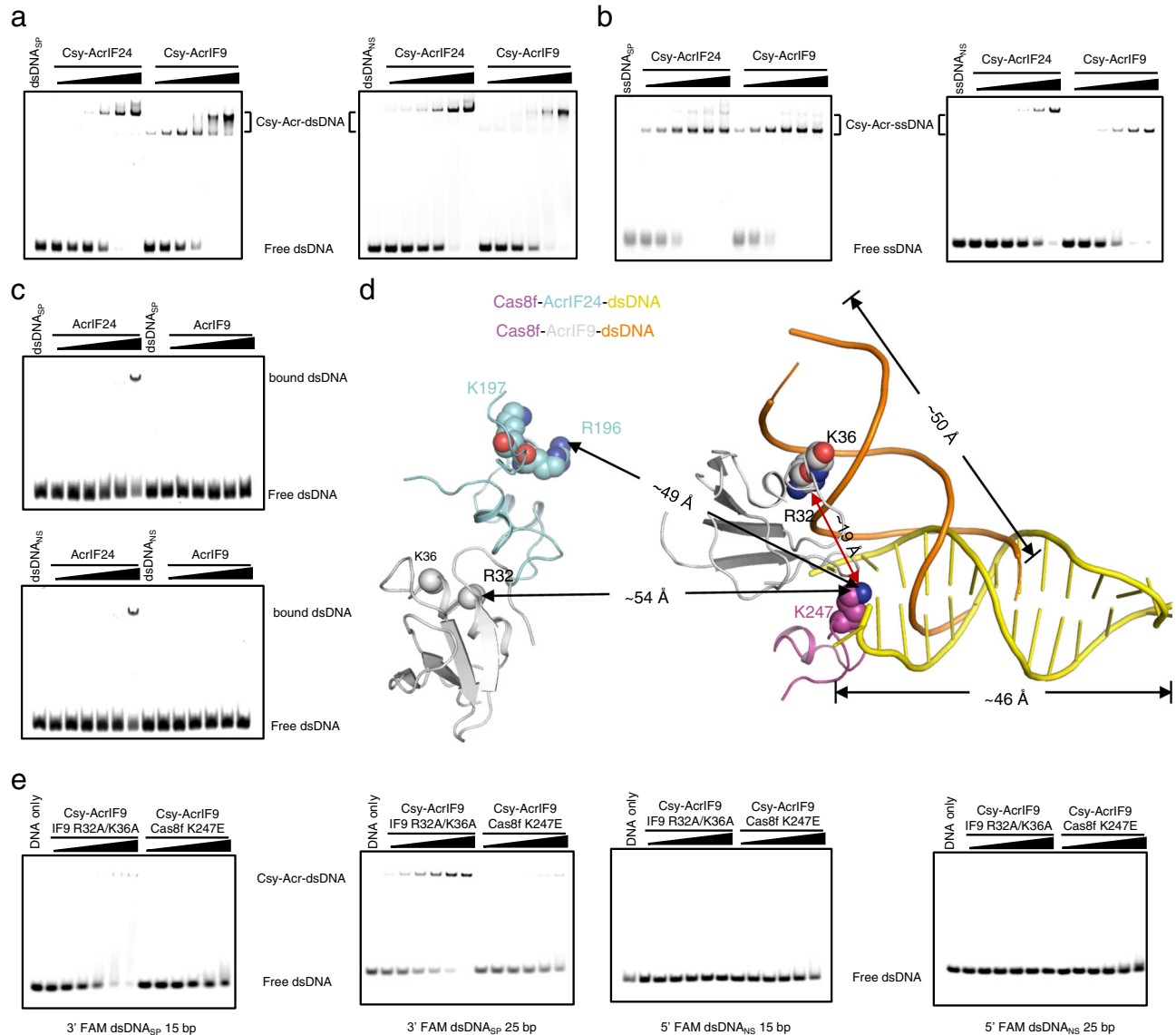

**Fig. 9 AcrIF24 and AcrIF9 show different characteristics in inducing non-specific DNA binding. a–b** EMSA used to test the non-specific DNA binding affinity and mode of Csy-AcrIF24 and Csy-AcrIF9 using dsDNA$_{SP}$/dsDNA$_{NS}$ (**a**), or ssDNA$_{SP}$/ssDNA$_{NS}$ (**b**). Reactions were performed as in Fig. 5c. **c** EMSA used to test the binding ability of AcrIF24 and AcrIF9 using dsDNA$_{SP}$ or dsDNA$_{NS}$. Reactions were performed as in Fig. 5c. **d** Structural superimposition of the Csy-AcrIF24-dsDNA$_{SP}$ and Csy-AcrIF9-dsDNA$_{NS}$ complexes. Only the Cas8f, Acr, and dsDNA regions are shown. K247 of Cas8f and the residues involved in non-specific DNA binding of AcrIF9 and AcrIF24 are shown in the sphere model. **e** EMSA was used to test the non-specific DNA binding of Csy-AcrIF9 with mutations in AcrIF9 or Csy using 3' FAM-labeled dsDNA$_{SP}$ or 5' FAM-labeled dsDNA$_{NS}$. Reactions were performed as in Fig. 5c. Please see the left two panels of Fig. 8c for the WT control of the experiments using 3' FAM dsDNA$_{SP}$ 15 bp and 25 bp, respectively. Please see the left two panels of Fig. 8b for the WT control of the experiments using 5' FAM dsDNA$_{SP}$ 15 bp and 25 bp, respectively.

and dsDNA$_{NS}$ using similar binding sites and mechanisms. However, the differences observed in the binding experiments of dsDNA with lengths <35-bp suggested that the affinity of dsDNA$_{NS}$ for the Csy complex is necessarily lower than that of dsDNA$_{SP}$, and a stable complex between dsDNA$_{NS}$ and the Csy complex can only be formed when the length of dsDNA$_{NS}$ is long enough (35 bp) to bind to the second DNA-binding site (HTH motif of AcrIF24).

**AcrIF24 and AcrIF9 use a similar strategy, but different mechanisms in inducing non-specific DNA binding.** Since AcrIF9 uses a similar strategy (providing a non-specific DNA binding surface) to induce non-specific DNA binding[17], we investigate the differences between their molecular mechanisms

by comparing their activities in different aspects. First, EMSA results showed that Csy-AcrIF9 mainly forms two types of ternary complexes with dsDNA$_{SP}$, and one type of ternary complex with dsDNA$_{NS}$ (Fig. 9a), consistent with the previous study[17]. Compared with Csy-AcrIF9, Csy-AcrIF24 primarily forms only one type of ternary complex with either dsDNA$_{SP}$ or dsDNA$_{NS}$ (Fig. 9a). For ssDNA, while Csy-AcrIF24 and Csy-AcrIF9 interact with ssDNA$_{SP}$ to form a complex with similar size, the Csy-AcrIF24-ssDNA$_{NS}$ has a larger molecular weight than Csy-AcrIF9-ssDNA$_{NS}$ (Fig. 9b). Second, while AcrIF24 contains a DNA binding motif itself, AcrIF9 does not show DNA binding under the concentrations used in this study (Fig. 9c). Third, in a previous study, R32 and K36 of AcrIF9 have been identified to be involved in non-sequence-specific DNA binding[17]. Structural superimposition between Csy-AcrIF24 and Csy-AcrIF9 indicates

that the motif involved in non-sequence-specific DNA binding of one AcrIF9 is more adjacent, and the other is more distant, to the K-wedge of Cas8f (Fig. 9d). To test this difference, we also performed EMSAs to examine the binding of the above dsDNA$_{SP}$ and dsDNA$_{NS}$ molecules of different lengths by Csy-AcrIF9. Unlike Csy-AcrIF24, Csy-AcrIF9 could bind both dsDNA$_{SP}$ and dsDNA$_{NS}$ molecules with lengths from 15-bp to 54-bp (Fig. 8b, c, and Supplementary Figs. 13a, b). Out of these, we are interested about the binding mode of the 15- and 25-bp dsDNA$_{NS}$/dsDNA$_{SP}$ in the Csy-AcrIF9 complex, since only 15-bp dsDNA$_{SP}$ of these four DNA molecules displayed binding to the Csy-AcrIF24 complex (Fig. 8b, c and Supplementary Figs. 13a, b). Here, mutagenesis studies showed that for 15/25-bp dsDNA$_{NS}$, both Cas8f K247E and AcrIF9 R32A/K36A mutations completely abolish the dsDNA binding (Fig. 9e). For 15/25-bp dsDNA$_{SP}$, while the R32A/K36A mutation of AcrIF9 only caused a weak decrease in DNA binding, the Cas8f K247E mutation markedly impaired the DNA binding (Fig. 9e). The above mutational results can be explained by the same effect as mentioned above for the lower affinity of dsDNA$_{NS}$ for the Csy complex. Taken together, AcrIF24 and AcrIF9 use a similar strategy, but distinct molecular mechanisms in inducing non-specific DNA binding in the Csy complex.

## Discussion

Acr proteins are highly diverse in sequence, structure and inhibition mechanisms. In this study, we reported the structures of AcrIF24, Csy-AcrIF24 and Csy-AcrIF24-dsDNA complexes, and uncovered its structural and biochemical mechanisms. AcrIF24 targets the Cas7f subunits by engaging up to five Cas7f subunits (Cas7.2f-7.6 f), sterically blocks access to the crRNA guide by the complementary target DNA, and induces non-sequence-specific DNA binding to the Csy complex. Previous to our study, induction of dimerization of the Cas effectors was only reported for class 2 Acrs AcrIIA6[29], AcrIIC3[30], and AcrVA4[31–33]. Notably, AcrIIA6 and AcrVA4 also harbor an HTH domain and induces the dimerization of Cas9 from *Streptococcus thermophilus* and Cas12a from *Lachnospiraceae bacterium*, respectively[29,33,34]. Recently, the AcrID1 dimer was reported to sequester the Cas10d large subunit of type I-D system to form a 4:2 complex[35]. Therefore, AcrIF24 also represents an Acr which induces the dimerization of the Cascade complex in class 1 CRISPR-Cas systems and a verified class 1 Acr with Aca-like functions.

Notably, while AcrIF24 displays no sequence or structural similarities to AcrIF9, they both induce Acr-triggered interactions with non-sequence-specific dsDNA in the Csy complex. However, while this activity has been proved important in the Acr activity of AcrIF9, it only plays a minor role in the Acr activity of AcrIF24. Therefore, the functional importance of this non-specific DNA binding is largely unproven for AcrIF24. Interestingly, the DNA binding affinity of AcrIIA11, a type II Acr, was improved through its interacting with Cas9[36]. Future studies should be conducted to investigate whether AcrIIA11 also induces non-sequence-specific DNA binding to Cas9 and the function of this binding.

Structural and biochemical results indicated that the Csy-AcrIF24 complex interacts with dsDNA$_{SP}$ and dsDNA$_{NS}$ through similar mechanisms. On the one hand, the mutational studies showed that the binding of dsDNA$_{SP}$ and dsDNA$_{NS}$ are both markedly decreased or abolished by the mutations of the PAM interaction site of Csy (Fig. 7a) and the R196 and K197 mutations of the AcrIF24-HTH domain (Fig. 7b). On the other hand, the competition EMSA with dsDNA$_{SP}$ and dsDNA$_{NS}$ also suggested that they bind the similar sites of Csy-AcrIF24 (Fig. 5e). That is, for the binding of non-specific DNA by Csy-AcrIF24, the PAM

interaction site of Csy and the HTH domain of AcrIF24 are also involved. However, since the dsDNA$_{NS}$ lacks the PAM region and protospacer sequence, it is sequence non-specifically recognized by the PAM-interacting region of Csy-AcrIF24, but not sequence specifically as dsDNA$_{SP}$.

Interestingly, the HTH motif of AcrIF24 has two functions. On the one hand, it functions as an Aca protein to repress the transcription of the promoter of *acrIF23-24* locus. On the other hand, it also plays a minor role in inhibiting the activity of the CRISPR-Cas complex by inducing non-specific DNA binding. It would be of interest to further investigate any other function of the HTH motif of AcrIF24 and the anti-CRISPR activity of HTH motif in other Acrs.

Notably, the PAM recognition loop of the Cas8f subunit was found to be essential for the binding of both dsDNA$_{SP}$ and dsDNA$_{NS}$. It has been suggested that the Csy complex uses a "PAM scan" method for target location[37], and a recent study also reported that the Cascade complex from type I-E system is rapidly probing DNA, driven by its PAM-interacting subunit Cas8e[38]. Based on the results obtained by our and these previous studies, we propose the following model to illustrate the mechanism of inhibition by AcrIF24. Normally, the Csy complex samples dsDNA with rapid association and dissociation kinetics, driven by the PAM recognition subunit Cas8f. Then, AcrIF24 tightly binds the Csy complex to form a Csy-AcrIF24 complex which still samples dsDNA. For dsDNA with PAM and protospacer sequence, the presence of AcrIF24 in the Csy-AcrIF24 complex precludes the hybridization between the target strand and the crRNA, but facilitates the non-hybridization binding of DNA through its HTH motif, together with the PAM recognition loop (K-wedge) of the Cas8f subunit. For dsDNA with non-target sequence, AcrIF24 may facilitate the stable binding of the DNA after it is non-specifically associated by the PAM recognition loop of the Csy-AcrIF24 complex. In all, our study reveals that AcrIF24 is an Acr with three layers of suppression (i.e., inactivate two Csy complexes at a time, prevention of DNA-crRNA hybridization and induction of non-specific DNA binding).

## Methods

**Protein expression and purification**. The full-length AcrIF24 and Cas2/3 genes were synthesized by GenScript, amplified by PCR, and cloned into pGEX6p-1 to produce GST-tagged fusion proteins with a PreScission Protease cleavage site between GST and the target protein, respectively. The full-length AcrIF9 was cloned into pET28a vector to produce a His$_6$-tagged fusion protein. The Cas8f/Cas5f, Cas7f/Cas6f, and crRNA fragment were cloned into pETDuet-1, pACYC-Duet-1, and pRSFDuet-1 respectively. The Csy complex was generated through co-expression of the three plasmids in *E. coli* strain BL21. Co-expression of Csy-Acr protein complex was generated by adding the Acr gene into the other multiple cloning site of the pRSFDuet-1 vector harboring the crRNA fragment. The mutants of AcrIF24 and Csy complex were generated by two-step PCR and were subcloned, overexpressed and purified in the same way as wild-type proteins. The proteins were expressed in *E. coli* strain BL21 and induced by 0.2 mM isopropyl-β-D-thiogalactopyranoside (IPTG) when the cell density reached an OD$_{600nm}$ of 0.8. For GST-tagged protein, after induction, the cells were harvested, re-suspended in lysis buffer (1× PBS, 2 mM DTT, and 1 mM PMSF) and lysed by sonication. The cell lysate was centrifuged at 20,000 × *g* for 45 min at 4 °C to remove cell debris. The supernatant was applied onto a self-packaged GST-affinity column (2 mL Glutathione Sepharose 4B; GE Healthcare) and contaminant proteins were removed with wash buffer (lysis buffer plus 200 mM NaCl). The fusion protein was then digested with PreScission protease at 4 °C overnight. The protein with an additional five-amino-acid tag (GPLGS) at the N-terminus was eluted with lysis buffer. The eluant was concentrated and further purified using a Superdex-75 (GE Healthcare) column equilibrated with a buffer containing 10 mM Tris-HCl pH 8.0, 200 mM NaCl, and 5 mM DTT.

For His$_6$-tagged protein, the protein was purified through Ni-column, anion exchange chromatography, and gel filtration. The purified protein was analyzed by SDS-PAGE. Selenomethionine (Se-Met)-labeled AcrF24 was expressed in *E. coli* BL21 cells grown in M9 minimal medium supplemented with 35 mg/L Se-Met (Sigma-Aldrich) and specific amino acids: Ile, Leu, and Val at 50 mg/L; Lys, Phe, and Thr at 100 mg/L. The Se-Met protein was purified as the native protein.

**Double-stranded DNA preparation**. For EMSA and in vitro DNA cleavage assay, various 5′-end or 3′-end FAM-labeled single-stranded DNA molecules were synthesized from Sangon, Shanghai, and were hybridized with their complementary unlabeled single-stranded DNA with a molar ratio of 1:1.5 to obtain double-stranded DNA. Specifically, dsDNA$_{SP}$ and dsDNA$_{NS}$ without FAM label were generated through the same method described above except that both strands were unlabeled and added with a 1:1 ratio.

All FAM-labeled full-length single-stranded DNA molecules were listed below, while 15/25/35/45-bp DNA molecules were intercepted from the 5′-end of them.

Target DNA sequence (54 bp; 5′-FAM or 3′-FAM fluorescein labeled) GGAAG CCATCCAGGTAGACGCGGACATCAAGCCCGCCGTGAAGGTGCAGCTGCT. Non-Target DNA sequence (54 bp; 5′-FAM fluorescein labeled). AGCAGCTGCA CCTTCACGGCGGGCTTGATGTCCGCGTCTACCTGGATGGCTTCC. Non-specific DNA sequence (54 bp; 5′-FAM or 3′-FAM fluorescein labeled). GAGCG ACTACGACATGAGCGCGCAGCTAAGACCGCCCGTAGATGCGTCGAG CGT. Inverted repeat DNA sequence (23 bp; 5′-FAM fluorescein labeled). GCATAGCTCGATCCGAGCTAGTA. Mutated inverted repeat DNA sequence (23 bp; 5′-FAM fluorescein labeled). GCAAAGGTCCATCGGACCTTGTA.

**In vitro cleavage assay**. For testing the activity of AcrIF24, AcrIF24ΔMD and their mutants, reactions were performed in a 20 μL buffer system containing 0.32 μM Csy complex, 0.16 μM Cas2/3, 0.04 μM dsDNA, and 0.16-0.64 μM AcrIF24, AcrIF24ΔMD or its mutants. For AcrIF24 N194A/R196A/K197A, the concentrations of AcrIF24 proteins were set as 0.16 μM. We first incubated AcrIF24, AcrIF24ΔMD, or its mutants with Csy at 37 °C in reaction buffer (20 mM HEPES pH 7.5, 100 mM KCl, 5% glycerol, 1 mM TCEP) for 30 min, and DNA was added and incubated for another 30 min. And then Cas2/3 was added to a final concentration of 0.16 μM, along with which 5 mM MgCl$_2$, 75 μM NiSO$_4$, 5 mM CaCl$_2$, and 1 mM ATP were added into the buffer. The reaction was further incubated for 30 min and quenched with the mixture of 5% SDS and 0.25 M EDTA. The products were separated by electrophoresis over 14% polyacrylamide gels containing 7 M urea and visualized by fluorescence imaging. For quantification, the intensity of the upper bands (labeled as substrate in the figures) of all lanes was quantified by the ImageJ software. And then the intensity of the lane with DNA only was set as 100%, and those of all the other lanes were compared with it to obtain the fraction DNA remaining (%). Finally, fraction DNA cleavage (%) was obtained.

**Crystallization, data collection, and structure determination**. The AcrIF24ΔMD protein was concentrated to 8 mg/mL in 10 mM Tris-HCl pH 8.0, 200 mM NaCl and 5 mM DTT. Crystals were grown using the sitting-drop vapor diffusion method. Crystals of AcrIF24ΔMD were grown at 18 °C by mixing an equal volume of the protein (8 mg/mL) with reservoir solution containing 15% 2-propanol and 0.2 M imidazole pH 7.6. The crystals appeared overnight and grew to full size in about two to three days. The crystals were cryoprotected in the reservoir solution containing 30% glycerol before its transferring to liquid nitrogen. Se-Met-labeled protein was crystallized in the same buffer, and the crystal diffraction was good enough. After crystal diffraction tests at home and beamlines BL17U1 and BL19U1 of the Shanghai Synchrotron Radiation Facility (SSRF), the crystal suitable for structure determination were finally obtained.

All the data were collected at SSRF beamlines BL17U1[39] and BL19U1[40], integrated, and scaled using the HKL2000 package[41]. The initial model was solved by Autosol in PHENIX[42] and refined manually using COOT[43]. The structure was further refined with PHENIX[42] against the native data using non-crystallographic symmetry and stereochemistry information as restraints. The final structure was obtained through several rounds of refinement. Data collection and structure refinement statistics are summarized in Supplementary Table 1.

**Multi-angle light scattering (MALS)**. MALS experiments were performed in 20 mM HEPES pH 7.5 and 200 mM NaCl using a Superdex-200 10/300 GL size-exclusion column from GE Healthcare. The used protein concentrations were as follows: AcrIF24 (1.5 mg/mL), AcrIF24ΔMD (15 mg/mL), the Csy complex (1.5 mg/mL), Csy-AcrIF24 complex (1.5 mg/mL), and mixes of AcrIF24 and Csy complex (7.5 mg/mL) with a molar ratio of 1: 0.2, 1: 1, and 1: 5, respectively. The chromatography system was connected to a Wyatt DAWN HELEOS Laser photometer and a Wyatt Optilab T-rEX differential refractometer. Wyatt ASTRA 7.3.2 software was used for data analysis.

**Gel filtration assay**. Protein samples along with the gel filtration standard purchased from Bio-Rad were respectively applied to a SEC column (Superdex-75 increase 10/300 GL, GE Healthcare) equilibrated with a buffer containing 10 mM Tris-HCl pH 8.0, 200 mM NaCl, and 5 mM DTT. The assays were performed with a flow rate of 0.5 mL/min and an injection volume of 0.5 mL for each run. Samples taken from relevant fractions were applied to SDS-PAGE and visualized by Coomassie blue staining. AcrIF24 was incubated with the Csy complex or Cas2/3 at 4 °C for 2 h with a molar ratio of 1:5. The incubation samples were analyzed as described above except that a different chromatography column (Superdex-200 increase 10/300 GL, GE Healthcare) was used.

**Circular dichroism**. Circular dichroism measurements were conducted with Chirascan-plus CD Spectrometer (Applied Photophysics). All the data were collected with 1.0 mg/mL protein samples in 10 mM Tris pH 8.0, 200 mM NaCl buffer over a wavelength range of 195–260 nm, with 1 nm increments, in a 0.01 mm path length rectangular cuvette at 25 °C. All the measurements were performed in triplicate, and the results were averaged.

**Inverted repeats DNA prediction**. The inverted repeats DNA sequence of AcrIF24 in *P. aeruginosa* and other species were predicted by Inverted repeat finder[44] (http://tandem.bu.edu/cgi-bin/irdb) among their promoter sequences.

**Electrophoretic mobility shift assay**. Reactions were performed in a 20 μL buffer system containing 1 μM dsDNA (except for binding affinity calculation), varied concentrations of Csy complex, AcrIF24, Csy-AcrIF24 complex or their truncations or mutants. All binding reactions were conducted at 37 °C for 30 min in the buffer containing 20 mM HEPES pH 7.5, 150 mM NaCl and 5% glycerol. Products of the reaction were separated using 5% native polyacrylamide gels and visualized by fluorescence imaging.

For binding affinity calculation in Fig. 2b, AcrIF24 or AcrIF24ΔMD was incubated in a concentration gradient (0, 0.01, 0.05, 0.1, 0.5, 1, 5, 10, 50, 100, 500, 1000, 5000, 10,000 nM) with 20 nM of 23-bp IR23-24. The fluorescence signal was measured using ImageJ[45]. The bound DNA was quantified and the binding curves were fitted to a one specific binding site model with Prism (GraphPad Software).

For competitive EMSA in Figs. 5e, f, 5, 10, and 20 μM dsDNA$_{NS}$ or ssDNA$_{NS}$ without FAM labels were added after the 30 min incubation of 10 μM Csy complex or Csy-AcrIF24 complex and 1 μM dsDNA$_{SP}$ or ssDNA$_{SP}$ at 37 °C in the reaction buffer mentioned above, respectively.

For WT or mutated Csy-AcrIF24 complex, 0.5, 1, 2, 4, 8, and 16 μM Csy-Acr complex was incubated with 1 μM DNA at 37 °C in the reaction buffer.

For testing the interaction between WT or mutated IR23-24 and AcrIF24 in Fig. 2c–f, 1 μM 23-bp IR23-24 was incubated with 0.125, 0.5, 2, 8, and 32 μM AcrIF24 at 37 °C for 30 min.

For EMSA displayed in Figs. 5a, b and 7b, 0.5, 1, 2, 4, and 8 μM AcrIF24 or its mutants were mixed with 8 μM Csy complex and incubated at 37 °C for 30 min before the addition of 1 μM dsDNA$_{SP}$ or dsDNA$_{NS}$ and further incubation for 30 min.

**β-galactosidase reporter assays**. The 132-bp putative promoter sequence containing IR23-24 was synthesized and cloned into the β-galactosidase reporter plasmid pDN19lacΩ. AcrIF24 or AcrIF24$^{R196A}$ was cloned into the pUCP19 vector with a C-terminal FLAG tag. PA14 harboring transformed plasmids were grown to OD$_{600 nm}$ of 0.8–1.0, and 1 mL bacteria culture was harvested and re-suspended into 3 mL Z buffer (60 mM Na$_2$HPO$_4$, 40 mM NaH$_2$PO$_4$, 10 mM KCl, 1 mM MgSO$_4$, 3.5 μL mL$^{-1}$ β-mercaptoethanol), and OD$_{600 nm}$ was determined immediately. Next, 300 μL mixture was added with 10 μL 0.1% SDS and shaken for 10 s, and then 100 μL ONPG (4 mg/mL) was included and incubated at 37 °C to start the reaction. 500 μL Na$_2$CO$_3$ (1 M) was added to stop the reaction, and reaction time was recorded (min). OD$_{420 nm}$ was recorded for the following equation to obtain the final miller unit.

$$\mathrm{Miller\ Units} = (2000 \times \mathrm{OD}_{420nm})/(\mathrm{T} \times \mathrm{OD}_{600nm})$$

**Binding test with native gel**. For binding test with native gel, 0.32 μM Csy complex was incubated with varying concentrations of AcrIF24 or its mutants at 37 °C for 30 min in the reaction buffer used in EMSA. Products were separated using 5% native polyacrylamide gels and visualized by Coomassie blue staining.

**SPR binding assay**. The SPR analysis was performed using a Biacore 8 K machine (GE Healthcare) at room temperature (25 °C). The Csy complex was immobilized on a CM5 chip to ~10,000 response units. To collect data for kinetic analysis, a concentration series of AcrIF24 in binding buffer (20 mM HEPES pH 7.5 200 mL NaCl and 0.05% (v/v) Tween-20) were injected over the chip at a flow rate of 30 mL/min. AcrIF24 was allowed to associate for 60 s and dissociate for 90 s. Data were analyzed with BIAevaluation 4.1 software (GE Healthcare) by fitting to a 1:1 Langmuir binding fitting model.

**Microscale thermophoresis (MST) measurements**. All MST measurements were performed on a NanoTemper Monolith NT.115 instrument (NanoTemper Technologies, Munich, Germany), using the Standard Treated Capillaries K002 of the supplier. Each of the 16 solutions of one titration series was filled into a capillary, which was measured successively to create the respective data points in the experiment. General settings were applied for all MST experiments as follows: manual temperature control: 25 °C, LED laser: RED, fluorescence measurement before MST: 5 s, MST (IR laser) on: 30 s, fluorescence after MST: 5 s, delay: 25 s. LED power is set at 20% and MST power is set at 80%.

For MST assay, all the proteins were exchanged into the MST buffer (20 mM HEPES pH 7.5, 150 mM KCl, 5% glycerol). The Csy-AcrIF24 and Csy-AcrIF9 complexes were fluorescence-labeled using the Protein Labeling Kit RED-NHS 2nd

Generation (NanoTemper Technologies) at 10 μM. 100 nM fluorescence-labeled Csy-AcrIF24 or Csy-AcrIF9 complexes were added respectively to a 1:1 dilution series with a final concentration of 50 μM down to 1.5269 nM for different types of DNA. Each experiment was conducted at least three times and a similar result was obtained each time. Each protein $K_D$ value was obtained with a signal-to-noise ratio higher than 7. Data sets were processed with the MO (Monolith). Affinity Analysis v2.3 software, using the signal from the initial fluorescence. The analysis of dose–response curves was carried out with Origin 2018.

**Cryo-electron microscopy**. For cryo-EM sample preparation, 4 μL aliquots of Csy-AcrIF24, Csy-AcrIF24-dsDNA$_{SP}$ (3 mg/mL), and Csy-AcrIF24-dsDNA$_{NS}$ sample (2.5 mg/mL) were applied to discharged 200-mesh Au R1.2/1.3 grids (Quantifoil, Micro Tools GmbH, Germany). Grids were blotted for 3 sec and plunged into liquid ethane using an FEI Mark IV Vitrobot operated at 8 °C and 100% humidity. Micrographs were collected on a Titan Krios microscope operated at a voltage of 300 kV by a Gatan K3 Summit direct electron detector with AutoEMation and EPU software at a nominal magnification of ×81,000 using super-resolution mode. The pixel size was 1.0742 Å/pixel and 1.1 Å/pixel, the defocus range was set from −1.3 μm to −2.3 μm. The total dose rate on the detector was ~50 e/Å$^2$ with a total exposure time of 3 s. Each micrograph stack contains 32 frames. All movies were corrected using MotionCor2 for sub-region motion correction and dose-weighting 46. Gctf was used to estimate the defocus values[46].

The 2374 CTF-corrected cryo-EM images of Csy-AcrIF24 complex were manually selected. In all, 1,122,799 particles were auto-picked on micrographs with dose-weighting using RELION 3.1.0[47]. Two rounds of 2D classification and two rounds of 3D classification were done with $K = 5$ classes and regularization parameter $T = 10$, which resulted in 256,130 good particles. 150,969 good particles were subjected to further 3D auto-refinement with C1 symmetry. The refinement resulted in an overall structure at a resolution of 3.11 Å, to further improve the resolution, we performed CTF refinement, which yielded a map at 2.88 Å resolution. In order to obtain a high-resolution reconstruction of Csy-AcrIF24 monomer and Cas8f, all good particles were signal subtracted from Csy-AcrIF24 monomer and then subjected to local re-alignment and 3D classification without alignment with a generous mask around Cas8f. Finally, 274,872 good Csy-AcrIF24 monomer particles were refined and three rounds of CTF parameter refinement, and post processed separately to give final maps at 2.82 Å resolution and Cas8f focused maps at 2.71 Å resolution.

For the data set of the Csy-AcrIF24-dsDNA$_{SP}$ complex, a total number of 2,793 movie stacks were acquired. A total of 933,567 particles were auto-picked using RELION 3.1.0, two rounds of 2D classification and two rounds of 3D classification resulted in 653,562 good particles with $K = 5$ classes and regularization parameter $T = 10$. Finally, 188,384 particles were refined and subjected to 3D refinement at 3.02 Å resolution in an overall structure. In order to obtain a high-resolution Csy-AcrIF24-dsDNA$_{SP}$ monomer structure, 188,384 particles were signal subtracted of Csy-AcrIF24-dsDNA$_{SP}$ monomer, and then subjected to local re-alignment and 3D classification without alignment with a generous mask around Cas8f ($K = 3$, $T = 20$). Finally, 212,780 good Csy-AcrIF24-dsDNA$_{SP}$ monomer particles were refined and three rounds of CTF parameter refinement, and post processed at 2.94 Å resolution and Cas8f focused maps at 2.61 Å resolution for model building.

For the data set of the Csy-AcrIF24-dsDNA$_{NS}$ complex were processed similarly to the Csy-AcrIF24 complex dataset. 4,294 CTF-corrected cryo-EM images were manually selected and totally extracted 1,079,487 particles from three rounds of 2D classification and two rounds of 3D classification with $K = 5$ classes. Finally, 474,421 particles were subjected to 3D refinement at 3.54 Å resolution in Csy-AcrIF24-dsDNA$_{NS}$ dimer structure and the final resolution is 3.20 Å. Csy-AcrIF24-dsDNA$_{NS}$ monomer particles were signal subtracted, and subjected to local re-alignment and 3D classification without alignment with a generous mask around Cas8f ($K = 8$, $T = 10$). Finally, 77,361 good Csy-AcrIF24-dsDNA$_{SP}$-Cas8f particles were refined and three rounds of CTF parameter refinement, Bayesian polishing, and post processed at 3.28 Å resolution for model building.

**Model building, refinement, and validation**. Atomic model of the Csy complex was modeled using the PDB 6B45. The AcrIF24 model was manually modeled by COOT[43] based on the crystal structure of AcrIF24ΔMD. The overall models of Csy-AcrIF24, Csy-AcrIF24-dsDNASP complexes, and Csy-AcrIF24-dsDNA$_{NS}$ were subjected to global refinement and minimization in real_space_refinement using PHENIX[42]. All the Figures were created in PyMOL[48], Chimera[49], and ChimeraX[50].

**Phage plaque assay**. Phage replication and plaque assays were performed by the method reported by Buyukyoruk et al.[28] with modifications. For phage replication, 100 μL of *E. coli* MG1655 bacterial culture grown to an OD$_{600 \, nm}$ of 0.5 was mixed with 5 mL soft LB agar and poured over an LB agar plate. Lambda (λ) phage was inoculated by the streak plate method with 3 mm intervals. Then, the plate was incubated at 37 °C overnight, and 5 mL SM buffer containing 50 mM Tris-HCl pH 7.5, 100 mM NaCl, 10 mM MgSO$_4$ was added to the plate to shake for 5 h. After filtration with a 0.2 μm filter, the lysate was used for the following phage plaque assay.

*E. coli* NovaBlue (DE3) cells were transformed with the type I-F system from *P. aeruginosa* (components of the Csy complex in pBAD24 vector, Cas2/3 in pET28a vector, CRISPR array targeting the genome of λ phage in the multiple cloning site 2 of pACYCDuet vector), type I-F system with WT or mutant AcrIF24 (in the multiple cloning site 1 of the CRISPR array vector), or corresponding empty vectors. For phage plaque assay, 100 μL of the *E. coli* NovaBlue (DE3) cells grown to OD$_{600 \, nm}$ of 0.5 was pre-induced by 0.25% arabinose and 0.25 mM IPTG for 3 h shaking at 37 °C, and subsequently mixed with 100 μL of phage lysate diluted 10$^6$ times and then incubated for 30 min. Next, the mixture was added to 5 mL of soft LB agar with additional 0.25% arabinose and 0.25 mM IPTG and poured over an LB agar plate. Plates were incubated at 37 °C overnight and plaques were counted the following day. The efficiency of plaquing (EOP) is the ratio of plaque-forming units of each strain compared with that harboring both type I-F system and WT AcrIF24. Plaque assays were performed in triplicate and the shown EOP is the mean ± standard deviation. A two-sided *t* test was performed (*$P < 0.05$).

**Statistics and reproducibility**. All experiments in the present manuscript were repeated at least three times independently with similar results, including those in Figs. 2c, e, g, 4f, g, 5a–f, 7a–c, 8b–e, 9a–c, 9e, supplementary figures 1, 2c, 2h, i, 3a–c, 4a, 10a–b, 11, 13a–c.

**Reporting summary**. Further information on research design is available in the Nature Research Reporting Summary linked to this article.

## Data availability

Cryo-EM maps have been deposited at the Electron Microscopy Data Bank (EMDB) with the following accession codes: EMD-31185 (Csy-AcrIF24), EMD-31186 (Csy-AcrIF24-dsDNASP), EMD-32440 (Csy-AcrIF24-dsDNANS) and EMD-32387 (the Cas8f region of the Csy-AcrIF24-dsDNANS). The atomic coordinates have been deposited at the Protein Data Bank (PDB) with the following accession codes: 7DTR (AcrIF24ΔMD), 7ELM (Csy-AcrIF24), 7ELN (Csy-AcrIF24-dsDNASP), 7WE6 (Csy-AcrIF24-dsDNANS). Structures of the Csy complex (PDB: 6B45), ClgR from *Corynebacterium glutamicum* (PDB: 3F51) and C.Csp231I complexed with DNA (PDB: 4JQD) were referenced in the manuscript. Source data are provided with this paper.

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

## Acknowledgements

We thank professor Jianlin Lei and Fan Yang (Tsinghua University), professor Peiyi Wang and Xiaomin Ma (Southern University of Science and Technology) for EM data collection. We thank professor Weihui Wu for kindly providing the pDN19lacΩ plasmid. We thank the Tsinghua University Branch of the China National Center for Protein Sciences (Beijing) and the Southern University of Science and Technology for providing cryo-EM facility support. We would like to thank the staff at beamlines BL17U1 and BL19U1 of the SSRF for their assistance with data collection. We would like to thank Mrs. Wu Yao (State Key Laboratory of Plant Genomics, Institute of Microbiology, Chinese Academy of Sciences) for assistance in the MST experiment. We would like to thank the Tsinghua University Branch of China National Center for Protein Sciences Beijing and Shilong Fan for providing facility support for X-ray diffraction of the crystal samples. This work was supported by the National Natural Science Foundation of China (32171274, 31822012, and 32000901), the National key research and development program of China (2017YFA0506500, 2019YFC1200500, and 2019YFC1200502), State Key Program of National Natural Science of China (U1808202), NSFC International (regional) cooperation and exchange program (31961143024), Beijing Nova program, the Beijing Natural Science Foundation (5204038) and the Fundamental Research Funds for the Central Universities (XK1802-8).

## Author contributions

Y.F. conceived and supervised the project. L.Y., P.Y., and H.D. purified the proteins, grew and optimized the crystals, and performed the activity analysis and binding assays supervised by Y.F. and Z.C. Y.F. and Y.Z. collected the diffraction data and solved the crystal structure with the help of Y.X., H.Z., and Q.W. L.Z. collected the cryo-EM data and solved the cryo-EM structures with the help of J.Z. and W.W., supervised by M.Y. Y.F. analyzed the data and wrote the paper with assistance from all the authors.

## Competing interests

The authors declare no competing interests.
