## [Peer Review File · Nature Communications]

Insights into the inhibition of type I-F CRISPR-Cas system by a multifunctional anti-CRISPR protein AcrIF24REVIEWER COMMENTS

Reviewer #1 (Remarks to the Author):

This manuscript describes the characterization of an anti-CRISPR, AcrIF24, that inhibits the type I-F CRISPR-Cas of *P. aeruginosa*. The authors have determined the X-ray crystal structure of AcrIF24 on its own, a cryoEM structure of AcrIF24 bound to the I-F CRISPR-Cas complex (Csy complex), and a cryoEM structure of a ternary complex of AcrIF24:Csy:DNA. AcrIF24 possesses a C-terminal domain that folds into a helix-turn-helix DNA-binding motif. The authors show that this domain binds to an inverted repeat sequence in the putative promoter region for the *acrIF24* gene. The authors also show that the AcrIF24:Csy complex binds to DNA in a non-specific manner. This non-specific binding is dependent on the HTH domain. This paper presents a large amount of new data that will be of interest to the field. However, there are deficiencies that must be addressed.

More than half of the results section of this paper is focused on the non-specific DNA-binding activity of the AcrIF24:Csy complex. The authors claim that AcrIF24 mutants with impaired non-specific DNA binding also show decreased capacity to inhibit activity of the CRISPR-Cas system. They compare the non-specific DNA-binding activity of the AcrIF24:Csy complex with that of the previously characterized AcrIF9:Csy complex and suggest that Acr-induced non-specific DNA binding may be a common feature that is selected for by evolution. While the AcrIF24:Csy complex clearly displays non-specific DNA-binding activity, the authors have provided no evidence that there is any biological relevance to this activity. The non-specific DNA-binding activity is mediated by the HTH domain. The authors show that this domain has non-specific DNA-binding activity (nsDNA-binding) on its own. It is not so surprising then that placing this HTH domain in the vicinity of the Csy complex, which also has an nsDNA binding surface required for stable R-loop formation, results in a nsDNA binding activity.

The authors claim that single amino acid substitutions in the AcrIF24 HTH domain, which abrogate the DNA-binding activity of the domain, also reduce its ability to block *in vitro* DNA cleavage by the complete CRISPR-Cas system. However, in examining the raw data for the quantitation of these assays shown in Fig. 7d, I am entirely unconvinced of the significance. The authors do not even explain how these data are quantitated, and they test only a single Acr concentration. In general, the *in vitro* cleavage assays are not good assays for evaluating the inhibitory activity of AcrIF24 and its mutants. Inhibition is not complete even with the highest AcrIF24 concentration. There is also variation in band intensities with different mutants tested. The results are difficult to quantitate. To make the study of mutants satisfactory, the authors should use an *in vivo* CRISPR-Cas inhibition assay. Phage plaquing assays, as have been used in previous studies, are simple to perform and should be used here. This issue is particularly critical in characterizing mutants that have lost nsDNA-binding activity.

The HTH domain of AcrIF24 likely functions to repress the *acr* promoter. Repression of *acr* promoters by HTH-containing Aca proteins has been shown to be crucial and the HTH domain here likely possesses the same function. The authors are proposing that the HTH domain has two functions: the likely repressor function and a role in inhibiting the activity of the CRISPR-Cas complex. Since this would be new function for an HTH domain associated with an anti-CRISPR, it would be an interesting result. However, the onus is on the authors to provide strong evidence. They have not done this. The authors must determine whether the anti-CRISPR activity of AcrIF24 remains if the C-terminal HTH domain is deleted. The activity must be measured accurately *in vivo* and *in vitro*. It is important to determine whether in the absence of the HTH domain AcrIF24 would inhibit the spDNA-binding activity of the Csy complex. The authors could also address whether Csy complex dimerization through AcrIF24 is important for its inhibitory activity. Measuring inhibitory activity of the F212/H216/Y217 mutant would be relevant to this issue.

The authors use EMSAs to show that the AcrIF24 Aca domain binds to the putative promoter region upstream of the *acrIF24* gene. This is an expected result because many *acr* operons contain HTH family repressors that attenuate transcription from the promoter upstream of the *acr* gene. There are several other examples where the HTH domain is fused to an Acr domain to produce a bifunctional protein with Acr activity and repressor activity. The authors conclude that AcrIF24 functions in a manner similar to other Acr-Repressor fusion proteins, but they do not prove this. What they are calling the promoter region is only a putative promoter since its activity has not been tested *in vivo*. The authors should set up an *in vivo* transcriptional assay similar to those described in other studies (e.g. use a b-gal fusion system) to prove that they have identified the *acr* gene promoter and that AcrIF24 really represses transcription from this promoter. In the absence of these data, the authors should clearly speak of a “putative” promoter and clearly acknowledge the uncertainty of their conclusions.

The description of the interaction between AcrIF24 and the Csy complex is very detailed with respect to describing individual residue-residue interactions. This is more detailed than necessary for most journal readers. However, the authors leave out some information that would be of broader interest. How large are the interfaces with the Csy complex formed by the NTD and the MD. Are these domains binding to the same regions bound by other I-F Acrs? This is a particularly important point. I would like to know whether AcrIF24 is interacting with the Csy complex in a unique manner.

Fig. 1a. The authors should explain their *in vitro* cleavage assay. Why is there a single cleavage product resulting from Cas3-mediated cleavage? How is that product produced? The effect of AcrIF24 in the cleavage assay is not very dramatic. What was the percent reduction in cleavage activity? The authors include no discussion of assay reproducibility.

Fig. 4f: The authors should discuss these data. Why does the highest concentration of F24 show shifted bands and 2 different mobilities?

My overall impression of this paper is that the data presented will be of interest to those with interest in CRISPR-Cas systems and how they are inhibited. However, many papers have now been published showing structures of Acrs bound to CRISPR-Cas complexes. The dsDNA-binding story is not fleshed out enough to have any meaning. Thus, I do not feel that this paper warrants publication in Nature Communications.

Reviewer #2 (Remarks to the Author):

The authors performed the structure-function analysis of the anti-CRISPR protein AcrIF24, combining complementary structural and biochemical approaches. They showed that AcrIF24 is the first example of a multifunctional anti-CRISPR protein targeting the class 1 CRISPR-Cas system. First, AcrIF24 forms homodimers in which the C-terminal domains harboring HTH motifs bind to the *acrIF23-24* promoter, and thereby functions as a transcriptional repressor. Second, one AcrIF24 homodimer can bind and inactivate two Csy complexes making it a potent CRISPR-Cas inhibitor. Third, the AcrIF24 N-terminal domain interacts with several Cas7f subunits of the Csy complex and by doing so prevents DNA-RNA hybridization. Lastly, the AcrIF24 C-terminal domain within the Csy-AcrIF24 complex is important to stabilize an unproductive DNA-bound state whether it is a specific or non-specific DNA. Although one or the other of these characteristics have already been described for various anti-CRISPR proteins (the type II AcrIIA1 and AcrIIA13-15 anti-CRISPR proteins function as transcriptional repressor; the type II/V AcrIIA6, AcrIIC3 and AcrVA4 induce Cas9/Cas12 dimerization; the type I AcrIF9 tethers non-specific DNA to the Csy complex), it is remarkable that AcrIF24 combines all these functionalities. This work expands our knowledge of anti-CRISPR proteins and is therefore significant to the field.

Overall the text and the figures are clear, with the exception of some panels showing 3D structures (see major point 5). The results are validated by different experimental approaches and appropriate controls, and they are presented and interpreted in the context of previous work.

Major points

1) I have found the sections entitled “The Csy-AcrIF24 complex interact with dsDNASP and dsDNANS through different mechanisms” (lines 383-438) and “AcrIF24 and AcrIF9 show different characteristics in inducing non-specific DNA binding” (lines 440-474) very confusing. In particular, I have some concerns regarding data interpretation and conclusions.

- In the section “The Csy-AcrIF24 complex interact with dsDNASP and dsDNANS through different mechanisms” (lines 383-438), what the EMSA experiments show, using dsDNASP and dsDNANS of different lengths and positions of fluorescein label, and wild-type or mutants of Cas8f and AcrIF24, is that the Cas8f PAM-binding site and AcrIF24 C-terminal domains are both involved in binding dsDNASP and

dsDNANS. Therefore, the Csy-AcrIF24 complex interacts with dsDNASP and dsDNANS using similar binding sites and mechanisms. The differences observed in the formation of Csy-AcrIF25-dsDNA complexes in the presence of dsDNASP-15bp, or of dsDNANS with length less than 35bp, are explained by the fact that the affinity of dsDNANS for the Csy complex being necessarily lower than that of dsDNASP, stable complex between dsDNANS and the Csy complex cannot be formed. It is only when the length of dsDNANS is long enough (35 bp) to bind to the second DNA-binding sites, that the complex Csy-AcrIF24-dsDNANS can form.

- In the section “AcrIF24 and AcrIF9 show different characteristics in inducing non-specific DNA binding” (lines 440-474), the authors begin by describing EMSA results (lines 444-450) without providing any interpretation on the mechanisms used by AcrIF24 and AcrIF9 to induce non-specific DNA binding. Second, the EMSA results obtained with the Cas8f and AcrIF9 mutants can be explained by the same effect as mentioned in my previous comment. The affinity of the dsDNASP for the PAM-binding site is high enough to form stable Csy-AcrIF9-dsDNASP complex in which the AcrIF9 DNA-binding residues are mutated. However, the lower affinity of dsDNANS for the PAM-binding site prevents its binding to the Csy-AcrIF9 complex in which the AcrIF9 DNA binding residues are mutated. Therefore, the sentence “The above mutational results indicated that the K-wedge of Cas8f is involved in the binding of both dsDNASP and dsDNANS, while R32 and K36 of AcrIF9 are mainly involved in the binding of dsDNANS of 15/25bp” (lines 469-471) is not correct. Moreover, it is hard to conceive how the AcrIF9 DNA-binding site could discriminate between specific and non-specific target DNA.

- In both sections, the authors focus on determining the minimal length of DNA tethered to the Acr-bound Csy complex. But is this relevant to the AcrIF24 mode of action? Indeed, as proposed by the authors in the discussion (lines 505-516), the AcrIF24-Csy complex scans foreign DNA and can either find and bind to its specific PAM and target sequence, or can bind to non-specific DNA while scanning the whole DNA molecule. In this context, the minimal length of DNA that can bind to the AcrIF24-Csy complex is meaningless.

All in all, the results presented in these sections show that AcrIF24 and AcrIF9 use a similar strategy to induce non-specific DNA-binding to the Csy complex, which consists in providing DNA-binding sites that, together with the PAM-binding site, stabilize non-productive DNA-bound states. However, the molecular mechanisms used by these anti-CRISPR proteins are different: the AcrIF24 dimer provides a DNA-binding domain rather distant from the PAM-binding site, while AcrIF9 provides two DNA-binding sites one being close to the PAM-binding site and the other being more distant.

2) The last sentence of the discussion “In all, our study reveals that AcrIF24 is an Acr with two layers of suppression, highlight the prevalence of induction of non-specific DNA binding, and provides striking example of convergent evolution of Acrs” (lines 516-518) should be amended. Since AcrIF24 can inactivate two Csy complexes at a time, I suggest to mention three “layers of suppression”. Also, to my knowledge, there are currently three examples of Acr inducing non-specific DNA binding including

AcrIIA11, AcrIF9 and AcrIF24, therefore “prevalence” (also written in the abstract, line 42) is not really appropriate.

3) The authors have co-expressed AcrIF24 and the Csy complex, and found that they form homotetrameric assemblies. Since AcrIF24 induces non-specific DNA binding to the Csy complex, I was expecting to read that the co-expressed Csy-AcrIF24 complex co-purified DNA as it is highly often the case with DNA-binding proteins. If so, high salt washes are usually enough to get rid of non-specific DNA in protein preparation.

Moreover, it is not clear in the text when and why they used the co-expressed Csy-AcrIF24 complex. For cryoEM experiments, did they use this assembly or did they mix the different partners (lines 199-201, 309-311)? Why did they use the co-expressed assembly for titration experiments (lines 290-293)? In this case, it is crucial to confirm that the co-expressed assembly is not purified with DNA.

4) Figure 3A.

The concentrations of AcrIF24 in solution are not appropriate, as shown by the poor fitting at high concentrations. If the KD is around 4 nM (as indicated in line 186), concentrations should instead be between 0.4 and 40 nM.

5) Figures 3D, 3E, 4A, 4B, 5A, 5B, 9D, S5A.

It is hard to see anything in these views. Increasing their size and/or changing colors to improve the contrast could help.

6) lines 134-139: “The NTD comprises a five-stranded antiparallel β sheet and a bent α helix. A Dali search with this domain revealed that the closest structural homolog is an auxiliary protein of soluble methane monooxygenase hydroxylase (MMOH), MMOD (Fig. 1e)^{22,23}. Interestingly, MMOD is an inhibitor of MMOH through associating with the canyon region of MMOH and inducing conformational changes of it.”

I do not understand what is the relevance of this comparison to decipher the mechanism of AcrIF24. Figure 1E should go in supplementary.

Minor points

1) Results of DALI search should be provided with the corresponding Z-score, rmsd, and number of aligned residues to judge on the significance of structural similarity (lines 135-137, 140-141).

2) lines 204, 210, 212, 313.

Are they RMSD on the backbone only or on the whole model?

3) line 233 “sidechain nitrogen atom of Cas7.6fL76”.

Main chain instead of side chain

4) lines 484-486 “AcrIIA6 and AcrVA4 also harbor HTH domain and induces the dimerization of Cas9 from *Streptococcus thermophilus*”.

AcrVA4 induces the dimerization of LbCas12

5) line 646 “AcrIF24 Δ MMD (15 mg/mL)”.

1.5 mg/mL instead of 15 mg/mL ?

6) Figure 8A.

What is the color code?

Reviewer #3 (Remarks to the Author):

Yang et al. present an X-ray structure of AcrIF24 Δ MMD, and cryo-EM structures of AcrIF24-Csy and AcrIF24-Csy-dsDNASP complex. AcrIF24 has not been characterized structurally or mechanistically. Using a combination of structural and biochemical analyses, the authors showed binding of AcrIF24 to Csy complex by interacting with Cas7 subunits block the hybridization of DNA to the crRNA, similar to AcrIF9. Remarkably, binding of AcrIF24 also induces the non-specific DNA binding to Csy complex, in which the PAM recognition motif and HTH motif plays an essential role.

The AcrIF24-Csy complex is interesting given that AcrIF24 contains a HTH motif that might be involved in the self-regulation. The authors give the first structural description of the HTH motif containing anti-CRISPR protein in complex with Csy complex. It would be of great interest that the authors give a discussion on how this dual anti-CRISPR protein works in both regulation of its own promotor and inhibition of target DNA cleavage.

Major points:

1) As shown in Csy-AcrIF24 complex (Supplementary Fig. 5b and 5c), AcrIF24 engages the target DNA binding site in the Csy complex to block hybridization of DNA to the crRNA. However, as judged by EMSA in Fig. 5a compared with Fig. 5c, it seems Csy-AcrIF24 complex still specifically binds to dsDNASP. Similarly, as shown in Fig. 9b, Csy-AcrIF9 still binds to ssDNA, what's the binding mode for specific target dsDNA and ssDNA binding of Csy-AcrIF9?

The DNA bound bands of right panel in Fig.5c are different from those in left panel shown in Fig. 5a, but both figures show the dsDNASP binding of Csy-AcrIF24, which is difficult to explain.

2) In Fig. 6d, the specific target DNA is split upon binding to Csy-AcrIF24 (Fig. 6c and 6d), but the model shown in Fig. 7e indicates it binds to Cas8f and AcrIF24 without being split, which is also difficult to explain. The structural data (Fig. 6c and 6d) is not supportive to the proposed model in Fig. 7e. Maybe it would be useful to get the complex structure of Csy-AcrIF24-dsDNANS for investigating the AcrIF24-mediated inhibition mechanisms by non-specific DNA binding.

3) Density of residue K247 should be shown in Fig. 6d to indicate the interaction mode between K247 and PAM sequence in target DNA, given previous studies showed K247 is crucial for PAM recognition. It's reasonable that mutation of K247 abolished the binding ability of Csy-AcrIF24 to DNASP, but it's difficult to explain that mutation of K247 also abolished the DNANS binding ability.

4) The WT controls in several figures are missing, such as Fig. 2e, 2f, 5b, 7a, 7c, 8d and 9e.

Minor points:

1. It would be useful to draw a schematic of AcrIF24 in Fig. 1b.

2. It's hard to see the grey color in Fig. 1e and 1f, the author should change it into a different color.

3. As shown in left panel of Fig. 2c, it's interesting that there are two DNA bound bands observed for IR23-24 but just one band for mutated IR23-24. Does the author have some explanation?

4. It would be helpful to color the domains of AcrIF24 differently in Fig. 4a-4e.

5. The densities of residues in Fig4b-4e should be shown.

6. Local resolution map of the cryoEM structures including bound DNA will be useful for evaluation of these structures.

7. It would help if the author provides the distance between the K-wedge of Cas8f and the non-specific DNA binding motif of AcrIF9, and the length of bound DNA, specifically when the author shows the EMSA results with different DNA lengths.

8. In supplementary Fig. 5a, the coloring of Acrs should be rearranged, now it's unclear.

REVIEWER COMMENTS

Reviewer #1 (Remarks to the Author):

This manuscript describes the characterization of an anti-CRISPR, AcrIF24, that inhibits the type I-F CRISPR-Cas of *P. aeruginosa*. The authors have determined the X-ray crystal structure of AcrIF24 on its own, a cryoEM structure of AcrIF24 bound to the I-F CRISPR-Cas complex (Csy complex), and a cryoEM structure of a ternary complex of AcrIF24:Csy:DNA. AcrIF24 possesses a C-terminal domain that folds into a helix-turn-helix DNA-binding motif. The authors show that this domain binds to an inverted repeat sequence in the putative promoter region for the *acrIF24* gene. The authors also show that the AcrIF24:Csy complex binds to DNA in a non-specific manner. This non-specific binding is dependent on the HTH domain. This paper presents a large amount of new data that will be of interest to the field. However, there are deficiencies that must be addressed.

We appreciate the reviewer's high evaluation of our work and constructive suggestions. We have addressed all of the comments listed below with additional experiments and have revised the manuscript accordingly, which we believe have significantly improved our manuscript.

More than half of the results section of this paper is focused on the non-specific DNA-binding activity of the AcrIF24:Csy complex. The authors claim that AcrIF24 mutants with impaired non-specific DNA binding also show decreased capacity to inhibit activity of the CRISPR-Cas system. They compare the non-specific DNA-binding activity of the AcrIF24:Csy complex with that of the previously characterized AcrIF9:Csy complex and suggest that Acr-induced non-specific DNA binding may be a common feature that is selected for by evolution. While the AcrIF24:Csy complex clearly displays non-specific DNA-binding activity, the authors have provided no evidence that there is any biological relevance to this activity. The non-specific DNA-binding activity is mediated by the HTH domain. The authors show that this domain has non-specific DNA-binding activity (nsDNA-binding) on its own. It is not so surprising then that placing this HTH domain in the vicinity of the Csy complex, which also has an nsDNA binding surface required for stable R-loop formation, results in a nsDNA binding activity.

Our responses:

As the reviewer suggested, both the Csy complex and AcrIF24 themselves have weak nsDNA binding activity. However, the nsDNA binding activity of Csy-AcrIF24 is still very interesting. As shown in the right panel of Fig. 5a in the original manuscript, 8 μ M AcrIF24 or Csy on their own display very weak (the last lane) or no binding (the 2nd lane) to 1 μ M dsDNA_{NS}, respectively. However, both the mix of them (each also at 8 μ M, the 6th lane of the original Fig. 5a) and the form of Csy-AcrIF24 complex (also 8 μ M, the last but one lane of the original Fig. 5d) showed complete binding of 1 μ M dsDNA_{NS}. This suggests that the association between Csy and AcrIF24 may also display a synergistic effect of the nsDNA binding activity, significantly higher than those of themselves. Meanwhile, as the reviewer suggested, it is the first report of the involvement of the HTH domain in anti-CRISPR functions. As stated below and suggested by the reviewer, we also elucidated the biological relevance to this activity. Overall, we think that our revisions suggested by the reviewer have significantly improved our work, especially in the biological significance of our findings.

The authors claim that single amino acid substitutions in the AcrIF24 HTH domain, which abrogate the DNA-binding activity of the domain, also reduce its ability to block *in vitro* DNA cleavage by the complete CRISPR-Cas system. However, in examining the raw data for the quantitation of these assays shown in Fig. 7d, I am entirely unconvinced of the significance. The authors do not even explain how these data are quantitated, and they test only a single Acr concentration. In general, the *in vitro* cleavage assays are not good assays for evaluating the inhibitory activity of AcrIF24 and its mutants. Inhibition is not complete even with the highest AcrIF24 concentration. There is also variation in band intensities with different mutants tested. The results are difficult to quantitate. To make the study of mutants satisfactory, the authors should use an *in vivo* CRISPR-Cas inhibition assay. Phage plaquing assays, as have been used in previous studies, are simple to perform and should be used here. This issue is particularly critical in characterizing mutants that have lost nsDNA-binding activity.

Our responses:

The reviewer raised a very good point. For Fig. 7d and all other figures of *in vitro* DNA cleavage in the manuscript, the quantitation method is as follows. The intensity of the upper bands (labeled as substrate in the figures) of all lanes were quantified by the ImageJ software. And then the intensity of the lane with DNA only was set as 100%, and those of all the other lanes were compared with it to obtain the fraction DNA remaining (%). Finally, fraction DNA cleavage (%) was obtained. We have included this quantification detail into the Methods section in the revised manuscript.

However, as the reviewer pointed out, we agree that there are limitations with the *in vitro* DNA cleavage assays. Therefore, following the reviewer's suggestion, we used phage plaquing assays to further verify the inhibitory activity of AcrIF24 and its mutants. As reported by Buyukroruk et al.¹, type I-F system from *P. aeruginosa* could be transformed into *E. coli* to target lambda (λ) phage. We have adopted this system to test the effect of AcrIF24 and its HTH mutations *in vivo*. Phage titration assays showed that while λ phage can infect *E. coli* NovaBlue (DE3) cells transformed with empty vectors, cells expressing Csy, Cas2/3, and a synthetic CRISPR array designed to target λ phage successfully caused about 6-log reduction of plaques (Response figure 1). However, co-transforming AcrIF24 into the cells expressing Csy, Cas2/3, and the synthetic CRISPR array rendered the cells sensitive to λ phage again, indicating the inhibition of the type I-F CRISPR-Cas by AcrIF24 (Response figure 1). Phage plaquing assay was then conducted to accurately evaluate the inhibitory activity of AcrIF24 and its mutants. As revealed in revised Fig. 7e, while AcrIF24 N194A exhibited similar inhibitory activity as WT AcrIF24, both AcrIF24 R196A and AcrIF24 K197A showed about two to three-fold reductions in activity. This is also consistent with their roles in inducing non-specific DNA binding and *in vitro* DNA cleavage assay. Together, this result provided the biological relevance of the non-specific DNA binding activity endowed by the HTH motif of AcrIF24.

Response figure 1. Reconstitution of type I-F CRISPR system in *E. coli* NovaBlue (DE3) cells

400 μ L of *E. coli* NovaBlue (DE3) cells grown to OD_{600 nm} of 0.9 was added into 5 ml of soft LB/agar with

0.25% arabinose and 0.25 mM IPTG and poured onto LB agar plates. 3 μ L of 10-fold dilution of λ phage lysate was titrated on the plate surface, and the plates were incubated at 37°C overnight.

The HTH domain of AcrIF24 likely functions to repress the *acr* promoter. Repression of *acr* promoters by HTH-containing Aca proteins has been shown to be crucial and the HTH domain here likely possesses the same function. The authors are proposing that the HTH domain has two functions: the likely repressor function and a role in inhibiting the activity of the CRISPR-Cas complex. Since this would be new function for an HTH domain associated with an anti-CRISPR, it would be an interesting result. However, the onus is on the authors to provide strong evidence. They have not done this. The authors must determine whether the anti-CRISPR activity of AcrIF24 remains if the C-terminal HTH domain is deleted. The activity must be measured accurately *in vivo* and *in vitro*. It is important to determine whether in the absence of the HTH domain AcrIF24 would inhibit the spDNA-binding activity of the Csy complex. The authors could also address whether Csy complex dimerization through AcrIF24 is important for its inhibitory activity. Measuring inhibitory activity of the F212/H216/Y217 mutant would be relevant to this issue.

Our responses:

The reviewer raised a very good point. The C-terminal HTH domain ranges from 138 to 228 residues of AcrIF24. To investigate the function of this domain, first we have tried four truncations of AcrIF24, i.e., residues 1-138, 1-149, 1-164 and 1-177, out of which only AcrIF24¹⁻¹³⁸ (hereafter named as AcrIF24 Δ CTD) could be purified, but with very little amount and poor stability (Response Figure 2). For the F212/H216/Y217 mutants of full-length AcrIF24, as stated in the original manuscript, they all displayed extremely poor solubility during expression. During the revision, although we have tried a lot to optimize the purification process of the mutants, we could only get very little amount of degraded AcrIF24 FHY/EEE protein (Response Figure 2). This suggests that both the HTH domain and dimerization of AcrIF24 are essential for its structural integrity and stability. Due to the poor behavior of the two purified mutant proteins, *in vitro* biochemical assays (native gel test for Csy binding, EMSA test for inhibition of DNA_{SP} binding and *in vitro* cleavage assay) could not be performed. Then we turned to *in vivo* phage plaquing assay to test the function of the HTH domain and dimerization of AcrIF24. As revealed by the *in vivo* assay, AcrIF24 Δ CTD and AcrIF24 FHY/EEE both exhibited reductions in inhibitory activity (revised Fig. 7e). Taken together, both the HTH domain and dimerization of AcrIF24 are essential for the structural integrity/stability of AcrIF24 and contribute to its inhibitory activity.

Response figure 2. Purified AcrIF24 and its mutants

Shown are the SDS-PAGE gel of WT AcrIF24, AcrIF24 F212E/H216E/Y217E and AcrIF24 Δ CTD mutants. The two AcrIF24 mutant proteins displayed poor solubility and stability.

The authors use EMSAs to show that the AcrIF24 Aca domain binds to the putative promoter region upstream of the *acrIF24* gene. This is an expected result because many *acr* operons contain HTH family repressors that attenuate transcription from the promoter upstream of the *acr* gene. There are several other examples where the HTH domain is fused to an Acr domain to produce a bifunctional protein with Acr activity and repressor activity. The authors conclude that AcrIF24 functions in a manner similar to other Acr-Repressor fusion proteins, but they do not prove this. What they are calling the promoter region is only a putative promoter since its activity has not been tested *in vivo*. The authors should set up an *in vivo* transcriptional assay similar to those described in other studies (e.g. use a β -gal fusion system) to prove that they have identified the *acr* gene promoter and that AcrIF24 really represses transcription from this promoter. In the absence of these data, the authors should clearly speak of a “putative” promoter and clearly acknowledge the uncertainty of their conclusions.

Our responses:

The reviewer raised a very good point. Following the reviewer’s suggestion, we have set up an *in vivo* transcriptional assay using β -gal fusion system. The results showed that the sequence previously identified is the Acr promoter and that AcrIF24 but not its HTH-motif mutant represses transcription from this promoter (revised Fig. 2f). We have also revised the related text accordingly based on the verification of the promoter region and the Aca function of AcrIF24.

The description of the interaction between AcrIF24 and the Csy complex is very detailed with respect to describing individual residue-residue interactions. This is more detailed than necessary for most journal readers. However, the authors leave out some information that would be of broader interest. How large are the interfaces with the Csy complex formed by the NTD and the MD. Are these domains binding to the same regions bound by other I-F Acrs? This is a particularly important point. I would like to know whether AcrIF24 is interacting with the Csy complex in a unique manner.

Our responses:

The reviewer raised a very good point. Following the reviewer’s suggestion, we have simplified the description of the detailed individual residue-residue interactions and included the information of the interface between Csy and the NTD and MD of AcrIF24, such as the interface area and its comparison with the bound regions of other I-F Acrs. The results showed that AcrIF24 is interacting with the Csy complex in a unique manner (revised Fig. 4a and Supplementary Figs. 6a-f) compared to Acrs with known binding mechanisms.

Fig. 1a. The authors should explain their *in vitro* cleavage assay. Why is there a single cleavage product resulting from Cas3-mediated cleavage? How is that product produced? The effect of AcrIF24 in the cleavage assay is not very dramatic. What was the percent reduction in cleavage activity? The authors include no discussion of assay reproducibility.

Our responses:

The reviewer raised a very good point. It is known that during the Cascade-mediated DNA target degradation by Cas3, Cas3 first nicks the non-target DNA strand at the PAM distal end through the nuclease domain, and then the combined actions of the helicase and nuclease domains of Cas3

progressively degrade the DNA^{2,3}. The target DNA used in our cleavage assay contains a 5' FAM-label in the non-target strand. Fig. 1a showed that the size of the main cleavage product in our study was below 10 nt. Therefore, this suggested that the non-target DNA strand is first nicked at a PAM distal site and then progressively degraded to a site less than 10 nt from the PAM proximal end. Notably, a defined single cleavage product in the *in vitro* DNA cleavage assay with type I-F CRISPR-Cas system was also observed in other studies^{3,4}, and Rollins et al. also showed that a small fragment of DNA of around 6 nt is the major cleavage product of type I-F CRISPR-Cas system after enough reaction time³.

For the percent reduction in cleavage activity, we have repeated the experiment and provided both quantitative and statistical analysis of the DNA cleavage results (revised Fig. 1a and Supplementary Fig. 1). Moreover, we have included discussion of assay reproducibility into the legends of this figure and other related figures.

Fig. 4f: The authors should discuss these data. Why does the highest concentration of F24 show shifted bands and 2 different mobilities?

Our responses:

The reviewer raised a very good point. Fig. 4f showed that the Csy-AcrIF24 complex migrates as two shifted bands with different mobilities. With the addition of lowest concentration AcrIF24, the higher band of Csy-AcrIF24 complex appears (lane 3 of the left panel of Fig. 4f). As the AcrIF24 concentration increases, the band corresponding to the apo Csy decreases and the higher band of Csy-AcrIF24 increases (lane 4 of the left panel of Fig. 4f). With the highest concentration of F24, all the apo Csy is occupied by AcrIF24 and a lower band of Csy-AcrIF24 complex appears. The original Fig. 3b could help explain this result. As stated in the figure legend of Fig. 4f, the concentration of Csy complex was 0.32 μ M and AcrIF24 was added with concentrations of 0.16, 0.32, and 0.64 μ M. That is, under the lowest concentration of AcrIF24, only Csy(2)-AcrIF24(2) complex forms with the excess of Csy. Under the highest concentration, however, Csy(1)-AcrIF24(2) complex starts to form when Csy is completely bound. Therefore, we speculated that the higher and lower shifted band of Csy-AcrIF24 represents the Csy(2)-AcrIF24(2) and Csy(1)-AcrIF24(2) form, respectively. This was confirmed by the co-expressed and purified Csy(2)-AcrIF24(2) complex (Response Fig. 3), whose molecular weight has been confirmed by the SLS assay (Supplementary Fig. 4b in the revised manuscript). We also performed another AcrIF24-Csy binding assay with higher concentrations of AcrIF24 to confirm our speculation (Response figure 3), which showed that the amount of Csy(1)-AcrIF24(2) complex increases as the concentration of AcrIF24 increases. In addition, we have updated the labels in the original Fig. 4f and Supplementary Fig. 8a to make it easier for readers to understand.

Response figure 3. Native gel used to test the binding between the Csy complex and AcrIF24

Reactions were performed with 0.32 μM Csy complex and AcrIF24 concentrations of 0.16, 0.32, 0.64, 1.28, 2.56, and 5.18 μM following the order indicated by the black triangle. ‘(+)’ indicated the concentration of AcrIF24 was 5.18 μM . AcrIF24(2): Csy(2) was the complex obtained from co-expression. The gel was stained with Coomassie blue staining.

My overall impression of this paper is that the data presented will be of interest to those with interest in CRISPR-Cas systems and how they are inhibited. However, many papers have now been published showing structures of Acrs bound to CRISPR-Cas complexes. The nsDNA-binding story is not fleshed out enough to have any meaning. Thus, I do not feel that this paper warrants publication in Nature Communications.

Our responses:

During the revision, we have completely addressed the comments by all the reviewers with a large number of additional experiments and revisions to the manuscript, which have significantly improved our manuscript. Particularly, the results of the *in vivo* assays suggested by the reviewer also greatly improved the biological significance of our findings. While many papers have now been published showing structures of Acrs bound to CRISPR-Cas complexes, AcrIF24 is very special in its “multi-functional” characteristics. First, AcrIF24 forms a homodimer, interacts with the CRISPR complex and further induces its dimerization. Second, AcrIF24 contains an HTH-motif in its CTD and functions as a transcriptional repressor. Third, as stated above, AcrIF24 interacts with the Csy complex in a unique manner compared to known Acrs and prevents DNA-RNA hybridization. Last but not least, binding of AcrIF24 further induces non-specific DNA binding, which contributes to its inhibition capacity. Each of the above four strategies has been revealed for specific Acrs, however, there is never an Acr adopting all of the four strategies as AcrIF24 does. Therefore, AcrIF24 is an excellent example of a multi-functional Acr with three layers of CRISPR-Cas suppression, i.e., inactivating two Csy complexes at a time, preventing DNA-RNA hybridization and inducing non-specific DNA binding. Taken together, AcrIF24 is a very special multi-functional Acr and our manuscript has also been significantly improved during the revision, and we hope that now it is suitable for publication in Nature Communications.

Again, we wish to thank the reviewer for her/his constructive comments and suggestions, which have helped us to improve our manuscript greatly.

Reviewer #2 (Remarks to the Author):

The authors performed the structure-function analysis of the anti-CRISPR protein AcrIF24, combining complementary structural and biochemical approaches. They showed that AcrIF24 is the first example of a multifunctional anti-CRISPR protein targeting the class 1 CRISPR-Cas system. First, AcrIF24 forms homodimers in which the C-terminal domains harboring HTH motifs bind to the *acrIF23-24* promoter, and thereby functions as a transcriptional repressor. Second, one AcrIF24 homodimer can bind and inactivate two Csy complexes making it a potent CRISPR-Cas inhibitor. Third, the AcrIF24 N-terminal domain interacts with several Cas7f subunits of the Csy complex and by doing so prevents DNA-RNA hybridization. Lastly, the AcrIF24 C-terminal domain within the Csy-AcrIF24 complex is important to stabilize an unproductive DNA-bound state whether it is a specific or non-specific DNA. Although one or the other of these characteristics have already been described for various anti-CRISPR proteins (the type II AcrIIA1 and AcrIIA13-15 anti-CRISPR proteins function as transcriptional repressor; the type II/V AcrIIA6, AcrIIC3 and AcrVA4 induce Cas9/Cas12 dimerization; the type I AcrIF9 tethers non-specific DNA to the Csy complex), it is remarkable that AcrIF24 combines all these functionalities. This work expands our knowledge of anti-CRISPR proteins and is therefore significant to the field.

Overall the text and the figures are clear, with the exception of some panels showing 3D structures (see major point 5). The results are validated by different experimental approaches and appropriate controls, and they are presented and interpreted in the context of previous work.

We appreciate the reviewer's high evaluation of our work and constructive suggestions. We have addressed all of the comments listed below with additional experiments and have revised the manuscript accordingly, which we believe have significantly improved our manuscript.

Major points

1) I have found the sections entitled "The Csy-AcrIF24 complex interact with dsDNASP and dsDNANS through different mechanisms" (lines 383-438) and "AcrIF24 and AcrIF9 show different characteristics in inducing non-specific DNA binding" (lines 440-474) very confusing. In particular, I have some concerns regarding data interpretation and conclusions.

- In the section "The Csy-AcrIF24 complex interact with dsDNASP and dsDNANS through different mechanisms" (lines 383-438), what the EMSA experiments show, using dsDNASP and dsDNANS of different lengths and positions of fluorescein label, and wild-type or mutants of Cas8f and AcrIF24, is that the Cas8f PAM-binding site and AcrIF24 C-terminal domains are both involved in binding dsDNASP and dsDNANS. Therefore, the Csy-AcrIF24 complex interacts with dsDNASP and dsDNANS using similar binding sites and mechanisms. The differences observed in the formation of Csy-AcrIF24-dsDNA complexes in the presence of dsDNASP-15bp, or of dsDNANS with length less than 35bp, are explained by the fact that the affinity of dsDNANS for the Csy complex being necessarily lower than that of dsDNASP, stable complex between dsDNANS and the Csy complex cannot be formed. It is only when the length of dsDNANS is long enough (35 bp) to bind to the second DNA-binding sites, that the complex Csy-AcrIF24-dsDNANS can form.

Our responses:

The reviewer raised a very good point. We appreciate the interpretation of our data by the reviewer and agree that our previous conclusion is not accurate. We have revised this part following the reviewer's suggestion. Firstly, we agree that Csy-AcrIF24 complex interacts with dsDNA_{SP} and dsDNA_{NS} using similar binding sites and mechanisms based on our previous data shown in this section. Therefore, we mainly revised the conclusion texts by changing the previous conclusion to the above one, including that the title of this section was also revised to "The Csy-AcrIF24 complex interacts with dsDNA_{SP} and dsDNA_{NS} through similar mechanisms". Secondly, as the reviewer pointed out below, it is less important to determine the minimal length of DNA tethered to the Acr-bound Csy complex. Therefore, we have simplified the content related to describing the binding results of various lengths of dsDNA (lines 399-438 in the original manuscript) into a briefer presentation. For example, we have deleted the content about discussing the negligible binding of 25-bp dsDNA_{SP} (lines 422-429 in the original manuscript) and the previous conclusion part about discussing the minimal length of dsDNA_{NS} for binding (lines 429-438 in the original manuscript). Thirdly, we have also moved the description in lines 402-409 (in original manuscript, now lines 394-400) to line 399 (in original manuscript, now line 394) for a better logic of this section. In all, we have revised this section by highlighting the similar binding mechanisms of dsDNA_{SP} and dsDNA_{NS} and simplifying the description of binding results of various lengths of DNA.

- In the section "AcrIF24 and AcrIF9 show different characteristics in inducing non-specific DNA binding" (lines 440-474), the authors begin by describing EMSA results (lines 444-450) without providing any interpretation on the mechanisms used by AcrIF24 and AcrIF9 to induce non-specific DNA binding. Second, the EMSA results obtained with the Cas8f and AcrIF9 mutants can be explained by the same effect as mentioned in my previous comment. The affinity of the dsDNA_{SP} for the PAM-binding site is high enough to form stable Csy-AcrIF9-dsDNA_{SP} complex in which the AcrIF9 DNA-binding residues are mutated. However, the lower affinity of dsDNA_{NS} for the PAM-binding site prevents its binding to the Csy-AcrIF9 complex in which the AcrIF9 DNA binding residues are mutated. Therefore, the sentence "The above mutational results indicated that the K-wedge of Cas8f is involved in the binding of both dsDNA_{SP} and dsDNA_{NS}, while R32 and K36 of AcrIF9 are mainly involved in the binding of dsDNA_{NS} of 15/25bp" (lines 469-471) is not correct. Moreover, it is hard to conceive how the AcrIF9 DNA-binding site could discriminate between specific and non-specific target DNA.

Our responses:

Same as above, we appreciate the interpretation of our data by the reviewer and agree that our previous interpretation is not correct. We have revised this part following the reviewer's suggestion. Firstly, we agree with the notion that AcrIF24 and AcrIF9 use a similar strategy, but different mechanisms based on our previous data shown in this section. Therefore, we mainly revised the conclusion text of the original manuscript to illustrate the above notion, including that the title of this section was also revised to "AcrIF24 and AcrIF9 use a similar strategy, but different mechanisms in inducing non-specific DNA binding". Secondly, as the reviewer suggested, a key point linking the results of this section and the above section is the lower affinity of dsDNA_{NS} for the Csy complex compared with dsDNA_{SP}. We have also included this point into the conclusion text of this section (lines 449-451 in the revised manuscript). Thirdly, we have also deleted the content about discussing DNA

lengths required for binding (lines 456-457 of the original manuscript) and the original conclusion part about comparing the characteristics of AcrIF24- and AcrIF9-induced non-specific DNA binding (lines 469-474 of the original manuscript). In all, we have revised this section by highlighting the similar strategy but different mechanisms of AcrIF24- and AcrIF9-induced non-specific DNA binding and also simplifying the description of binding results of various lengths of DNA.

- In both sections, the authors focus on determining the minimal length of DNA tethered to the Acr-bound Csy complex. But is this relevant to the AcrIF24 mode of action? Indeed, as proposed by the authors in the discussion (lines 505-516), the AcrIF24-Csy complex scans foreign DNA and can either find and bind to its specific PAM and target sequence, or can bind to non-specific DNA while scanning the whole DNA molecule. In this context, the minimal length of DNA that can bind to the AcrIF24-Csy complex is meaningless.

Our responses:

The reviewer raised a very good point. It is less important to determine the minimal length of DNA tethered to the Acr-bound Csy complex. We have reduced and simplified the related content in the revised manuscript.

All in all, the results presented in these sections show that AcrIF24 and AcrIF9 use a similar strategy to induce non-specific DNA-binding to the Csy complex, which consists in providing DNA-binding sites that, together with the PAM-binding site, stabilize non-productive DNA-bound states. However, the molecular mechanisms used by these anti-CRISPR proteins are different: the AcrIF24 dimer provides a DNA-binding domain rather distant from the PAM-binding site, while AcrIF9 provides two DNA-binding sites one being close to the PAM-binding site and the other being more distant.

Our responses:

Again, we appreciate the interpretation of our data by the reviewer and have revised the related parts accordingly.

2) The last sentence of the discussion “In all, our study reveals that AcrIF24 is an Acr with two layers of suppression, highlight the prevalence of induction of non-specific DNA binding, and provides striking example of convergent evolution of Acrs” (lines 516-518) should be amended. Since AcrIF24 can inactivate two Csy complexes at a time, I suggest to mention three “layers of suppression”. Also, to my knowledge, there are currently three examples of Acr inducing non-specific DNA binding including AcrIIA11, AcrIF9 and AcrIF24, therefore “prevalence” (also written in the abstract, line 42) is not really appropriate.

Our responses:

The reviewer raised a very good point. We have changed the former sentence into “In all, our study reveals that AcrIF24 is an Acr with **three layers of suppression (i.e., inactivate two Csy complexes at a time, prevention of DNA-crRNA hybridization and induction of non-specific DNA binding), highlight the potential universality of induction of non-specific DNA binding, and provides a striking example of convergent evolution of Acrs” (now lines 511-515). Moreover, “prevalence” in the abstract has also been revised to “potential universality”.**

3) The authors have co-expressed AcrIF24 and the Csy complex, and found that they form homotetrameric

assemblies. Since AcrIF24 induces non-specific DNA binding to the Csy complex, I was expecting to read that the co-expressed Csy-AcrIF24 complex co-purified DNA as it is highly often the case with DNA-binding proteins. If so, high salt washes are usually enough to get rid of non-specific DNA in protein preparation. Moreover, it is not clear in the text when and why they used the co-expressed Csy-AcrIF24 complex. For cryoEM experiments, did they use this assembly or did they mix the different partners (lines 199-201, 309-311)? Why did they use the co-expressed assembly for titration experiments (lines 290-293)? In this case, it is crucial to confirm that the co-expressed assembly is not purified with DNA.

Our responses:

The reviewer raised a very good point. The co-expressed Csy-AcrIF24 was purified successively through Ni-affinity, anion exchange and gel filtration chromatography (Response Fig. 4). To investigate whether the finally obtained co-expressed Csy-AcrIF24 complex contains nucleic acids, the protein was subjected to heparin column chromatography to separate protein and possibly co-purified nucleic acids. As shown in Response Fig. 5, the Csy-AcrIF24 co-purified with RNA, which could not be separated by the heparin column. As a control, Cas9 from *Streptococcus pyogenes* (SpyCas9) purified after Ni-column also co-purified with RNA, which could be separated by the heparin column. Therefore, the RNA in the Csy-AcrIF24 complex should be the crRNA within the Csy complex and no non-specific nucleic acids was co-purified with the sample used in the study. As the reviewer suggested, the non-specific nucleic acids co-purified with Csy-AcrIF24 complex might be washed off by the high salt buffer during Ni-column and anion-exchange chromatography (Response Fig. 4). Therefore, we confirmed that the co-expressed and purified Csy-AcrIF24 assembly does not contain non-specific nucleic acids.

Response figure 4. Scheme of the purification process of co-expressed Csy-AcrIF24

The co-expressed Csy-AcrIF24 was purified through Ni-affinity, anion exchange and gel filtration chromatography, successively.

Response figure 5. Comparison between purified Csy-AcrIF24 and SpyCas9

(a and b) The co-expressed Csy-AcrIF24 purified with Ni-affinity chromatography, ion exchange chromatography, and size exclusion chromatography (a) and SpyCas9 purified with Ni-affinity chromatography (b) were subjected to heparin column to wash out possible non-specific binding nucleic acids. (c and d) Injection samples (labeled as input), A and B peaks (labeled as A and B, respectively) from the elution of the heparin column were analyzed with SDS-PAGE (c) and agarose gel (d). In panel d, peak B of Csy-AcrIF24 (a) and peak A of SpyCas9 (b) were treated with DNase and RNase for 30 min at 37°C before electrophoresis, respectively.

According to the reviewer's suggestion, we have checked the manuscript and indicated clearly the approach we obtained the Csy-AcrIF24 complex. Actually, all the Csy-AcrIF24 complex used in this study was obtained through co-expression. Co-expressed Csy-AcrIF24 was used for cryoEM experiments of both Csy-AcrIF24 and Csy-AcrIF24-dsDNA_{SP}. We have indicated this clearly in the corresponding texts. For the titration experiments (lines 290-293), we intended to compare the DNA binding affinity between Csy and Csy-AcrIF24. Therefore, we had to confirm pure Csy(2)-AcrIF24(2) complex is used without any free Csy or AcrIF24. Co-expression is the simplest approach to obtain this complex, and therefore all the Csy-AcrIF24 complex used in our study was obtained through co-expression.

4) Figure 3A.

The concentrations of AcrIF24 in solution are not appropriate, as shown by the poor fitting at high concentrations. If the KD is around 4 nM (as indicated in line 186), concentrations should instead be between 0.4 and 40 nM.

Our responses:

According to the reviewer's suggestion, we have re-performed the SPR experiment using AcrIF24 concentrations of 1.25, 2.5, 5, 10, 20, 40 and 80 nM, respectively.

5) Figures 3D, 3E, 4A, 4B, 5A, 5B, 9D, S5A.

It is hard to see anything in these views. Increasing their size and/or changing colors to improve the contrast could help.

Our responses:

According to the reviewer's suggestion, we have revised these figures by increasing their size and/or changing colors.

6) lines 134-139: "The NTD comprises a five-stranded antiparallel β sheet and a bent α helix. A Dali search with this domain revealed that the closest structural homolog is an auxiliary protein of soluble methane monooxygenase hydroxylase (MMOH), MMOD (Fig. 1e)^{22,23}. Interestingly, MMOD is an inhibitor of MMOH through associating with the canyon region of MMOH and inducing conformational changes of it." I do not understand what is the relevance of this comparison to decipher the mechanism of AcrIF24. Figure 1E should go in supplementary.

Our responses:

We think that a similar structure may have functional relevance. For example, the fact that MMOD is employed to engage with other proteins indicates that this domain can work as a protein-protein interaction module. According to the reviewer's suggestion, we have moved Figure 1E to the supplementary figures as Supplementary Fig. 2e.

Minor points

1) Results of DALI search should be provided with the corresponding Z-score, rmsd, and number of aligned residues to judge on the significance of structural similarity (lines 135-137, 140-141).

Our responses:

Following the reviewer's suggestion, we have provided the Z-score, RMSD and number of aligned residues into the DALI search results as Supplementary Fig. 2f and 2g.

2) lines 204, 210, 212, 313.

Are they RMSD on the backbone only or on the whole model?

Our responses:

They are RMSD on the backbone and we have included this information into the revised manuscript.

3) line 233 "sidechain nitrogen atom of Cas7.6fL76". Main chain instead of side chain

Our responses:

We have corrected this in the revised manuscript.

4) lines 484-486 "AcrIIA6 and AcrVA4 also harbor HTH domain and induces the dimerization of Cas9 from

Streptococcus thermophilus". AcrVA4 induces the dimerization of LbCas12

Our responses:

We thank the reviewer for pointing this out and have corrected this in the revised manuscript.

5) line 646 "AcrIF24ΔMD (15 mg/mL)". 1.5 mg/mL instead of 15 mg/mL ?

Our responses:

We indeed used 15 mg/mL AcrIF24ΔMD in the SLS assay, under which concentration we could get data with good signal to noise ratio for AcrIF24ΔMD.

6) Figure 8A. What is the color code?

Our responses:

We thank the reviewer for pointing this out and have added the color code in the figure legend.

Again, we wish to thank the reviewer for her/his constructive comments and suggestions, which have helped us to improve our manuscript greatly.

Reviewer #3 (Remarks to the Author):

Yang et al. present an X-ray structure of AcrIF24ΔMD, and cryo-EM structures of AcrIF24-Csy and AcrIF24-Csy-dsDNASP complex. AcrIF24 has not been characterized structurally or mechanistically. Using a combination of structural and biochemical analyses, the authors showed binding of AcrIF24 to Csy complex by interacting with Cas7 subunits block the hybridization of DNA to the crRNA, similar to AcrIF9. Remarkably, binding of AcrIF24 also induces the non-specific DNA binding to Csy complex, in which the PAM recognition motif and HTH motif plays an essential role.

The AcrIF24-Csy complex is interesting given that AcrIF24 contains a HTH motif that might be involved in the self-regulation. The authors give the first structural description of the HTH motif containing anti-CRISPR protein in complex with Csy complex. It would be of great interest that the authors give a discussion on how this dual anti-CRISPR protein works in both regulation of its own promoter and inhibition of target DNA cleavage.

We appreciate the reviewer's high evaluation of our work and constructive suggestions. We have addressed all of the comments listed below with additional experiments and have revised the manuscript accordingly, which we believe have significantly improved our manuscript. Moreover, we have included further discussion on how this dual anti-CRISPR protein works.

Major points:

1) As shown in Csy-AcrIF24 complex (Supplementary Fig. 5b and 5c), AcrIF24 engages the target DNA binding site in the Csy complex to block hybridization of DNA to the crRNA. However, as judged by EMSA in Fig. 5a compared with Fig. 5c, it seems Csy-AcrIF24 complex still specifically binds to dsDNASP.

Similarly, as shown in Fig. 9b, Csy-AcrIF9 still binds to ssDNA, what's the binding mode for specific target dsDNA and ssDNA binding of Csy-AcrIF9?

Our responses:

We believe that the Csy-AcrIF24 complex does not specifically binds to dsDNA_{SP}, but non-specifically. As the reviewer pointed out, the original Supplementary Fig. 5b and 5c clearly showed that AcrIF24 block hybridization of DNA_{SP} to the crRNA. However, EMSA experiments showed that the Csy-AcrIF24 complex still binds both dsDNA_{SP} and dsDNA_{NS} (Fig. 5a, c and d in the original manuscript). Taken together, these results indicated that Csy-AcrIF24 binds these DNA molecules non-specifically. AcrIF24 contains an HTH motif in its C-terminus, which has weak non-specific DNA-binding activity itself (Supplementary Fig. 1e in the original manuscript). On the other hand, the Csy complex also contains weak non-specific DNA binding activity to sample dsDNA for PAM sequence recognition⁵. Regarding the binding mode of dsDNA_{SP} by Csy-AcrIF24, our Csy-AcrIF24-dsDNA_{SP} structure has shown that the PAM recognition loop is involved in dsDNA_{SP} binding (Fig. 6d in the original manuscript), which was also confirmed by mutagenesis studies (Fig. 7a in the original manuscript). Moreover, EMSA with AcrIF24 mutants also showed that R196 and K197 within the HTH motif of AcrIF24 are also involved in dsDNA_{SP} binding (Fig. 7b in the original manuscript). Therefore, both the PAM recognition loop of Csy and the HTH motif of AcrIF24, but not hybridization of DNA-crRNA, mediate dsDNA_{SP} binding.

For the ssDNA binding by Csy-AcrIF9 in Fig. 9b, similarly, AcrIF9 also sterically prevents hybridization of DNA to the crRNA and Csy-AcrIF9 also harbors non-specific DNA binding activity^{6,7}. Lu et al. have performed extensive experiments on the non-specific DNA binding activity of Csy-AcrIF9⁶. Their study showed that the R-loop binding channel of the Csy complex and a conserved positively charged surface on AcrIF9 are involved in non-specific DNA binding of Csy-AcrIF9⁶. Although mutagenesis studies have not been performed on the ssDNA binding of Csy-AcrIF9, ssDNA binding should also be mediated by both the Csy-complex and AcrIF9 like dsDNA, but not through ssDNA-crRNA hybridization. Overall, the binding between target ds/ssDNA and Csy-AcrIF9 result from the coaction of the R-loop binding channel of Csy and the conserved positively charged surface on AcrIF9, instead of target DNA-crRNA hybridization.

The DNA bound bands of right panel in Fig.5c are different from those in left panel shown in Fig. 5a, but both figures show the dsDNA_{SP} binding of Csy-AcrIF24, which is difficult to explain.

Our responses:

Different samples were used in these two figures. In the right panel of Fig. 5c, the purified Csy-AcrIF24 heterotetramer complex was used. Therefore, the DNA-bound bands represent the Csy-AcrIF24-dsDNA_{SP} complex in which the dsDNA_{SP} is bound non-specifically to the Csy-AcrIF24 complex, as stated in the last response. In the left panel of Fig. 5a, a mix of Csy and AcrIF24 was used, in which AcrIF24 was added with an increasing concentration. As mentioned in the figure legend, the concentration of Csy is 8 μ M, and concentrations of AcrIF24 were set as 1, 2, 4 and 8 μ M, respectively. Therefore, under lower concentrations of AcrIF24, there was a large amount of free Csy, and therefore Csy-dsDNA_{SP} was the major product (the lower DNA-bound band as indicated by lane 2). With the increasing concentration of AcrIF24, the amount of Csy-AcrIF24-dsDNA_{SP} complex (the higher DNA-

bound band) increases, as the concentration of formed Csy-AcrIF24 complex increases. This is why different patterns are formed between the two panels.

2) In Fig. 6d, the specific target DNA is split upon binding to Csy-AcrIF24 (Fig. 6c and 6d), but the model shown in Fig. 7e indicates it binds to Cas8f and AcrIF24 without being split, which is also difficult to explain. The structural data (Fig. 6c and 6d) is not supportive to the proposed model in Fig. 7e. Maybe it would be useful to get the complex structure of Csy-AcrIF24-dsDNA_{NS} for investigating the AcrIF24-mediated inhibition mechanisms by non-specific DNA binding.

Our responses:

The reviewer raised a very good point. For Fig. 6c and 6d, the target DNA is split at the A¹(TS)-T¹(NTS) position, but no density is observed for the following sequence of the protospacer. In light of the prevention of target DNA-crRNA hybridization by AcrIF24 on the Csy, we think that the specific target DNA should be only partially split around this position. In Fig. 7e, we were proposing a general model of the working mechanism of AcrIF24 on the inhibition of type I-F CRISPR-Cas system. Therefore, the DNA molecule in the model represents non-sequence-specific DNA here. As the reviewer suggested, it would be better to get the structure of the complex of Csy-AcrIF24-dsDNA_{NS} for investigating the AcrIF24-mediated inhibition mechanisms by non-specific DNA binding. Nevertheless, while we did not solve the complex structure of Csy-AcrIF24-dsDNA_{NS}, we think that the structure of Csy-AcrIF24-dsDNA_{SP} structure could generally explain the binding mode of non-specific DNA by the Csy-AcrIF24 complex. On the one hand, the mutational studies showed that the binding of dsDNA_{SP} and dsDNA_{NS} are both markedly decreased or abolished by the mutations of the PAM interaction site of Csy (Fig. 7a in the original manuscript) and the R196 and K197 mutations of the AcrIF24-HTH domain (Fig. 7b in the original manuscript). On the other hand, the competition EMSA with dsDNA_{SP} and dsDNA_{NS} also suggested that they bind the similar sites of Csy-AcrIF24 (Fig. 5e in the original manuscript). That is, for the binding of non-specific DNA by Csy-AcrIF24, the PAM interaction site of Csy and the HTH domain of AcrIF24 are also involved. However, since the dsDNA_{NS} lacks the PAM region and protospacer sequence, its double-strand cannot be split and its detailed interactions with the PAM interaction site of Csy-AcrIF24 cannot be exactly the same as that of dsDNA_{SP}. Therefore, as a working model, we mainly show the involvement of both the PAM interaction site of Csy and the HTH domain of AcrIF24 in non-specific DNA binding of Csy-AcrIF24. Future studies on the complex structure of Csy-AcrIF24-dsDNA_{NS} will further optimize this model. We have included these discussions into the revised manuscript.

3) Density of residue K247 should be shown in Fig. 6d to indicate the interaction mode between K247 and PAM sequence in target DNA, given previous studies showed K247 is crucial for PAM recognition. It's reasonable that mutation of K247 abolished the binding ability of Csy-AcrIF24 to DNASP, but it's difficult to explain that mutation of K247 also abolished the DNANS binding ability.

Our responses:

The reviewer raised a very good point. We have shown the density of K247 in Fig. 6d. We think the following two points may help explain the important role of K247 in the DNANS binding. First, as revealed by the result of the DNA competition assay (original Fig. 5e), dsDNA_{SP} and dsDNA_{NS} possess the same binding site on the Csy-AcrIF24 complex. Therefore, in light of the role of the K-wedge of

Cas8f subunit in dsDNA_{SP} interaction, it may also mediate dsDNA_{NS} binding. Second, it has been proposed that the Csy complex uses a “PAM scan” method for target location with its K-wedge of Cas8f subunit. That is, the K-wedge should possess some weak binding ability for non-specific DNA. As we suggested in the discussion section, we believe that AcrIF24 may facilitate the stable binding of the dsDNA_{NS} after it is non-specifically associated by the PAM recognition loop of the Csy-AcrIF24 complex.

4) The WT controls in several figures are missing, such as Fig. 2e, 2f, 5b, 7a, 7c, 8d and 9e.

Our responses:

For Fig. 2e, its WT control is the left six lanes of Fig. 2c. To be clearer, we have split the gel of Fig. 2c into left and right, two panels. For Fig. 2f, we have added a WT control in this panel in the revised manuscript. For Fig. 5b, the WT controls of the experiments using dsDNA_{SP} and dsDNA_{NS} are the left and right panel of Fig. 5a, respectively. For Fig. 7a and 7c, the WT control of the experiments using dsDNA_{SP} and dsDNA_{NS} are the right panels of Fig. 5c and 5d, respectively. To be clearer, we have repeated the experiment and split the gel of Fig. 5c and 5d each to two gels, in which WT control gels are still on the right. For Fig. 8d, the WT control of the experiments using dsDNA_{NS} 35 bp and 45 bp are the right two panels of Fig. 8b, respectively. For Fig. 9e, the WT control of the experiments using 3' FAM dsDNA_{SP} 15 bp and 25 bp are the left two panels of Fig. 8c, respectively. The WT control of the experiments using 5' FAM dsDNA_{SP} 15 bp and 25 bp in Fig. 9e are the left two panels of Fig. 8b, respectively. To facilitate reading, we have added the information of the panels of WT controls in the figure legends of the above panels. For example, figure legends in Fig. 2e: Please see the left panel of Fig. 2c for comparison with WT control.

Minor points:

1. It would be useful to draw a schematic of AcrIF24 in Fig. 1b.

Our responses:

The reviewer raised a very good point. We have added a schematic of AcrIF24 in Fig. 1b in the revised manuscript.

2. It's hard to see the grey color in Fig. 1e and 1f, the author should change it into a different color.

Our responses:

We have changed the grey color in Fig. 1e and 1f to marine in the revised manuscript.

3. As shown in left panel of Fig. 2c, it's interesting that there are two DNA bound bands observed for IR23-24 but just one band for mutated IR23-24. Does the author have some explanation?

Our responses:

The reviewer raised a very good point. It has been reported that proteins that bind to nucleic acids with a sequence-specific manner may also possess weak affinity for non-specific DNA. We speculated that the lower band of bound DNA represents sequence-specific DNA binding, while the upper band

with the same size in both WT and mutated IR23-24 groups represent non-sequence-specific DNA binding. For IR23-24, only sequence-specific DNA binding occurs under low concentrations of AcrIF24, and non-sequence-specific DNA binding appears only with extremely high concentration of AcrIF24. For mutated IR23-24, however, only non-sequence-specific DNA binding was observed with high protein concentration. Similar non-specific binding band in the EMSA was also observed in other studies (Fig. 1A and B in the reference paper)⁸. Moreover, we also tested the binding between target DNA and the Csy complex with higher concentrations and similar non-specific DNA binding band was also observed (Response figure 6). We have adjusted the label in Fig. 2c and other relevant figures to make it easier for readers to understand.

Response figure 6. EMSA used to test the binding affinities between Csy complex and its target dsDNA
 Reactions were performed with 6.25 nM of 54 bp-dsDNA (5'-FAM in the TS) and a concentration gradient (0, 0.003125, 0.03125, 0.15625, 0.3125, 1.5625, 3.125, 31.25, 312.5, 3125, 31250 nM) of Csy complex following the order indicated by the black triangle.

4. It would be helpful to color the domains of AcrIF24 differently in Fig. 4a-4e.

Our responses:

According to the reviewer's suggestion, we have colored the domains of AcrIF24 differently in Fig. 4a-4e.

5. The densities of residues in Fig4b-4e should be shown.

Our responses:

According to the reviewer's suggestion, we have shown the densities of the residues in Fig4b-4e.

6. Local resolution map of the cryoEM structures including bound DNA will be useful for evaluation of these structures.

Our responses:

According to the reviewer's suggestion, we have included the local resolution maps of the cryoEM structures as Supplementary Fig. 7b in the revised manuscript.

7. It would help if the author provides the distance between the K-wedge of Cas8f and the non-specific DNA binding motif of AcrIF9, and the length of bound DNA, specifically when the author shows the EMSA results with different DNA lengths.

Our responses:

The reviewer raised a very good point. We have provided the distance and length of bound DNA in the revised Fig. 9d.

8. In supplementary Fig. 5a, the coloring of Acrs should be rearranged, now it's unclear.

Our responses:

According to the reviewer's suggestion, we have rearranged the coloring of Acrs in supplementary Fig. 5a (now supplementary Fig. 6f) in the revised manuscript. Moreover, to make it clearer, we have also included another five panels showing the binding of Acrs separately (revised supplementary Figs. 6a-e).

Again, we wish to thank the reviewer for her/his constructive comments and suggestions, which have helped us to improve our manuscript greatly.

References

1. Buyukyoruk, M. & Wiedenheft, B. Type I-F CRISPR-Cas provides protection from DNA, but not RNA phages. *Cell Discov* **5**, 54 (2019).
2. Mulepati, S. & Bailey, S. In vitro reconstitution of an Escherichia coli RNA-guided immune system reveals unidirectional, ATP-dependent degradation of DNA target. *J Biol Chem* **288**, 22184-92 (2013).
3. Rollins, M.F. et al. Cas1 and the Csy complex are opposing regulators of Cas2/3 nuclease activity. *Proc Natl Acad Sci U S A* **114**, E5113-E5121 (2017).
4. Zhang, K. et al. Inhibition mechanisms of AcrF9, AcrF8, and AcrF6 against type I-F CRISPR-Cas complex revealed by cryo-EM. *Proc Natl Acad Sci U S A* **117**, 7176-7182 (2020).
5. Rollins, M.F., Schuman, J.T., Paulus, K., Bukhari, H.S. & Wiedenheft, B. Mechanism of foreign DNA recognition by a CRISPR RNA-guided surveillance complex from *Pseudomonas aeruginosa*. *Nucleic Acids Res* **43**, 2216-22 (2015).
6. Lu, W.T., Trost, C.N., Muller-Esparza, H., Randau, L. & Davidson, A.R. Anti-CRISPR AcrIF9 functions by inducing the CRISPR-Cas complex to bind DNA non-specifically. *Nucleic Acids Res* **49**, 3381-3393 (2021).
7. Hirschi, M. et al. AcrIF9 tethers non-sequence specific dsDNA to the CRISPR RNA-guided surveillance complex. *Nat Commun* **11**, 2730 (2020).
8. Taylor, J.A. et al. Specific and non-specific interactions of ParB with DNA: implications for chromosome segregation. *Nucleic Acids Res* **43**, 719-31 (2015).

REVIEWER COMMENTS

Reviewer #1 (Remarks to the Author):

This revised manuscript has been improved substantially from the initial submission with the addition of useful experimental data and improved descriptions.

The main problem I have left with this work is in the interpretation of the in vivo phage plaquing data. It is good that the authors included these experiments to address the in vivo significance of non-specific binding. However, the authors need to mention important caveats to their results. As the authors show in their rebuttal, the I-F CRISPR system causes a 6-log reduction in the plaque forming ability of phage lambda. Co-expressing AcrIF24 seems to increase plaquing ability by at least 5-logs. In light of this, a 2 to 3-fold reduction in Acr activity resulting from amino acid substitutions is actually a very small reduction, even though it may be statistically significant. It is also very notable that the AcrIF24 mutants lacking the HTH domain or unable to dimerize (FYI/EEE) also show only 3-fold reductions in Acr activity. Considering that these mutants seemed unstable in vitro and expressed poorly, their relatively high activity compared to no Acr emphasizes to me that the HTH domain has little importance for Acr activity. So, while there is some effect on Acr activity from altering the HTH, my major conclusion from these data would be that the HTH domain is mostly or completely unnecessary for Acr activity in the assays shown. The authors must clearly explain the caveats inherent in the in vivo assays, primarily that an amino acid substitution affecting a key residue for Acr activity would reduce plaquing levels by many orders of magnitude, not just 2 to 3-fold. The authors should show data similar to what is shown in the rebuttal (i.e. the reader must know that activity could be reduced by many orders of magnitude) so that the reader can evaluate the importance of a 2 to 3 fold drop in activity. My laboratory performs these types of phage plaquing assays extensively and we consider a 10-fold drop in activity on the borderline of being worth mentioning.

It is important to note that in the publication on AcrIF9, 3 different in vivo assays were performed in an effort to demonstrate an in vivo role for the residues involved in non-specific DNA-binding. It was also shown that the residues involved in non-specific DNA-binding were conserved in an alignment of diverse AcrIF9 homologues. Amino acid substitutions at these positions only affected non-specific DNA-binding and did not affect binding of the Acr to the Csy complex. Non-specific DNA-binding was shown to be induced by two diverse AcrIF9 homologues acting on two different Csy complexes. Finally, AcrIF9 is a small single domain protein with no recognizable DNA-binding domain and no DNA-binding activity on its own. All of these facts led to the conclusion that the non-specific DNA-binding activity of AcrIF9 has biological importance. In comparison, the evidence for the biological importance of the non-specific DNA-binding activity of AcrIF24 is very weak. The authors need to take these points into careful consideration and very much soften their conclusions about non-specific DNA-binding activity.

Reviewer #2 (Remarks to the Author):

All my comments have been carefully addressed.

The authors have performed additional experiments and revised the text.

I recommend publication of the revised manuscript.

Reviewer #3 (Remarks to the Author):

Yang et al. present an X-ray structure of AcrIF24 Δ MD, and cryo-EM structures of AcrIF24-Csy and AcrIF24-Csy-dsDNASP complex. Using a combination of structural and biochemical analyses, the authors showed binding of AcrIF24 to Csy complex by interacting with Cas7 subunits block the hybridization of DNA to the crRNA, and binding of AcrIF24 also induces the non-specific DNA binding to Csy complex, in which the PAM recognition motif and HTH motif plays an essential role. During revision, the author has addressed lots of points, but some of my major points have not been addressed.

Major points 1:

During the revision, the author shows AcrIF24 repress its transcription by specifically interacting with IR23-24. In Fig. 2c, the HTH motif of AcrIF24 shows higher affinity with specific dsDNA sequence than non-specific dsDNA, and only binds non-specific dsDNA in a high concentration. However, in the following session, the author shows the HTH motif greatly contributes to the non-specific dsDNA binding. It's not supervised that HTH motif of AcrIF24 binds dsDNA non-specifically, but what's the affinity for this non-specific binding, most HTH motif could bind to dsDNA nonspecifically, is it meaningful in vivo in this case?

Major points 2:

Also, the published paper about AcrIF9 (PMID: 33660777) shows that Csy:AcrIF9 binds dsDNANS and dsDNASP at an overlapping site, but ssDNASP is bound differently, the Csy:AcrIF9 complex binds specifically to ssDNA, but not nonspecifically as the author describes.

The author mentioned that Csy-AcrIF24 binds ssDNA non-specifically, have the author tested of the ability of the non-specific ssDNA for competing with specific ssDNA?

Regarding to my first and second questions previously raised, I am not convinced. Could the author provide the affinity parameters of Csy-AcrIF9 towards dsDNASP, dsDNANT, ssDNASP and ssDNANT respectively? These parameters will help to understand the binding mode of Csy-AcrIF9 to the targets both in vitro and in vivo.

Major points 3:

In Fig. 6d, the specific target DNA is split upon binding to Csy-AcrIF24 (Fig. 6c and 6d), with missing DNA density between Csy and AcrIF24, there are not enough convincing evidence for supporting the proposed model. It would be useful to get the complex structure of Csy-AcrIF24-dsDNANS for investigating the AcrIF24-mediated inhibition mechanisms by non-specific DNA binding.

REVIEWER COMMENTS

Reviewer #1 (Remarks to the Author):

This revised manuscript has been improved substantially from the initial submission with the addition of useful experimental data and improved descriptions.

We appreciate the reviewer's high evaluation of the improvement of our manuscript during the revision. In addition, we have carefully addressed the comment below concerning the interpretation of the *in vivo* phage plaquing data, which we believe have further improved our manuscript.

The main problem I have left with this work is in the interpretation of the *in vivo* phage plaquing data. It is good that the authors included these experiments to address the *in vivo* significance of non-specific binding. However, the authors need to mention important caveats to their results. As the authors show in their rebuttal, the I-F CRISPR system causes a 6-log reduction in the plaque forming ability of phage lambda. Co-expressing AcrIF24 seems to increase plaquing ability by at least 5-logs. In light of this, a 2 to 3-fold reduction in Acr activity resulting from amino acid substitutions is actually a very small reduction, even though it may be statistically significant. It is also very notable that the AcrIF24 mutants lacking the HTH domain or unable to dimerize (FYI/EEE) also show only 3-fold reductions in Acr activity. Considering that these mutants seemed unstable *in vitro* and expressed poorly, their relatively high activity compared to no Acr emphasizes to me that the HTH domain has little importance for Acr activity. So, while there is some effect on Acr activity from altering the HTH, my major conclusion from these data would be that the HTH domain is mostly or completely unnecessary for Acr activity in the assays shown. The authors must clearly explain the caveats inherent in the *in vivo* assays, primarily that an amino acid substitution affecting a key residue for Acr activity would reduce plaquing levels by many orders of magnitude, not just 2 to 3-fold. The authors should show data similar to what is shown in the rebuttal (i.e. the reader must know that activity could be reduced by many orders of magnitude) so that the reader can evaluate the importance of a 2 to 3 fold drop in activity. My laboratory performs these types of phage plaquing assays extensively and we consider a 10-fold drop in activity on the borderline of being worth mentioning.

Our responses:

The reviewer raised a very good point. We agree with the reviewer that the HTH domain of AcrIF24 may play a minor role in its *in vivo* Acr activity. Following the reviewer's suggestion, first, we have included explanation for our data that approximately 10-fold reductions in activity, as seen in the experiments of AcrIF9 point mutations¹, should be the borderline for mutants of a key residue in phage plaquing assays. Secondly, we have included the Response figure 1 (Reconstitution of type I-F CRISPR system in *E. coli* Novablue (DE3) cells) in the last Response to Reviewers file as Supplementary Fig. 12 in the revised manuscript, to help the readers better evaluate the effects of the AcrIF24 mutations. Last but not least, as mentioned below, we have softened our conclusions about non-specific DNA-binding activity in the Acr activity of AcrIF24.

It is important to note that in the publication on AcrIF9, 3 different *in vivo* assays were performed in an effort to demonstrate an *in vivo* role for the residues involved in non-specific DNA-binding. It was also shown that the residues involved in non-specific DNA-binding were conserved in an alignment of diverse AcrIF9

homologues. Amino acid substitutions at these positions only affected non-specific DNA-binding and did not affect binding of the Acr to the Csy complex. Non-specific DNA-binding was shown to be induced by two diverse AcrIF9 homologues acting on two different Csy complexes. Finally, AcrIF9 is a small single domain protein with no recognizable DNA-binding domain and no DNA-binding activity on its own. All of these facts led to the conclusion that the non-specific DNA-binding activity of AcrIF9 has biological importance. In comparison, the evidence for the biological importance of the non-specific DNA-binding activity of AcrIF24 is very weak. The authors need to take these points into careful consideration and very much soften their conclusions about non-specific DNA-binding activity.

Our responses:

The reviewer raised a very good point. As the reviewer mentioned, in the AcrIF9 paper, different *in vivo* assays and experiments of two diverse AcrIF9 homologues have been performed to illustrate the biological importance of non-specific DNA-binding activity induced by AcrIF9, which does not have DNA-binding activity on its own¹. Compared with the study of AcrIF9, the evidence for the biological importance of non-specific DNA-binding activity is weaker for AcrIF24. However, a point worth mentioning is that, the non-specific DNA binding activity of AcrIF24 itself is actually fairly weak (lines 267-271 in our revised manuscript of last round, and revised Fig. 2b in the revised manuscript this time). However, the binding affinity for non-sequence-specific DNA of AcrIF24 could be greatly enhanced when it is complexed with the Csy complex (Please compare the last two lanes in both gels in Fig. 5a). Taken together, following the reviewer's suggestion, we have carefully considered these points and significantly softened the conclusions about the biological importance of non-specific DNA-binding activity of AcrIF24 in the revised manuscript.

Again, we wish to thank the reviewer for her/his constructive comments and suggestions, which have helped us to improve our manuscript greatly.

Reviewer #2 (Remarks to the Author):

All my comments have been carefully addressed.

The authors have performed additional experiments and revised the text.

I recommend publication of the revised manuscript.

We appreciate the reviewer's high evaluation of our work and recommendation of its publication.

Reviewer #3 (Remarks to the Author):

Yang et al. present an X-ray structure of AcrIF24 Δ MD, and cryo-EM structures of AcrIF24-Csy and AcrIF24-Csy-dsDNASP complex. Using a combination of structural and biochemical analyses, the authors showed binding of AcrIF24 to Csy complex by interacting with Cas7 subunits block the hybridization of DNA to the crRNA, and binding of AcrIF24 also induces the non-specific DNA binding to Csy complex, in which the PAM recognition motif and HTH motif plays an essential role. During revision, the author has addressed lots of points, but some of my major points have not been addressed.

We appreciate the reviewer's high evaluation of our manuscript and its improvement during the

revision. We have addressed all the comments listed below with additional experiments and have revised the manuscript accordingly, which we believe have further significantly improved our manuscript.

Major points 1:

During the revision, the author shows AcrIF24 repress its transcription by specifically interacting with IR23-24. In Fig. 2c, the HTH motif of AcrIF24 shows higher affinity with specific dsDNA sequence than non-specific dsDNA, and only binds non-specific dsDNA in a high concentration. However, in the following session, the author shows the HTH motif greatly contributes to the non-specific dsDNA binding. It's not supervised that HTH motif of AcrIF24 binds dsDNA non-specifically, but what's the affinity for this non-specific binding, most HTH motif could bind to dsDNA nonspecifically, is it meaningful *in vivo* in this case?

Our responses:

The reviewer raised a very good point. Following the reviewer's suggestion, we have measured the binding affinity between AcrIF24 and dsDNA_{NS} through EMSA as in Fig. 2b. As shown in the revised Fig. 2b in the revised manuscript, AcrIF24 binds dsDNA with a binding K_D of 3.461 μ M, much weaker than that of IR23-24 WT ($K_D = 20.18$ nM). As the reviewer suggested, most HTH motif could bind dsDNA nonspecifically, and the above assay indicated that AcrIF24 alone exhibits a fairly weak binding to dsDNA_{NS}. However, an interesting point about AcrIF24 is that its binding affinity for non-specific-sequence DNA could be greatly enhanced when it is complexed with the Csy complex (Please compare the last two lanes in both gels in Fig. 5a). Moreover, our *in vivo* phage plaquing assays also showed that two mutations in the HTH motif of AcrIF24, R196A and K197A, which display decreased non-sequence-specific DNA binding, are also compromised in the inhibition activity of AcrIF24 (Fig. 7e). Therefore, we think that the non-specific DNA binding activity of AcrIF24 HTH enhanced by the Csy complex may play a role in the *in vivo* activity of AcrIF24. However, due to the limited activity reductions by the AcrIF24 HTH mutants, we will soften the conclusion of the *in vivo* function of non-specific DNA-binding activity of AcrIF24 in the revised manuscript as another reviewer suggested.

Major points 2:

Also, the published paper about AcrIF9 (PMID: 33660777) shows that Csy:AcrIF9 binds dsDNANS and dsDNASP at an overlapping site, but ssDNASP is bound differently, the Csy:AcrIF9 complex binds specifically to ssDNA, but not nonspecifically as the author describes.

The author mentioned that Csy-AcrIF24 binds ssDNA non-specifically, have the author tested of the ability of the non-specific ssDNA for competing with specific ssDNA?

Regarding to my first and second questions previously raised, I am not convinced. Could the author provide the affinity parameters of Csy-AcrIF9 towards dsDNASP, dsDNANT, ssDNASP and ssDNANT respectively? These parameters will help to understand the binding mode of Csy-AcrIF9 to the targets both *in vitro* and *in vivo*.

Our responses:

This is a very good point. We would like to thank the reviewer for pointing this out and apologize for mistakenly describing the interaction between ssDNA_{SP} and Csy-AcrIF9 as nonspecifically in the last Response to Reviewers file.

We have tested the ability of the non-specific ssDNA for competing with specific ssDNA on AcrIF24-Csy in Fig. 5f in the original manuscript. The result showed that ssDNA_{NS} cannot compete ssDNA_{SP} off the Csy-AcrIF24 complex, reminiscent of AcrIF9, suggesting that Csy-AcrIF24 may also bind ssDNA_{SP} specifically as Csy-AcrIF9 does. However, we have not included this interpretation in our original manuscript, which may mislead the readers to think that Csy-AcrIF24 binds ssDNA also non-specifically as dsDNA. We have revised the corresponding interpretation parts in the revised manuscript to indicate that Csy-AcrIF24 binds ssDNA_{SP} specifically as Csy-AcrIF9 does.

Following the reviewer's suggestion, we have performed binding assays to investigate the binding of Csy-AcrIF9 towards dsDNA_{SP}, dsDNA_{NS}, ssDNA_{SP} and ssDNA_{NS}. Moreover, we also performed the binding of the four types of DNA molecules with Csy-AcrIF24 to compare with that of Csy-AcrIF9. The MST (microscale thermophoresis) assay suggested that the binding modes of the four types of DNA molecules are similar for Csy-AcrIF24 and Csy-AcrIF9 (revised Supplementary Fig. 7a,b). While each Csy-Acr complex displays comparable binding affinities for dsDNA_{SP} and dsDNA_{NS}, it displays the highest binding affinity for ssDNA_{SP}, approximately 10-fold higher than that of ssDNA_{NS}. This is consistent with the results of the competition experiments using dsDNA_{SP}-dsDNA_{NS} pair (Fig. 5e) and ssDNA_{SP}-ssDNA_{NS} pair (Fig. 5f) for Csy-AcrIF24 and the corresponding experiments for Csy-AcrIF9 complex¹, again suggesting that both Csy-AcrIF24 and Csy-AcrIF9 complexes bind specifically to ssDNA_{SP}. Notably, the Csy-AcrIF9 complex displays a slightly higher binding affinity than Csy-AcrIF24 for each of the four types of DNA molecules. We have included these new results and discussions into the revised manuscript.

Supplementary Fig. 7 MST assays of the binding of four types of DNA molecules to Csy-AcrIF24 and Csy-AcrIF9

Means are indicated by circles and individual values from three independent experiments are shown. Error bars represent SD. Binding curves and K_D values are also shown.

Major points 3:

In Fig. 6d, the specific target DNA is split upon binding to Csy-AcrIF24 (Fig. 6c and 6d), with missing DNA density between Csy and AcrIF24, there are not enough convincing evidence for supporting the proposed model. It would be useful to get the complex structure of Csy-AcrIF24-dsDNA_{NS} for investigating the AcrIF24-mediated inhibition mechanisms by non-specific DNA binding.

Our responses:

The reviewer raised a very good point. We have solved the complex structure of Csy-AcrIF24-dsDNA_{NS} (Fig. 6f, Supplementary Fig. 9 and Supplementary Table 3), which also revealed three discontinuous stretches of density for DNA, two near the hook domains of the two Cas8f subunits and one over the AcrIF24 CTD. The cylindrical density above the AcrIF24 CTD in the Csy-AcrIF24-dsDNA_{NS} structure is similar with that in the cryo-EM map of Csy-AcrIF24-dsDNA_{SP}, which makes it difficult to determine the orientation of this segment of dsDNA. Meanwhile, in the Csy-AcrIF24-dsDNA_{NS} structure, the densities near the hook domains of both Cas8f subunits are also very weak, making it only possible to determine the orientation but not the sequence of the DNA segments (revised Fig. 6g). This is very different from the situation of the Csy-AcrIF24-dsDNA_{SP} structure, in which the density of DNA is much clearer and separation of the dsDNA at the A₁(TS)-T₁(NTS) position was also observed (Fig. 6d). Our explanation for the difference of the DNA densities near the hook domain of Cas8f between the two Csy-AcrIF24-dsDNA structures is as follows. It has been suggested that Cas8f has the ability to bind sequence-non-specific DNA molecules with its PAM recognition region². Since dsDNA_{NS} cannot be sequence-specifically bound by the Csy-dsDNA complex, different regions of dsDNA_{NS} molecules may be bound by the hook domain of Cas8f subunits in different Csy-AcrIF24 complexes, thus blurring the DNA density in this region in the Csy-AcrIF24-dsDNA_{NS} structure. For dsDNA_{SP}, however, while the PAM region of the dsDNA could be recognized by the Cas8f subunit, the presence of AcrIF24 in the Csy-AcrIF24 complex precludes further hybridization between the target strand and the crRNA. Despite of the density differences between the dsDNA_{SP} and dsDNA_{NS} in the Cas8f hook domain, structural superimposition between the two complexes showed that dsDNA_{NS} and dsDNA_{SP} share a similar binding site near the hook domain of Cas8f (Fig. 6h).

Combining the structures of Csy-AcrIF24 complexed with both types of dsDNA molecules, we would like to make the following revision to the model in Fig. 7f. Based on the two structures, there is no evidence showing that one dsDNA molecule is binding with both Cas8f and the HTH domain of AcrIF24 simultaneously. Therefore, we now put three stretches of dsDNA in the model, as was observed in the two structures. This is also supported by the EMSA results shown in Fig. 7. Since the model is illustrating the inhibition mechanism of AcrIF24 by non-specific DNA binding, the two strands of the dsDNA in the hook domains are not separated (revised Fig. 7f).

Again, we wish to thank the reviewer for her/his constructive comments and suggestions, which have helped us to improve our manuscript greatly.

References

1. Lu, W.T., Trost, C.N., Muller-Esparza, H., Randau, L. & Davidson, A.R. Anti-CRISPR AcrIF9 functions by inducing the CRISPR-Cas complex to bind DNA non-specifically. *Nucleic Acids Res* **49**, 3381-3393

(2021).

2. Rollins, M.F., Schuman, J.T., Paulus, K., Bukhari, H.S. & Wiedenheft, B. Mechanism of foreign DNA recognition by a CRISPR RNA-guided surveillance complex from *Pseudomonas aeruginosa*. *Nucleic Acids Res* **43**, 2216-22 (2015).

REVIEWERS' COMMENTS

Reviewer #1 (Remarks to the Author):

The authors have only partially addressed my previous criticism, which pertained to the interpretation of the non-specific DNA-binding activity of AcrIF24. They toned down their interpretation of the in vivo data in the results section as follows:

“Therefore, the results here may suggested (should be suggest) that the non-specific DNA-binding activity of AcrIF24 does contribute to its inhibitory activity but may play a minor role.”

--The emphasis here and in the rest of the paper should really be on “minor role”.

However, the statement in the abstract on this subject is more vague:

“AcrIF24 mutations with impaired non-specific DNA binding also decrease its inhibition capacity.”

--This sentence would certainly lead a reader to think that the non-specific DNA-binding activity is functionally important.

They also finish the abstract with:

“our findings.....shed light on the potential universality of Acr-induced non-specific DNA binding and convergent evolution of Acrs”

--This sentence does not make sense. What does “potential universality” mean. It sounds like the authors are saying that all Acrs induce non-specific binding. I don't see how this is convergent evolution when the importance of the non-specific binding of AcrIF24 is minimal.

In the discussion, they state:

“Notably, while AcrIF24 displays no sequence or structural similarities to AcrIF9, they share a similar function in Acr-triggered interactions with non-sequence-specific dsDNA. This suggests that sequestering the surveillance system through non-productive associations with dsDNA might be a common mechanism shared by Acrs.”

--Again, this is misleading because the non-specific binding activity of AcrIF24 does not appear to be very important.

I am sorry to belabor this point about the non-specific binding activity of AcrIF24. However, it is crucial to note that this Acr is fundamentally different from AcrIF9, and here they are being presented as almost the same thing. Most important, AcrIF24 is a 2 domain protein, one domain has Acr activity and the other is clearly functioning as an Aca. It has been shown that Aca repressor function is crucial for phage viability. So AcrIF24 has 2 domains, each with an obvious important function with the function of the HTH domain being transcriptional repression. DNA-binding domains naturally have some non-specific binding activity. The type I-F CRISPR-Cas system also has a large non-specific DNA-binding surface that is required for stabilizing the D-loop. Placing two non-specific DNA-binding surfaces in proximity in the Csy:AcrIF24 complex creates a large non-specific binding surface, which imparts non-specific DNA-binding activity. I fundamentally do not believe that there was an evolutionary drive for AcrIF24 to acquire the ability to make the Csy complex bind DNA non-specifically. It is much more likely that evolutionary selection was for the Aca activity imparted by the HTH domain. The resulting non-specific binding activity is an adventitious result of adding a DNA-binding domain to the C-terminus of this Acr. The data presented on the function of the HTH domain in imparting Acr activity is not strong enough to argue against the simplest hypothesis to explain the observations.

In contrast to AcrIF24, AcrIF9 is a single small domain that encompasses a Csy complex binding surface, and a non-specific DNA-binding surface. Residues involved in non-specific DNA-binding are conserved and substituting them causes *in vivo* effects in three different assays. Importantly, one *in vivo* assay shows that the Csy:AcrIF9 complex still binds its specific target when non-specific DNA-binding is eliminated, illustrating a large *in vivo* impact of non-specific binding. So, the argument for the functional importance of the non-specific DNA-binding activity of AcrIF9 is much stronger.

Although I feel strongly about the point above, I am not meaning to say that the observations here about non-specific binding are uninteresting. It is very interesting that another Acr is able to impart such strong non-specific DNA-binding activity. It is very important, though, for the authors to emphasize that the functional importance of this non-specific DNA binding is unproven. And they should clearly state, in the paper, the major caveats in comparing AcrIF24 to AcrIF9. The abstract must also explicitly reflect these caveats.

Reviewer #3 (Remarks to the Author):

This manuscript has been improved with the additional experiments and revised descriptions. The authors have addressed all my concerns.

Reviewer #1 (Remarks to the Author):

The authors have only partially addressed my previous criticism, which pertained to the interpretation of the non-specific DNA-binding activity of AcrIF24. They toned down their interpretation of the in vivo data in the results section as follows:

“Therefore, the results here may suggested (should be suggest) that the non-specific DNA-binding activity of AcrIF24 does contribute to its inhibitory activity but may play a minor role.”

--The emphasis here and in the rest of the paper should really be on “minor role”.

We agree with the reviewer that we should keep the consistency of the entire manuscript on the minor role of non-specific DNA-binding in the inhibitory activity of AcrIF24. We have changed “suggested” to “suggest”.

However, the statement in the abstract on this subject is more vague:

“AcrIF24 mutations with impaired non-specific DNA binding also decrease its inhibition capacity.”

--This sentence would certainly lead a reader to think that the non-specific DNA-binding activity is functionally important.

Our responses:

The reviewer raised a good point. We have deleted this sentence.

They also finish the abstract with:

“our findings.....shed light on the potential universality of Acr-induced non-specific DNA binding and convergent evolution of Acrs”

--This sentence does not make sense. What does “potential universality” mean. It sounds like the authors are saying that all Acrs induce non-specific binding. I don't see how this is convergent evolution when the importance of the non-specific binding of AcrIF24 is minimal.

Our responses:

The reviewer raised a good point. We have deleted the words such as “potential universality” and “convergent evolution”, and revised this sentence as “our findings highlight a multifunctional Acr and suggest potential wide distribution of Acr-induced non-specific DNA binding”.

In the discussion, they state:

“Notably, while AcrIF24 displays no sequence or structural similarities to AcrIF9, they share a similar function in Acr-triggered interactions with non-sequence-specific dsDNA. This suggests that sequestering the surveillance system through non-productive associations with dsDNA might be a common mechanism shared by Acrs.”

--Again, this is misleading because the non-specific binding activity of AcrIF24 does not appear to be very important.

Our responses:

The reviewer raised a good point. We have revised this part as “Notably, while AcrIF24 displays no sequence or structural similarities to AcrIF9, they both induce Acr-triggered interactions with non-sequence-specific dsDNA in the Csy complex. However, while this activity has been proved important in the Acr activity of AcrIF9, it only plays a minor role in the Acr activity of AcrIF24. Therefore, the functional importance of this non-specific DNA binding is largely unproven for AcrIF24.”.

I am sorry to belabor this point about the non-specific binding activity of AcrIF24. However, it is crucial to note that this Acr is fundamentally different from AcrIF9, and here they are being presented as almost the same thing. Most important, AcrIF24 is a 2 domain protein, one domain has Acr activity and the other is clearly functioning as an Aca. It has been shown that Aca repressor function is crucial for phage viability. So AcrIF24 has 2 domains, each with an obvious important function with the function of the HTH domain being transcriptional repression. DNA-binding domains naturally have some non-specific binding activity. The type I-F CRISPR-Cas system also has a large non-specific DNA-binding surface that is required for stabilizing the D-loop. Placing two non-specific DNA-binding surfaces in proximity in the Csy:AcrIF24 complex creates a large non-specific binding surface, which imparts non-specific DNA-binding activity. I fundamentally do not believe that there was an evolutionary drive for AcrIF24 to acquire the ability to make the Csy complex bind DNA non-specifically. It is much more likely that evolutionary selection was for the Aca activity imparted by the HTH domain. The resulting non-specific binding activity is an adventitious result of adding a DNA-binding domain to the C-terminus of this Acr. The data presented on the function of the HTH domain in imparting Acr activity is not strong enough to argue against the simplest hypothesis to explain the observations.

Our responses:

We agree with the reviewer that the non-specific DNA-binding activity in the Csy-AcrIF24 complex plays a minor role in anti-CRISPR activity but might be an adventitious result of putting two non-specific DNA-binding surfaces close to each other. We have checked throughout the entire manuscript and revised it to keep consistency of this notion.

In contrast to AcrIF24, AcrIF9 is a single small domain that encompasses a Csy complex binding surface, and a non-specific DNA-binding surface. Residues involved in non-specific DNA-binding are conserved and substituting them causes in vivo effects in three different assays. Importantly, one in vivo assay shows that the Csy:AcrIF9 complex still binds its specific target when non-specific DNA-binding is eliminated, illustrating a large in vivo impact of non-specific binding. So, the argument for the functional importance of the non-specific DNA-binding activity of AcrIF9 is much stronger.

Our responses:

We agree with the reviewer that the non-specific DNA-binding activity of AcrIF9 plays a much stronger role in the Acr activity.

Although I feel strongly about the point above, I am not meaning to say that the observations here about non-specific binding are uninteresting. It is very interesting that another Acr is able to impart such strong non-specific DNA-binding activity. It is very important, though, for the authors to emphasize that the functional importance of this non-specific DNA binding is unproven. And they should clearly state, in the paper, the major caveats in comparing AcrIF24 to AcrIF9. The abstract must also explicitly reflect these caveats.

Our responses:

The reviewer raised a good point. We have emphasized that the functional importance of this non-specific DNA binding by AcrIF24 is unproven and that AcrIF9 is different from AcrIF24 in the functional importance of this non-specific DNA-binding activity.

Again, we wish to thank the reviewer for her/his constructive comments and suggestions, which have helped us to improve our manuscript greatly.

Reviewer #3 (Remarks to the Author):

This manuscript has been improved with the additional experiments and revised descriptions. The authors have addressed all my concerns.

We appreciate the reviewer's high evaluation of the improvement of our manuscript during the revision.